# Mitochondrial protein import clogging as a mechanism of disease

Liam P Coyne[1], Xiaowen Wang[1], Jiyao Song[2,3], Ebbing de Jong[4], Karin Schneider[5], Paul T Massa[5,6], Frank A Middleton[1,7], Thomas Becker[3], Xin Jie Chen[1,7]*

[1]Department of Biochemistry and Molecular Biology, State University of New York Upstate Medical University, Syracuse, United States; [2]Institute of Biochemistry and Molecular Biology, Faculty of Medicine, University of Freiburg, Freiburg, Germany; [3]Institute of Biochemistry and Molecular Biology, Faculty of Medicine, University of Bonn, Bonn, Germany; [4]Proteomics and Mass Spectrometry Core Facility, State University of New York Upstate Medical University, Syracuse, United States; [5]Department of Microbiology and Immunology, State University of New York Upstate Medical University, Syracuse, United States; [6]Department of Neurology, State University of New York Upstate Medical University, Syracuse, United States; [7]Department of Neuroscience and Physiology, State University of New York Upstate Medical University, Syracuse, United States

*For correspondence:
chenx@upstate.edu

Competing interest: The authors declare that no competing interests exist.

**Abstract** Mitochondrial biogenesis requires the import of >1,000 mitochondrial preproteins from the cytosol. Most studies on mitochondrial protein import are focused on the core import machinery. Whether and how the biophysical properties of substrate preproteins affect overall import efficiency is underexplored. Here, we show that protein traffic into mitochondria can be disrupted by amino acid substitutions in a single substrate preprotein. Pathogenic missense mutations in ADP/ATP translocase 1 (ANT1), and its yeast homolog ADP/ATP carrier 2 (Aac2), cause the protein to accumulate along the protein import pathway, thereby obstructing general protein translocation into mitochondria. This impairs mitochondrial respiration, cytosolic proteostasis, and cell viability independent of ANT1's nucleotide transport activity. The mutations act synergistically, as double mutant Aac2/ANT1 causes severe clogging primarily at the translocase of the outer membrane (TOM) complex. This confers extreme toxicity in yeast. In mice, expression of a super-clogger ANT1 variant led to neurodegeneration and an age-dependent dominant myopathy that phenocopy ANT1-induced human disease, suggesting clogging as a mechanism of disease. More broadly, this work implies the existence of uncharacterized amino acid requirements for mitochondrial carrier proteins to avoid clogging and subsequent disease.

## Editor's evaluation

This manuscript describes important insight into the molecular mechanism by which destabilized mitochondrial proteins 'clog' import channels and contribute to the pathologic mitochondrial and cellular dysfunction implicated in human disease. The evidence supporting this conclusion is convincing, utilizing yeast, mammalian cell culture, and mouse models. This work, which defines an interesting mechanism of disease pathogenesis, will be of broad interest to researchers in the fields of mitochondrial biology, protein quality control and proteostasis.

**eLife digest** Inside our cells, compartments known as mitochondria generate the chemical energy required for life processes to unfold. Most of the proteins found within mitochondria are manufactured in another part of the cell (known as the cytosol) and then imported with the help of specialist machinery. For example, the TOM and TIM22 channels provide a route for the proteins to cross the two membrane barriers that separate the cytosol from the inside of a mitochondrion.

ANT1 is a protein that is found inside mitochondria in humans, where it acts as a transport system for the cell's energy currency. Specific mutations in the gene encoding ANT1 have been linked to degenerative conditions that affect the muscles and the brain. However, it remains unclear how these mutations cause disease.

To address this question, Coyne et al. recreated some of the mutations in the gene encoding the yeast equivalent of ANT1 (known as Aac2). Experiments in yeast cells carrying these mutations showed that the Aac2 protein accumulated in the TOM and TIM22 channels, creating a 'clog' that prevented other essential proteins from reaching the mitochondria. As a result, the yeast cells died. Mutant forms of the human ANT1 protein also clogged up the TOM and TIM22 channels of human cells in a similar way.

Further experiments focused on mice genetically engineered to produce a "super-clogger" version of the mouse equivalent of ANT1. The animals soon developed muscle and neurological conditions similar to those observed in human diseases associated with ANT1.

The findings of Coyne et al. suggest that certain genetic mutations in the gene encoding the ANT1 protein cause disease by blocking the transport of other proteins to the mitochondria, rather than by directly affecting ANT1's nucleotide trnsport role in the cell. This redefines our understanding of diseases associated with mitochondrial proteins, potentially altering how treatments for these conditions are designed.

## Introduction

Mitochondria are essential organelles responsible for a wide range of cellular functions. To carry out these functions, they are equipped with a proteome of 1,000–1,500 proteins (*Sickmann et al., 2003*; *Pagliarini et al., 2008*; *Morgenstern et al., 2017*; *Morgenstern et al., 2021*). The vast majority of these proteins are encoded by the nuclear genome, synthesized in the cytosol, and sorted into one of the four mitochondrial sub-compartments, namely the outer mitochondrial membrane (OMM), inter-membrane space (IMS), inner mitochondrial membrane (IMM), and the matrix (*Neupert and Herrmann, 2007*; *Endo and Yamano, 2009*; *Chacinska et al., 2009*; *Wiedemann and Pfanner, 2017*). The entry gate by which >90% of mitochondrial proteins enter mitochondria is the translocase of the outer membrane (TOM) complex. Therefore, proper function of the TOM complex is paramount for mitochondrial function and cell viability.

After passage through the TOM complex, specialized protein translocases transport preproteins into the mitochondrial sub-compartments (*Neupert and Herrmann, 2007*; *Endo and Yamano, 2009*; *Chacinska et al., 2009*; *Wiedemann and Pfanner, 2017*; *de Marcos-Lousa et al., 2006*). The import of mitochondrial carrier proteins to the protein-dense IMM is particularly challenging. Molecular chaperones like Hsp70 and Hsp90 target these highly hydrophobic proteins through the cytosol to the Tom70 receptor, which is associated with the TOM complex on the mitochondrial surface (*Young et al., 2003*; *Bhangoo et al., 2007*). After transport through the channel-forming Tom40 subunit of the TOM complex, the heterohexameric small TIM chaperones (i.e. the Tim9-Tim10 complex) transport the preprotein to the carrier translocase of the inner membrane (TIM22 complex) (*Koehler et al., 1998*; *Sirrenberg et al., 1998*; *Wiedemann et al., 2001*; *Truscott et al., 2002*; *Webb et al., 2006*; *Weinhäupl et al., 2018*; *Ellenrieder et al., 2019*). The carrier translocase is a multisubunit complex that inserts carrier proteins into the IMM in a membrane potential ($\Delta \phi$)-dependent manner (*Sirrenberg et al., 1996*; *Kerscher et al., 1997*; *Kovermann et al., 2002*; *Rehling et al., 2003*; *Zhang et al., 2021*; *Qi et al., 2021*). The central subunit Tim22 integrates carrier proteins into the IMM (*Rehling et al., 2003*; *Peixoto et al., 2007*). It associates with different partner proteins in yeast and human mitochondria. In yeast, Tim18 and Sdh3 are required for assembly and stability of the carrier translocase, whereas Tim54 tethers the small TIM chaperones (Tim9-Tim10-Tim12 complex) to the

translocase (*Wagner et al., 2008*; *Gebert et al., 2011*). In human cells, TIM22 associates with the acylglycerol kinase (AGK) and TIM29, which are both required for full import capacity (*Kang et al., 2016*; *Callegari et al., 2016*; *Kang et al., 2017*; *Vukotic et al., 2017*). Human TIM22 also associates with the TIM9-TIM10 complex (*Qi et al., 2021*).

Several diseases are associated with mutations directly affecting the protein import machinery of the carrier pathway. For example, mutations in *TIMM8A* have been found in patients suffering Mohr-Tranebjaerg syndrome/deafness dystonia syndrome (*Koehler et al., 1999*). Mutations in *TOMM70* and *TIMM22* have been linked to a devastating neurological syndrome and a mitochondrial myopathy, respectively (*Pacheu-Grau et al., 2018*; *Dutta et al., 2020*; *Wei et al., 2020*). Mutations in another TIM22 complex subunit, AGK, has been associated with the Senger's syndrome marked by cataracts, hypertrophic cardiomyopathy, skeletal myopathy, and exercise intolerance (*Kang et al., 2017*; *Vukotic et al., 2017*; *Mayr et al., 2012*). Clearly, defective protein import is linked to neurological and musculoskeletal diseases. However, whether protein import defects contribute to diseases not directly related to mutations in the core protein import machinery is unclear.

Here, we show that pathogenic missense mutations in a mitochondrial carrier protein, ADP/ATP translocase 1 (ANT1) or ADP/ATP carrier 2 (Aac2) in yeast, cause arrest of the protein at the translocases during import into mitochondria. This effectively 'clogs' the protein import pathway to obstruct general protein import and is associated with muscle and neurological disease in mice. Our findings demonstrate that global protein import is vulnerable to missense mutations in mitochondrial preproteins, and also provide strong evidence that protein import clogging contributes to neurological and muscular syndromes caused by dominant mutations in Ant1 (*Kaukonen et al., 2000*; *Siciliano et al., 2003*; *Simoncini et al., 2017*).

## Results

### Super-toxic Aac2p mutants dominantly kill cells

We previously showed that four pathogenic ANT1 variants modeled in the yeast Aac2p (*Figure 1A*) share numerous dominant phenotypes including cold sensitivity, mitochondrial DNA (mtDNA) instability, a propensity to misfold inside mitochondria, and hypersensitivity to low $\Delta\psi$ conditions (*Wang et al., 2008*; *Liu et al., 2015*; *Coyne and Chen, 2019*). We reasoned that if the mutant proteins share a common mechanism of toxicity that drives these phenotypes, then combining mutations into a single protein may enhance toxicity. To test this, we transformed the wild-type M2915-6A yeast strain with centromeric plasmids expressing wild-type, single and double mutant *aac2* alleles, and selected for Ura$^+$ transformants on glucose medium. We found that transformants expressing *aac2$^{M114P,A128P}$* and *aac2$^{M114P,A137D}$* formed smaller colonies on the selective medium at 25°C relative to wild-type *AAC2* and single mutant alleles (*Figure 1B*). Strikingly, transformants expressing *aac2$^{A106D,M114P}$*, *aac2$^{A106D,A128P}$*, *aac2$^{A106D,A137D}$*, and *aac2$^{A128P,A137D}$* were unable to form visible colonies at 25°C. Growth of transformants expressing some of the double mutants were improved at 30°C (*Figure 1—figure supplement 1A*). These data suggest that combining missense mutations into a single Aac2 protein increases toxicity, even when expressed from a centromeric vector.

We integrated a single copy of *aac2$^{A128P,A137D}$* into the genome of both *AAC2* and *aac2Δ* strains in the W303-1B background. We chose this strain background for two reasons: first, it is more tolerant of mutant *aac2* expression (*Wang et al., 2008*); and second, mutant *aac2* expression is not $\rho^0$-lethal in this background (like it is in M2915-6A and BY4741), meaning that cells expressing mutant *aac2* alleles can survive the loss of mtDNA. This allows us to score mtDNA instability via quantitation of smaller white colonies ('petites') that form when mtDNA is mutated or depleted (*Chen and Clark-Walker, 2000*). We found that growth of cells co-expressing *aac2$^{A128P,A137D}$* and *AAC2* is reduced on glucose medium (*Figure 1C*), and they form petites at a much higher frequency compared with those expressing the *aac2$^{A128P}$* and *aac2$^{A137D}$* single mutant alleles (*Figure 1D–E*). In the *aac2Δ* background, neither Aac2p$^{A137D}$ nor Aac2p$^{A128P,A137D}$ supported respiratory growth (*Figure 1C*), consistent with the A123D/A137D mutation eliminating nucleotide transport activity (*Palmieri et al., 2005*). Cells expressing only *aac2$^{A128P,A137D}$* were barely viable, and cell growth was completely inhibited at 25°C even on glucose medium (*Figure 1C*). This is in sharp contrast to the *AAC2*-null strain, which strongly supports a dominant effect of *aac2$^{A128P,A137D}$* on cell viability.

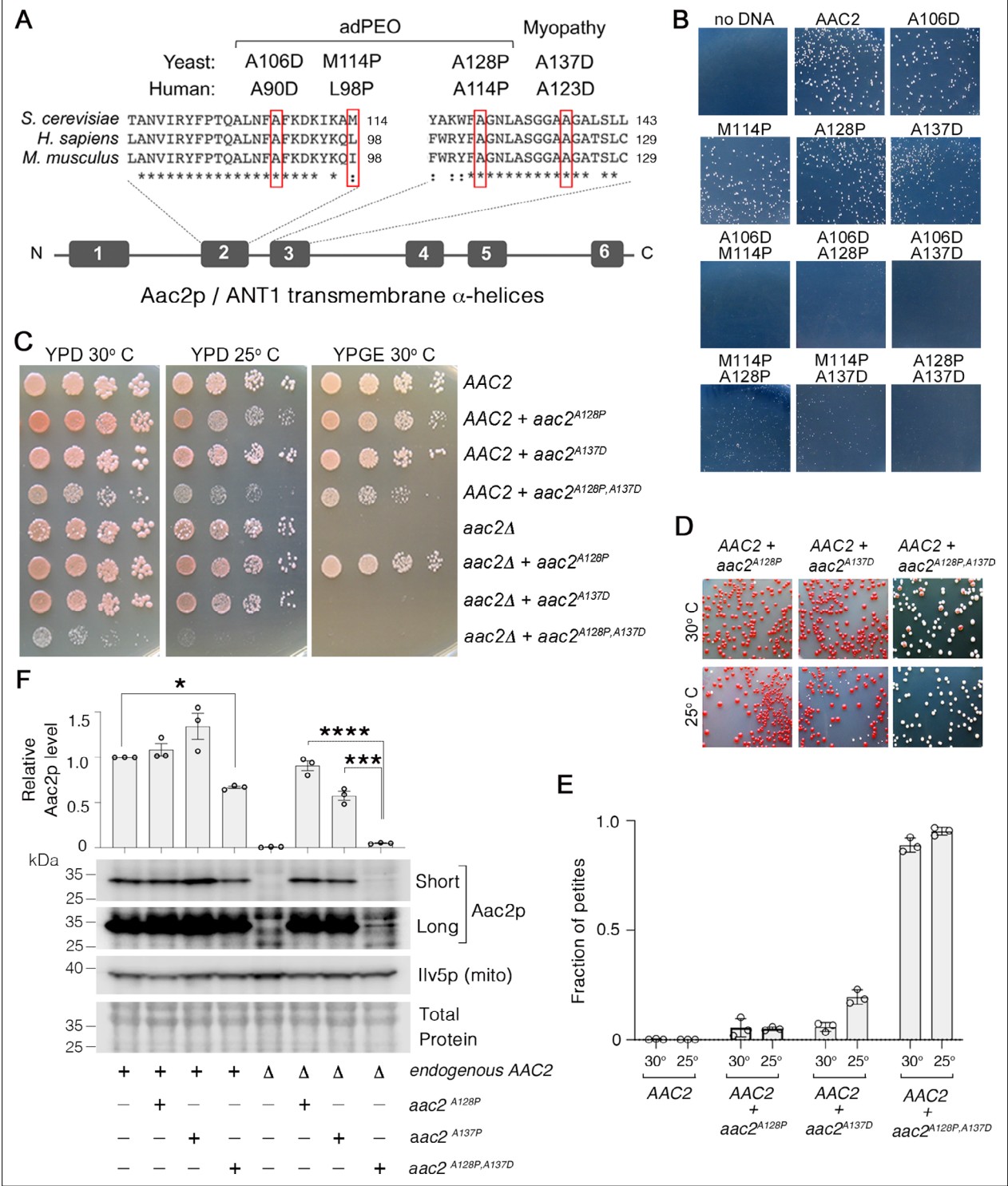

**Figure 1.** Super-toxic Aac2p mutants dominantly kill cells. (**A**) Schematic showing the location of pathogenic mutations in transmembrane α-helices 2 and 3 of the ADP/ATP translocator in human (ANT1) compared with mouse (Ant1) and yeast (Aac2p). adPEO, autosomal dominant progressive external ophthalmoplegia. (**B**) Expression of double mutant *aac2* alleles is highly toxic. The yeast M2915-6A strain was transformed with the centromeric vector pRS416 (*URA3*) expressing wild-type or mutant *aac2* alleles and transformants were grown on selective glucose medium lacking uracil at 25°C for 3 days. (**C**) Growth of yeast cells after serial dilution, showing dominant toxicity of *aac2^A128P,A137D* that is integrated into the genome in the W303-1B strain background. YPD, yeast peptone dextrose medium; YPGE, yeast peptone glycerol ethanol medium. (**D**) The *aac2^A128P,A137D* allele dominantly increases the frequency of 'petite' colonies, which are white. This indicates mitochondrial DNA (mtDNA) destabilization. (**E**) 'Petite' frequencies of yeast strains expressing the mutant alleles of *aac2*. The strains were first grown in YPD medium for 24 hr before being plated on YPD medium for scoring 'petite'

*Figure 1 continued on next page*

*Figure 1 continued*

colonies. (**F**) Immunoblot analysis showing extremely low levels of Aac2p[A128P,A137D] (lower panels) Ilv5p was used as a loading control for mitochondrial protein. Total protein determined with total protein stain (LI-COR). Short, short exposure; Long, long exposure. Upper panel, quantitation from three independent experiments. Aac2p values were normalized by Ilv5p to control for mitochondrial content, and data were represented as relative to wild-type; * indicates p<0.05, ***p<0.001, ****p<0.0001 from one-way ANOVA with Tukey's multiple comparisons test. Data represented as mean ± SEM.

The online version of this article includes the following source data and figure supplement(s) for figure 1:

**Source data 1.** Uncropped Western blots from *Figure 1F*.

**Figure supplement 1.** Toxicity and low-level accumulation of double mutant Aac2 proteins.

**Figure supplement 1—source data 1.** Uncropped Western blots from *Figure 1—figure supplement 1B-C*.

Interestingly, we found that Aac2p[A128P,A137D] accumulates to just 4.7% of wild-type Aac2 levels (*Figure 1F*). We did not observe any accumulation of aggregated Aac2p[A128P,A137D] (*Figure 1—figure supplement 1B–C*), indicating that Aac2p[A128P,A137D] degradation (see below), rather than aggregation, is the likely explanation for low protein recovery. Taken together, the data indicate that *aac2[A128P,A137D]* imparts potent toxicity through a mechanism that is independent of nucleotide transport.

## Super-toxic Aac2 proteins clog the TOM complex

We hypothesized that the highly toxic double mutant Aac2p is arrested during mitochondrial protein import thereby clogging protein import. To test this, we imported [35]S-labeled Aac2p variants into wild-type mitochondria and analyzed the import reaction by blue native polyacrylamide gel electrophoresis (BN-PAGE) followed by autoradiography (*Figure 2A*; *Ellenrieder et al., 2019*; *Ryan et al., 1999*). We did not observe a significant reduction in the amount of IMM-inserted 'mature' Aac2p[A128P] or Aac2p[A137D] compared with wild-type Aac2p, suggesting no obvious defect for importing into wild-type mitochondria that are fully energized in vitro. In contrast, integration of the double mutant Aac2p[A128P,A137D] into the IMM was reduced by >70% (*Figure 2A–B*). To determine whether the mutant preprotein can enter mitochondria, we treated the import reaction with proteinase K to digest non-imported preproteins. We found that a significant portion of mutant Aac2p, particularly Aac2p[A128P,A137D], is sensitive to proteinase K digestion (*Figure 2C–D*). These data suggest that either the Aac2p[A128P,A137D] preprotein is not transported to the TOM complex, or it fails to fully traverse the TOM complex. To decipher between these possibilities, we imported Aac2p variants into mitochondria containing hemagglutinin (HA)-tagged Tom40 for affinity purification of the TOM complex. Indeed, Aac2p[A128P,A137D] had increased association with Tom40-HA compared with wild-type (*Figure 2E–F*). These observations suggest that a significant fraction of Aac2p[A128P,A137D] is arrested at the TOM complex during import into fully energized wild-type mitochondria.

Next we wondered whether Aac2p[A128P,A137D] clogs the TOM complex in vivo. If it does, we would expect the accumulation of unprocessed mitochondrial preproteins that travel through the TOM complex, which would include the vast majority of intramitochondrial proteins including both TIM22 and TIM23 substrates. Indeed, precursor of the mitochondrial matrix protein Hsp60p was readily detectable in cells expressing Aac2p[A128P,A137D] despite extremely low levels of the mutant protein (*Figure 2G*). We also acutely expressed *aac2* from galactose-inducible *GAL10* promoter to limit possible indirect effects on protein import efficiency. As expected, expression of mutant *aac2* from the *GAL10-* promoter was highly toxic (*Figure 2—figure supplement 1A*). Importantly, the timing of Hsp60p precursor accumulation (*Figure 2—figure supplement 1B*) was coupled with the induction of the mutant Aac2p (*Figure 2—figure supplement 1C*). The data suggest that the mutant Aac2p directly affects the import of other mitochondrial proteins in vivo.

We extended our analysis to include additional single and double mutant *aac2* alleles (see *Figure 1A*). In transformants expressing double mutant *aac2* alleles from a centromeric vector, the level of un-cleaved Hsp60p correlated with toxicity (*Figure 2H*, see also *Figure 1B* and *Figure 1—figure supplement 1A*). Moreover, total Aac2p levels were reduced by all double mutant Aac2p variants except the least toxic Aac2p[M114P,A137D] (*Figure 2H–I*). This suggests impaired biogenesis of endogenous wild-type Aac2p. The extent of endogenous Aac2 reduction also correlated with cell toxicity. These data further support the idea that protein import clogging underlies the super-toxicity of Aac2p double mutant.

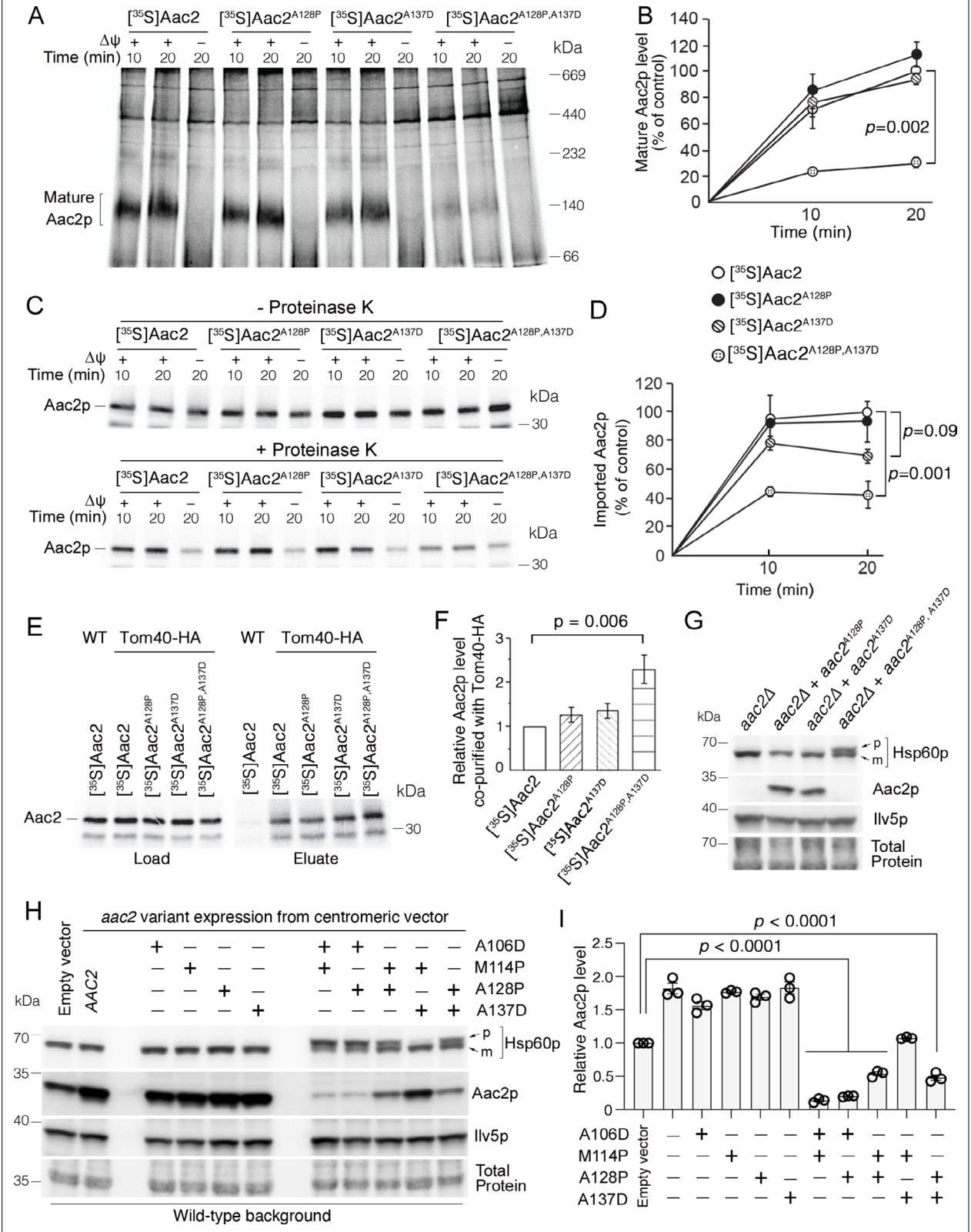

**Figure 2.** Super-toxic ADP/ATP carrier 2 (Aac2) proteins clog the translocase of the outer membrane (TOM) complex. (**A**) In vitro protein import assay. 35S-labeled Aac2p and mutant variants were imported into wild-type mitochondria for 10 or 20 min and analyzed by blue native electrophoresis and autoradiography. (**B**) Quantitation from three independent experiments depicted in (**A**). p-Value from two-way repeated measures ANOVA with Sidak's multiple comparisons test. (**C**) 35S-labeled Aac2p and mutant variants were imported into wild-type mitochondria without (upper) or with (lower panel)

*Figure 2 continued on next page*

*Figure 2 continued*

subsequent proteinase K treatment to degrade non-imported preproteins. Reaction analyzed by SDS-PAGE and autoradiography. (**D**) Quantitation from three independent experiments depicted in (**C**). p-Values were calculated as in (**B**). (**E**) Preferential association of mutant Aac2p with Tom40-HA. $^{35}$S-labeled Aac2p and mutant variants were imported into Tom40-HA mitochondria, followed by anti-HA immunoprecipitation and analysis by SDS-PAGE and autoradiography. (**F**) Quantitation from three independent experiments depicted in (**E**). p-Value was calculated with a one-way ANOVA and Dunnett's multiple comparisons test. (**G**) Immunoblot analysis showing accumulation of the un-cleaved precursor of Hsp60p (**p**) in cells expressing chromosomally integrated *aac2$^{A128P,A137D}$*. Cells were grown in YPD at 30°C. m, mature (i.e. cleaved). (**H**) Immunoblot analysis showing accumulation of un-cleaved Hsp60p precursor (**p**) in cells expressing *aac2* alleles from a centromeric vector. Cells were grown in yeast nitrogen-based dextrose media with supplemented casamino acids, lacking uracil at 25°C. (**I**) Quantitation from three replicates of (**H**). Aac2p values normalized by the mitochondrial protein Ilv5p, and then normalized to vector-transformed samples. p-Values were calculated as in (**F**). Data represented as mean ± SEM.

The online version of this article includes the following source data and figure supplement(s) for figure 2:

**Source data 1.** Uncropped autoradiographs (*Figure 2A,C,E*) and Western blots (*Figure 2G,H*).

**Figure supplement 1.** Acute expression of mutant Aac2p is toxic and impairs protein import.

**Figure supplement 1—source data 1.** Uncropped Western blots from *Figure 2—source data 1B-C*.

## Yme1p, but not the proteasome, contributes substantially to Aac2p$^{A128P,A137D}$ degradation

We hypothesized that the proteasome should degrade Aac2$^{A128P,A137D}$ as it clogs at the TOM complex. To test this, we first generated strains lacking the drug efflux pump Pdr5p, which allows accumulation of the proteasome inhibitor MG132 in yeast cells. Proteasome inhibition by MG132 was confirmed by the accumulation of Sml1p, a labile proteasome substrate (*Figure 3A*; *Andreson et al., 2010*). The experiment was performed in *aac2Δ* background to facilitate Aac2p$^{A128P,A137D}$ detection. Interestingly, we found that MG132 failed to increase the steady-state level of Aac2p$^{A128P,A137D}$ (*Figure 3A*). As an orthogonal approach, we tested whether proteasome dysfunction induced by two temperature-sensitive mutants, *ump1Δ* and *pre9Δ*, could increase Aac2p$^{A128P,A137D}$ levels, and observed no effect (*Figure 3B–C*). Cyclohexamide chase following a galactose-induced Aac2p$^{A128P,A137D}$ synthesis confirmed that the mutant protein is unstable compared with the wild-type Aac2p (*Figure 3D–E*). Consistently, MG132 failed to stabilize acutely induced Aac2p$^{A128P,A137D}$. These data suggest that little, if any, Aac2p$^{A128P,A137D}$ is degraded by the proteasome.

Next we tested whether an intramitochondrial proteolytic system is responsible for Aac2p$^{A128P,A137D}$ degradation. One likely candidate is Yme1p, which is an IMS-facing AAA protease whose genetic deletion is synthetically lethal with single mutant *aac2* variants (*Wang et al., 2008*). Indeed, we found that Aac2p$^{A128P,A137D}$ is significantly but not fully stabilized in cells disrupted of *YME1* (*Figure 3D–E*). Inhibition of proteasomal function with MG132 does not significantly stabilize Aac2p$^{A128P,A137D}$ in *yme1Δ* cells. These data strongly suggest that Yme1p plays an important role in the degradation of Aac2p$^{A128P,A137D}$, possibly serving as a quality control mechanism for degrading stalled import substrates in the vicinity of TIM22 and/or the TOM complex (*Wu et al., 2018*).

## Aac2p$^{A128P}$ accumulates along the carrier import pathway and induces protein import stress

The single mutant Aac2p variants did not show significantly reduced import in vitro and only a mild accumulation at the TOM complex (*Figure 2A–B; E-F*). However, acute overexpression of the single mutant Aac2p$^{A128P}$ led to accumulation of the precursor form of Hsp60p, consistent with clogging of the translocation apparatus (*Figure 2—figure supplement 1B*). To explore an arrest of Aac2p$^{A128P}$ at protein translocases, we performed affinity purification Aac2p-HIS$_6$ and Aac2p$^{A128P}$-HIS$_6$ followed by a quantitative proteomic comparison of the co-purified proteins (*Figure 4A–B*; *Figure 4—figure supplement 1A–C*). Numerous proteins preferentially co-purify with Aac2p$^{A128P}$-HIS$_6$ over Aac2p-HIS$_6$ (*Figure 4—source data 1*), which were enriched for chaperones involved in targeting of mitochondrial preproteins through the cytosol (*Figure 4C*; *Figure 4—figure supplement 1D–F*). Aac2p$^{A128P}$ also has increased association with the TOM complex, as well as the translocase of the inner membrane (TIM22) complex that is responsible for the $\Delta\psi$-dependent insertion of carrier proteins into the IMM (*Figure 4C*). The association with Tim22p was confirmed by immunoblotting (*Figure 4—figure supplement 1G–H*). Expectedly, there is no increased association of Aac2p$^{A128P}$ with Tim23p, which is known to associate with wild-type Aac2p and is not involved in its import (*Figure 4—figure supplement*

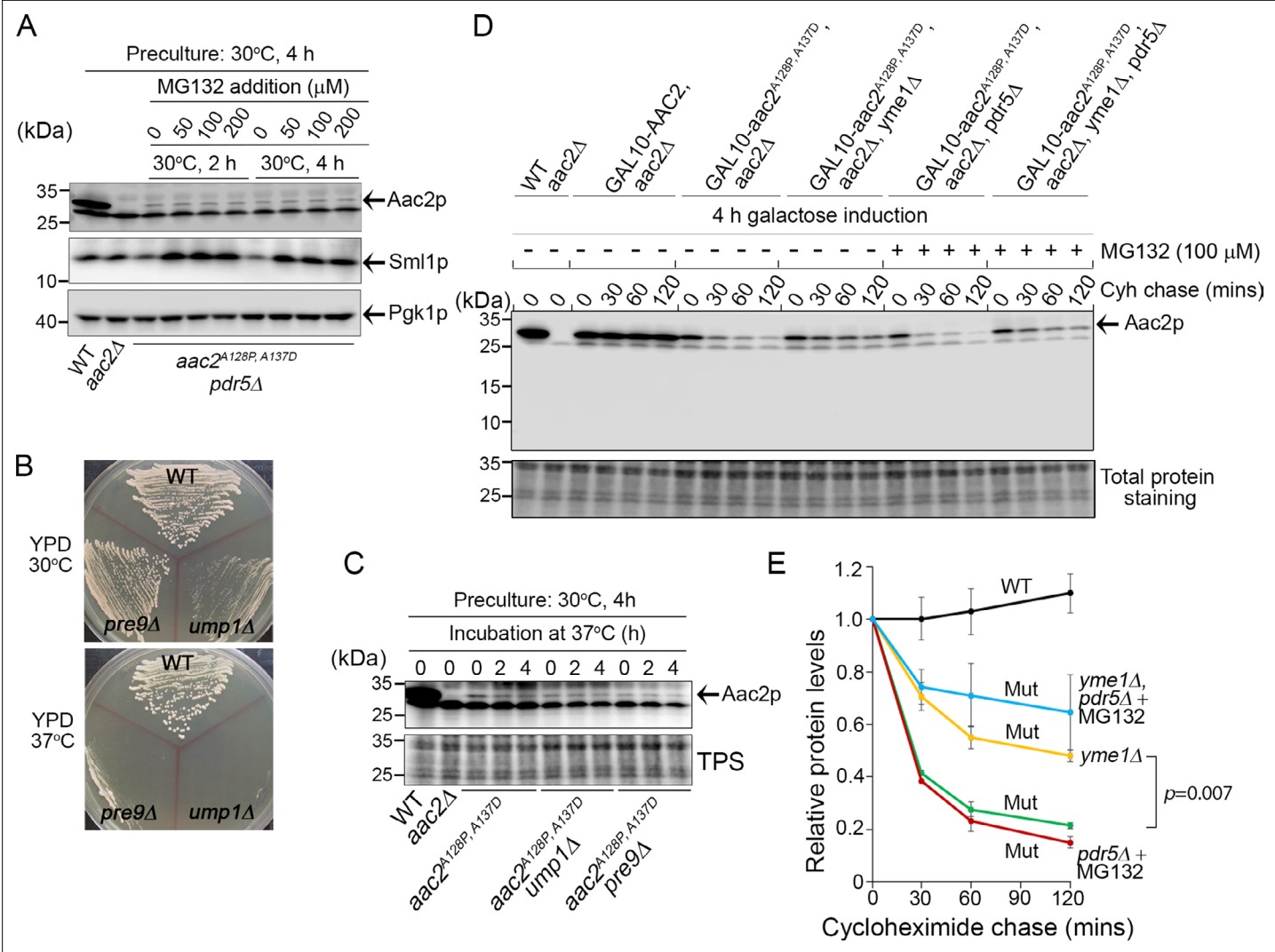

**Figure 3.** Degradation of Aac2p[A128P, A137D] by Yme1p. (**A**) Effect of MG132 on the steady-state level of Aac2p[A128P, A137D] in a strain disrupted of *PDR5* and *AAC2*. Cells were first grown in YPD medium at 30°C for 4 hr before MG132 was added at the indicated concentrations. Cells were cultured for another 2 or 4 hr before being harvested for western blot analysis. Sml1 was used as a control for proteasome inhibition. (**B**) Temperature-sensitive phenotype of *ump1Δ* and *pre9Δ* cells. Cells were grown at the indicated temperatures for 2 days before being photographed. (**C**) Western blot analysis showing that *ump1Δ* and *pre9Δ* do not affect the steady-state level of Aac2p[A128P, A137D]. Cells were grown in YPD medium for 2 and 4 hr at the restrictive temperature (37°C) before being analyzed for Aac2p[A128P, A137D] levels. TPS, total protein staining. (**D**) Western blot showing the stability of Aac2p[A128P, A137D] after cycloheximide (Cyh) chase in cells disrupted of *YME1* or treated with MG132, following *GAL10*-induced synthesis of Aac2p[A128P, A137D] in galactose medium at 30°C for 4 hr. (**E**) Quantification of data for the turnover rate of Aac2p[A128P, A137D] (Mut) and its wild-type control (WT) depicted in (**D**). Aac2 levels were first normalized by total protein stain and then plotted as values relative to time zero. Depicted are mean values ± SEM from three independent experiments. p-Value was calculated with a two-way repeated measures ANOVA with Tukey's multiple comparisons test to compare genotypes at time = 120 min.

The online version of this article includes the following source data for figure 3:

**Source data 1.** Uncropped Western blots from *Figure 3A,C,D*.

1I; *Dienhart and Stuart, 2008*). Repeat affinity purification, except with double the ionic strength in the binding buffer, reproduced these data (*Figure 4—figure supplement 2A–E*, *Figure 4—figure supplement 2—source data 1*).

To quantitate the increased association of Aac2p[A128P] to Tim22p relative to the wild-type Aac2p, we performed targeted quantitative proteomics (*Figure 4—figure supplement 2F*). We found that ~5 mmoles of Tim22p are pulled down per mole of Aac2p[A128P]-HIS[6] (*Figure 4D*). If 5 mmoles per mole of Aac2p[A128P] molecules are bound to Tim22p in vivo, this could theoretically occupy ~85% of Tim22p

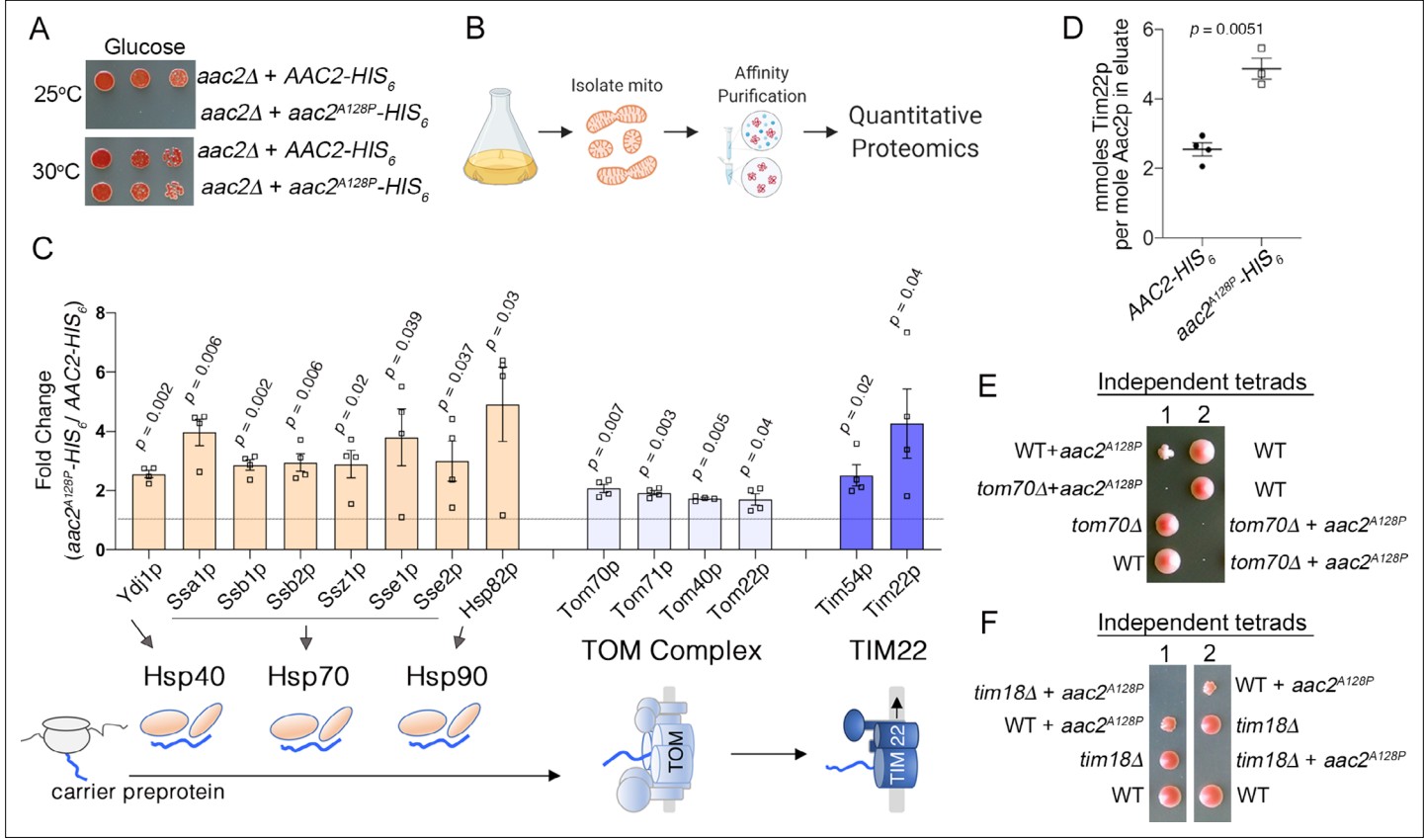

**Figure 4.** Aac2p[A128P] accumulates along the carrier import pathway and induces protein import stress. (**A**) Growth of cells after serial dilution showing that *aac2[A128P]*-HIS$_6$ is toxic at 25°C on glucose medium in an M2915-6A-derived strain. (**B**) Schematic of our approach to identify aberrant protein-protein interactions of Aac2p[A128P]-HIS$_6$. (**C**) Co-purified proteins significantly enriched in Aac2p[A128P]-HIS$_6$ eluate compared with Aac2p-HIS$_6$. See Materials and methods for details on abundance value calculation. FDR-corrected p-values depicted are from multiple *t* test analysis. Lower panel is a schematic of the mitochondrial carrier protein import pathway. (**D**) Absolute quantities of Aac2p and Tim22p in Aac2p-HIS$_6$ and Aac2p[A128P]-HIS$_6$ eluates, as determined by parallel reaction monitoring (PRM) mass spectrometry. p-Value was calculated with Student's *t* test. (**E–F**) Tetrad analysis demonstrating lethality of *aac2[A128P]* expression with genetic defects in carrier protein import in the M2915-6A strain background. Cell were grown on YPD at 30°C. Data depicted as mean ± SEM.

The online version of this article includes the following source data and figure supplement(s) for figure 4:

**Source data 1.** Proteins that preferentially interact with Aac2p[A128P]-HIS$_6$, compare with Aac2-HIS$_6$ in low-salt conditions.

**Figure supplement 1.** Affinity purification in low-salt conditions suggested that Aac2p[A128P] accumulates along the carrier protein import pathway.

**Figure supplement 1—source data 1.** Uncropped Western blots from *Figure 4—figure supplement 1A,G*.

**Figure supplement 2.** Affinity purification in high-salt conditions confirmed that Aac2p[A128P] accumulates along the carrier protein import pathway.

**Figure supplement 2—source data 1.** Proteins that preferentially interact with Aac2p[A128P]-HIS$_6$, compare with Aac2p-HIS$_6$ in high-salt conditions.

**Figure supplement 2—source data 2.** Uncropped Western blots from *Figure 4—figure supplement 2C*.

**Figure supplement 3.** Cellular responses to *aac2[A128P]* expression support mitochondrial protein import clogging.

channels, as Aac2p is present at ~188,000 molecules per cell and Tim22p at just 1100 (***Morgenstern et al., 2017***). Taken together, the data suggest that Aac2p[A128P] is stalled at both the TOM and TIM22 complexes during import.

Genetic and transcriptional analyses provided additional support for a clogging activity associated with Aac2p[A128P]. First, we observed that *aac2[A128P]*-expressing cells are sensitive to genetic perturbation to the carrier import pathway. *TOM70* and *TIM18* are two non-essential genes directly involved in the import of mitochondrial carrier proteins such as Aac2, serving as an import receptor at the TOM complex and a component of the TIM22 complex, respectively (***Young et al., 2003***; ***Wagner et al., 2008***). We found that meiotic segregants combining *aac2[A128P]* expression with *tom70Δ* or *tim18Δ* form barely visible microcolonies at 30°C on glucose medium (***Figure 4E–F***). It was recently proposed

that Tom70p's primary function is to recruit chaperones to the mitochondrial surface to prevent proteotoxicity in this protein-dense region (*Backes et al., 2021*). Thus, deletion of *TOM70* may affect *aac2*[A128P] cells by two mechanisms: first, by further reducing carrier protein import and subsequently by impairing cytosolic proteostasis in the context of mitochondrial preprotein overaccumulation in the cytosol (*Wang and Chen, 2015*).

Next, we wondered whether the cellular response to acute Aac2p[A128P] expression could be indicative of protein import stress. To test this, we surveyed expression levels of a panel of genes known to be upregulated by clogging the TOM complex. These experiments were performed in the BY4742 strain background, in which expression of *aac2*[A128P] is incompatible with mtDNA loss (*Figure 4—figure supplement 3A*), thus minimizing any transcriptional effects secondary to mtDNA loss. We found that *CIS1* transcript levels are acutely increased upon expression of *aac2*[A128P], but not of *AAC2*, from a *GAL10* promoter (*Figure 4—figure supplement 3B*). *CIS1* upregulation is the hallmark of the mitochondrial compromised protein import response (mitoCPR) (*Weidberg and Amon, 2018*). We also found that *RPN4*, *HSP82*, *SSA3*, and *SSA4* are transiently upregulated (*Figure 4—figure supplement 3C–F*), consistent with previously published gene activation patterns induced by a synthetic mitochondrial protein import clogger (*Boos et al., 2019*). Thus, transcriptional responses further suggest that Aac2p[A128P] clogs the protein import machinery.

Accumulating evidence suggests a functional crosstalk between the mitochondrial protein translocases and phospholipid biosynthesis/trafficking pathways (*Hoffmann and Becker, 2022*; *Garg et al., 2022*). Interestingly, we found that *aac2*[A128P,A137D] expression renders cells hypersensitive to defects in phospholipid homeostasis. We expressed *aac2*[A128P,A137D] in cells impaired in production of cardiolipin (CL) (*pel1Δ*) and phosphatidylethanolamine (PE) (*psd1Δ*). Expression of *aac2*[A128P,A137D] strongly inhibited the growth of cells in these genetic backgrounds (*Figure 4—figure supplement 3G–H*). The data are consistent with previous observations that CL and PE facilitate mitochondrial protein import (*Becker et al., 2013*; *Gebert et al., 2009*). Alternatively, mitochondrial protein import clogging may affect phospholipid homoeostasis, which synergizes with *pel1Δ* and *psd1Δ* to cause cell lethality.

## ANT1[A114P] and ANT1[A114P,A123D] clog mitochondrial protein import in human cells

We introduced equivalent mutations in human *SLC25A4* (encoding the ANT1) with a C-terminal HA-tag, and transiently expressed the mutant proteins in HeLa cells (see *Figure 1A*). Like in yeast, combining missense mutations in Ant1 dramatically reduced steady-state protein levels, suggesting impaired ANT1 biogenesis (*Figure 5A*). Most strikingly, the level of ANT1[A114P,A123D], equivalent to the yeast Aac2p[A128P,A137D], is only 2.2% of the wild-type. Three lines of evidence suggest that ANT1[A114P] and ANT1[A114P,A123D] obstruct protein import into mitochondria. First, immunoprecipitation of ANT1[A114P]-HA demonstrated that the mutant protein has increased interactions with components of both TOM and TIM22 complexes (*Figure 5B–C*), as observed with its yeast ortholog Aac2p[A128P]. A meaningful immunoprecipitation of Aac2p[A128P,A137D]-HA was precluded by the extremely low-level accumulation of the mutant protein. Second, protease protection assay demonstrated that 42% and 52% of total ANT1[A114P] and ANT1[A114P,A123D] proteins are exposed on the OMM, whereas the wild-type ANT1 is protected from proteinase K digestion (*Figure 5D–E*). This is consistent with in vitro import studies of the Aac2p variants in yeast mitochondria (*Figure 2A–F*). Third, proteomics demonstrated that mitochondrial proteins accumulate in the cytosol of *SLC25A4*[p.A114P] and *SLC25A4*[p.A114P,A123D]-transfected cells, compared with wild-type Ant1-transfected cells (*Figure 5F*). Moreover, mitochondrial matrix proteins were significantly enriched in the cytosol of ANT1[A114P,A123D] versus ANT1-expressing cells (FDR<$3\times10^{-6}$) (*Figure 5G*; *Figure 5—source data 1*), corroborating that the double mutant Aac2p impairs global protein import. We therefore conclude that ANT1[A114P] and ANT1[A114P,A123D] clog protein import in human cells, with ANT1[A114P,A123D] having enhanced clogging activity despite low protein levels.

Regarding the function of the non-mitochondrial proteins increased in the cytosol of ANT1[A114P,A123D]-expressing cells, we found that the most significantly enriched Biological Process, Molecular Function, and KEGG pathway were 'chaperone', 'stress response', and 'protein processing in the ER' (*Figure 5H*). This may represent a stress response directed toward the increased mitochondrial protein burden in the cytosol.

An alternative explanation for reduced protein import in mutant Ant1-transfected cells is general mitochondrial damage and/or apoptosis activation leading to reduced $\Delta\psi$, which would also cause

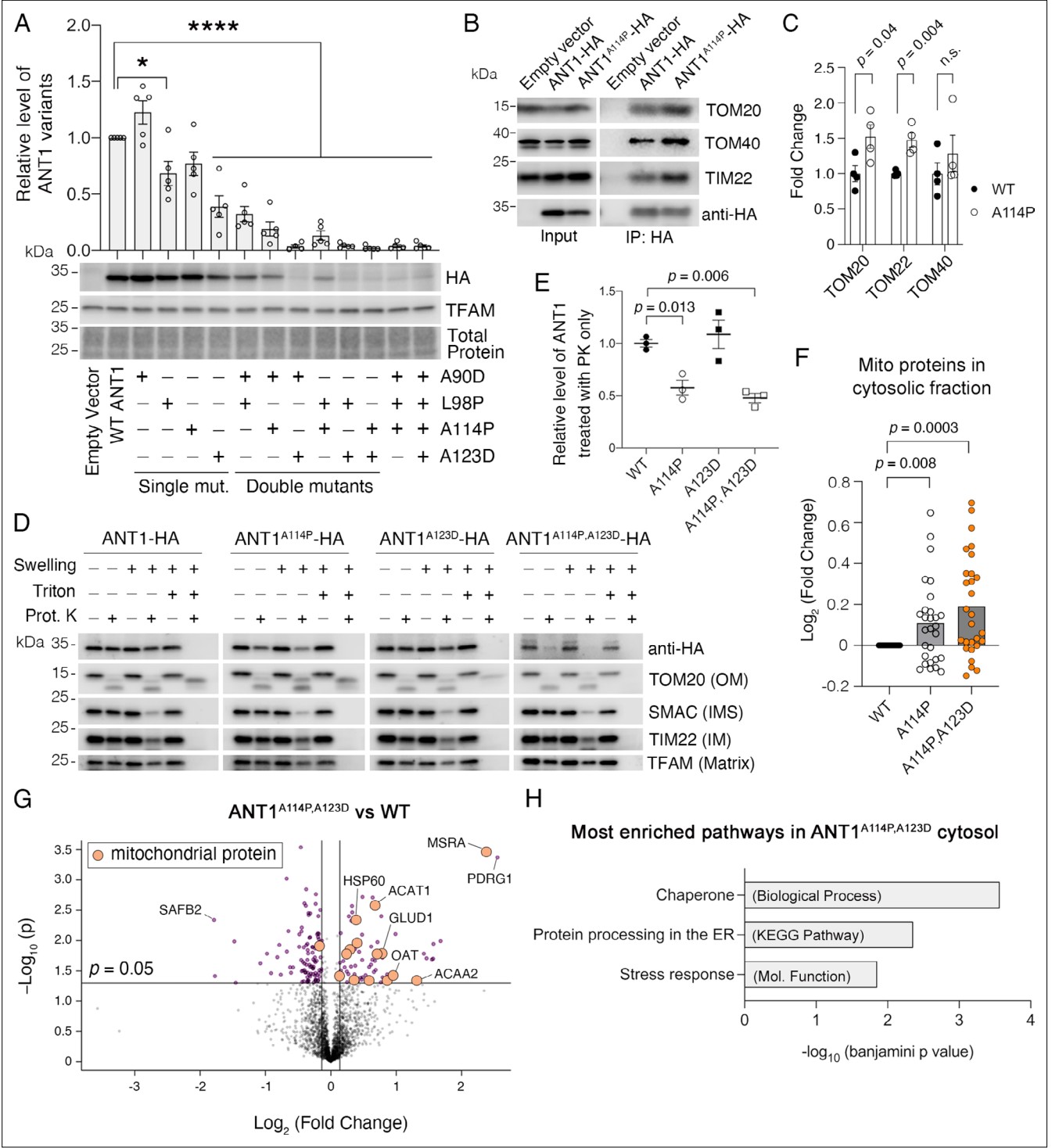

**Figure 5.** ANT1[A114P] and ANT1[A114P,A123D] clog mitochondrial protein import in human cells. (**A**) Combining pathogenic mutations in ADP/ATP translocase 1 (ANT1) strongly reduces protein levels, as indicated by immunoblot analysis of ANT1-hemagglutinin (HA) levels 24 hr after transfecting HeLa cells. ANT1 variant levels were normalized by TFAM, then plotted as relative to wild-type level. * indicates p<0.05, ****p<0.0001 from one-way ANOVA with Dunnett's multiple comparisons test. (**B**) Immunoprecipitation (IP) of ANT1-HA and ANT1[A114P]-HA from transiently transfected HeLa cells followed by immunoblot analysis, showing that ANT1[A114P] has increased interaction with the protein import machinery like its yeast ortholog Aac2p[A128P]. (**C**) Quantitation from four independent IP, one of which is depicted in (**B**). p-Values were calculated with a Student's t test. (**D**) Immunoblot analysis following protease protection assay showing that ANT1[A114P] and ANT1[A114P,A123D] are sensitive to proteinase K (PK) in isolated mitochondria. Swelling in hypotonic buffer was used to burst the outer membrane, and Triton X-100 was used to disrupt all membranes. OM, outer membrane; IMS, intermembrane space; IM, inner membrane. (**E**) Quantitation of the wild-type and mutant ANT1 pools that are protected from PK degradation

*Figure 5 continued on next page*

*Figure 5 continued*

in intact mitochondria. All HA levels were normalized by TFAM, then plotted as relative to its untreated sample. Replicates from three independent transfections. p-Values were calculated with a one-way ANOVA and Holm-Sidak's multiple comparisons test. (**F**) ANT1$^{A114P}$ and ANT1$^{A114P,A123D}$ obstruct general mitochondrial protein import. Proteomics of the cytosolic fraction of transfected HeLa cells reveals increase in mitochondrial proteins caused by ANT1$^{A114P}$-HA and ANT1$^{A114P,A123D}$-HA expression relative to ANT1-HA. p-Values were calculated with a Student's *t* test of the average abundance levels of each mitochondrial protein. (**G**) Volcano plot comparing the cytosolic proteome of *SLC25A4* $^{p.A114P,A123D}$ vs *SLC25A4*-transfected HeLa cells. Data represented as mean ± SEM. (**H**) Enrichment analysis of proteins significantly increased in the cytosol of *SLC25A4* $^{p.A114P,A123D}$-transfected HeLa cells. Depicted are the most significant enriched protein groups generated from three different databases: GO: Biological Process (top), KEGG pathway (middle), and GO: Molecular Function (bottom).

The online version of this article includes the following source data and figure supplement(s) for figure 5:

**Source data 1.** Proteomic comparison of the cytosolic fraction of SLC25A4 $^{p.A114P,A123D}$- versus *SLC25A4*-transfected HeLa cells.

**Source data 2.** Uncropped Western blots from *Figure 5A,B,D*.

**Figure supplement 1.** Mutant ADP/ATP translocase 1 (ANT1) does not reduce Δ$\phi$ or increase apoptosis in HeLa cells.

a reduction in protein import. However, neither ANT1$^{A114P,A123D}$ nor its single mutant counterparts reduced Δ$\phi$ or significantly increased cell death compared with wild-type (*Figure 5—figure supplement 1*). Thus, the effect on protein import by ANT1$^{A114P}$ and ANT1$^{A114P,A123D}$ is likely due to their physical retention in the import pathway rather than due to reduction of Δ$\phi$. It is interesting that ANT1$^{A114P,A123D}$ does not increase apoptosis in the immortalized HeLa cells, despite clearly clogging protein import. This may be related to the transient nature of ANT1 expression or an inherent resistance of the immortalized cells to cell death.

## Ant1$^{A114P,A123D}$ causes dominant muscle and neurological disease in mice

We generated a knock-in *Slc25a4* $^{p.A114P,A123D}$/+ (expressing Ant1$^{A114P,A123D}$) mouse line to model protein import clogging in vivo (*Figure 6A–B*). We observed a neurodegeneration phenotype that culminates in paralysis in some *Slc25a4* $^{p.A114P,A123D}$/+ mice (*Figure 6C*; *Video 1*; *Video 2*). This phenotype occurred in only four *Slc25a4* $^{p.A114P,A123D}$/+ mice, with a penetrance of 3.4% among the heterozygous mice that have reached 15 months of age. The presenting symptom is typically altered gait after the age of 11 months, followed by weight loss and death within 2–3 weeks of symptom onset. Histological analysis of the lumbar spinal cord in a symptomatic *Slc25a4* $^{p.A114P,A123D}$/+ mouse demonstrated dissolution of Nissl substance in the cell bodies of ventral horn neurons (*Figure 6D*), consistent with motor neuron degeneration (*Bodian and Mellors, 1945*). Neurodegeneration in the spinal cord was also indicated by GFAP accumulation by immunofluorescence and immunoblotting (*Figure 6E–G*). Transmission electron microscopy of ventral horn neurons revealed loss of cristae density of mitochondria, which suggests defects in mitochondrial biogenesis (*Figure 6H*).

Dominant *SLC25A4*-induced diseases primarily affect skeletal muscle in addition to low-penetrant neurological involvement (*Kaukonen et al., 2000*; *Siciliano et al., 2003*; *Simoncini et al., 2017*; *Kaukonen et al., 1999*; *Napoli et al., 2001*; *Deschauer et al., 2005*). Key muscle features include mildly reduced mitochondrial respiratory function, COX-deficient muscle fibers, and muscle weakness. Consistent with clinical phenotypes, we found that the maximal respiratory rate (state 3) of *Slc25a4* $^{p.A114P,A123D}$/+ muscle mitochondria was reduced by ~20% and ~30% when utilizing complex I in 9- and 24-month-old mice, respectively (*Figure 7A*). The respiratory control ratio, which is the single most useful and sensitive general measure of energy coupling efficiency (*Brand and Nicholls, 2011*), was reduced by ~31% and ~42% in young and old mice, respectively. Surprisingly, when complex I is inhibited and complex II substrate (succinate) is present, maximal respiration is increased by ~23% in the mutant mice at 9 months of age (*Figure 7B*). This result is important because it confirms that ATP/ADP transport is not a limiting factor for sustaining a high respiratory rate in *Slc25a4* $^{p.A114P,A123D}$/+ mice.

In addition to mild bioenergetic defect, we detected significant reduction in myofiber diameter in the skeletal muscle from aged *Slc25a4* $^{p.A114P,A123D}$/+ mice (*Figure 7C–D*). Sequential COX/SDH histochemical assay failed to detect any COX-negative/SDH-positive fibers in the aged *Slc25a4* $^{p.A114P,A123D}$/+ muscles. Instead, we found that ~3.5% of myofibers have reduced COX and SDH activity centrally, but remained COX and SDH-positive in the periphery (*Figure 7E–G*). Finally, we found that the *Slc25a4* $^{p.A114P,A123D}$/+ mice display muscle weakness (*Figure 7H–I*). These data demonstrate that *Slc25a4* $^{p.A114P,A123D}$ induces a dominant myopathic phenotype.

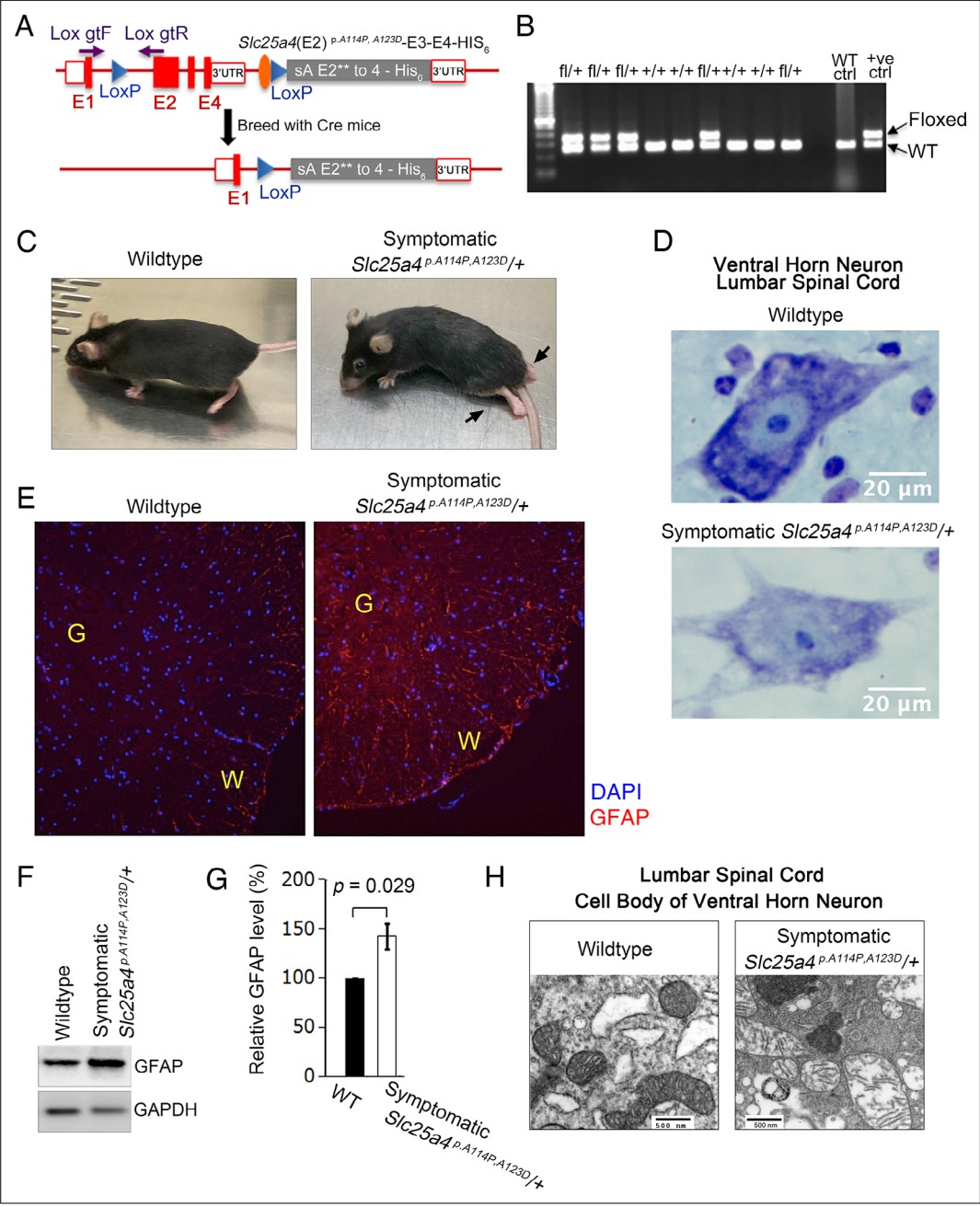

**Figure 6.** *Slc25a4 p.A114P,A123D/+* mouse generation and neurodegeneration. (**A**) Schematic of the strategy by which the knock-in *Slc25a4 p.A114P,A123D/+* knock-in mice were generated. E1–E4, exons 1–4 of *Slc25a4*. In gray is the inserted cDNA containing two missense mutations in exon 2, followed by the endogenous 3' UTR. Lox gtF and gtR indicate genotyping primers. (**B**) Agarose gel electrophoresis of PCR genotyping using genotyping primers indicated in (**A**). Fl, floxed. (**C**) Ascending paralytic phenotype of an *Slc25a4 p.A114P,A123D/+* mouse at 11 months of age, and its wild-type littermate. Arrows point to paralyzed hindlimbs. (**D**) Nissl-stained lumbar spinal cord neuron of a symptomatic *Slc25a4 p.A114P,A123D/+* mouse and wild-type littermate. This neuron shows loss of Nissl substance and blurring of nuclear boundaries, process known as 'chromatolysis', which indicates neuron degeneration. (**E**) Indirect immunofluorescence for the astrocyte marker glial fibrillary acidic protein (GFAP) indicating spinal cord gliosis in a symptomatic *Slc25a4 p.A114P,A123D/+* mouse. G, gray matter; W, white matter. (**F**) Immunoblot analysis of lumbar spinal cord lysate confirmed increase in GFAP in symptomatic *Slc25a4 p.A114P,A123D/+* mice. (**G**) Quantitation from (**F**) showing significant increase in GFAP in the spinal cord of a symptomatic *Slc25a4 p.A114P,A123D/+* mouse indicating neuroinflammation. p-Value was calculated from Student's *t* test. (**H**) Transmission electron microscopy of a ventral horn neuron of a symptomatic *Slc25a4 p.A114P,A123D/+* mouse and wild-type littermate control.

*Figure 6 continued on next page*

*Figure 6 continued*

The online version of this article includes the following source data for figure 6:

**Source data 1.** Uncropped Western blots from *Figure 6F*.

## Ant1<sup>A114P,A123D</sup> clogs protein import in vivo

For biochemical characterization of the mutant protein, we turned to homozygous *Slc25a4* *p.A114P,A123D*/*Slc25a4* *p.A114P,A123D* mice as to eliminate the detection of confounding wild-type Ant1. We found that the Ant1[A114P,A123D] protein is virtually undetectable in the total lysates of mouse tissues, suggesting a rapid degradation of the mutant protein as observed in yeast and human cells. (*Figure 8A*). Only in isolated skeletal muscle mitochondria were we able to detect Ant1[A114P,A123D], which was present at ~0.1% of wild-type level (*Figure 8B–C*). Protease protection assay on isolated muscle mitochondria demonstrated that Ant1[A114P,A123D] is more sensitive to proteinase K digestion compared with wild-type Ant1 (*Figure 8D–E*), consistent with import clogging in vivo. Moreover, Smac also appears to be more sensitive to proteinase K in *Slc25a4* *p.A114P,A123D*/*Slc25a4* *p.A114P,A123D* mitochondria, while Mdh2 and Tim23 do not (*Figure 8F–H*). While this could indicate instability or partial rupture of the OMM in mutant mitochondria, it could also be a result of clogging and imply that clogging does not affect all mitochondrial preproteins equally. Consistent with this concept, our experiments in yeast showed minimal effects of clogging on Ilv5p, a mitochondrial matrix protein, in contrast to Hsp60p (*Figure 1F*; *Figure 2H–I*). This could be explained by the observation that the Tom40 β-barrel pore contains several distinct protein paths (*Shiota et al., 2015*; *Araiso et al., 2019*).

Finally, to test if Ant1[A114P,A123D] obstructs general mitochondrial protein import in vivo, we compared the cytosolic proteomes of aged muscle from wild-type and *Slc25a4* *p.A114P,A123D*/+ mice using tandem mass tagged (TMT) quantitative proteomics (*Figure 8I*). We found a striking global increase in mitochondrial proteins in the cytosol of *Slc25a4* *p.A114P,A123D*/+ muscle (*Figure 8J*). Among the 75 proteins increased by at least 25% in the cytosol (p<0.05), proteins assigned to the mitochondrion accounted for 45 of them (*Figure 8—source data 1*). This enrichment was highly significant (FDR<10<sup>−33</sup>). Taken together, the data suggest that Ant1[A114P,A123D] clogs general protein import in vivo. Supporting this is a trend of increase in proteasomal subunits and Hsp70 proteins in the cytosol of *Slc25a4* *p.A114P,A123D*/+muscle, which suggests an anti-mPOS response (*Figure 8—figure supplement 1A–B*).

To evaluate the impact of *Slc25a4* *p.A114P,A123D* expression on mitochondrial proteostasis, we assessed the assembly state of protein complexes and supercomplexes in the inner membrane with

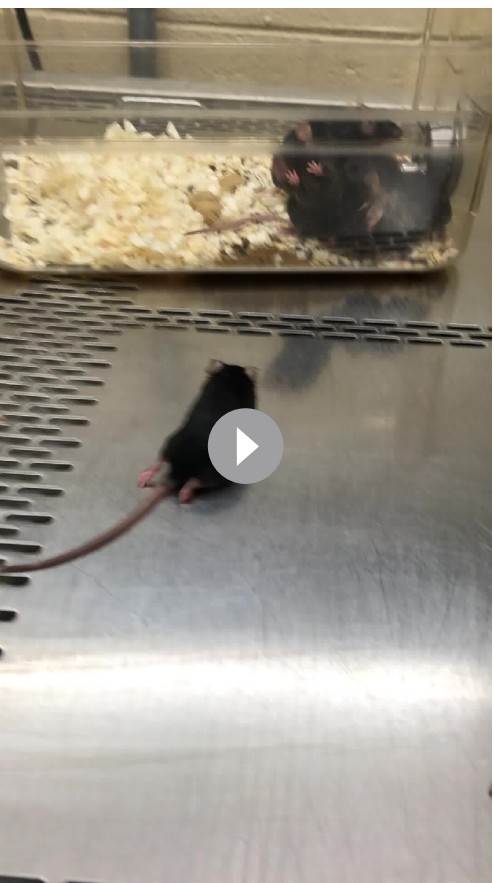

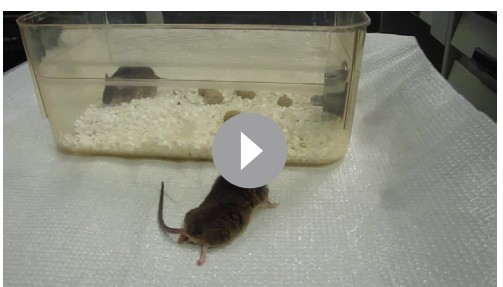

**Video 1.** Paralytic phenotype of an *Slc25a4* *p.A114P,A123D*/+ mouse at the age of 12 months.

https://elifesciences.org/articles/84330/figures#video1

**Video 2.** Paralytic phenotype of an Slc25a4 *p.A114P,A123D*/+ mouse at the age of 16 months.

https://elifesciences.org/articles/84330/figures#video2

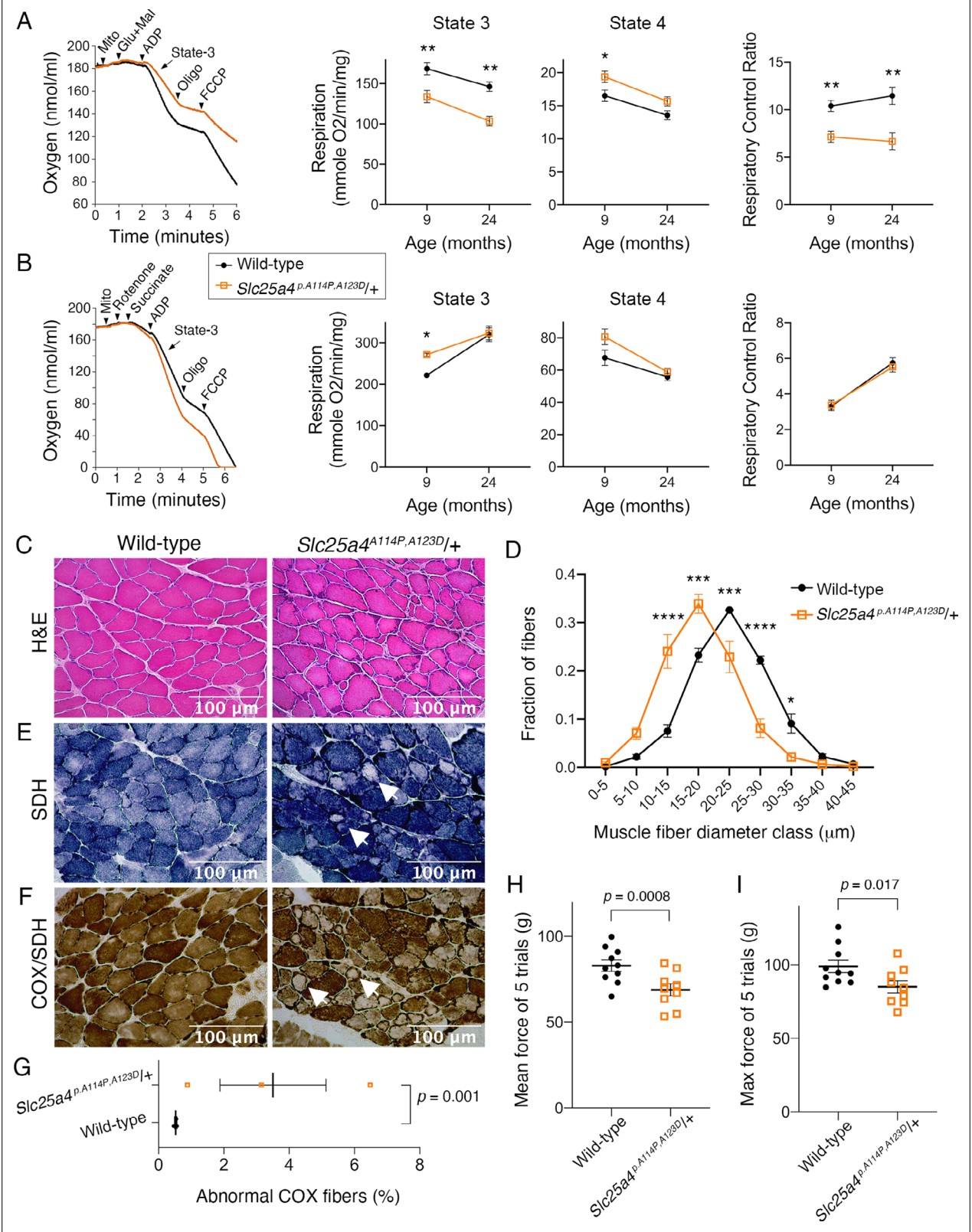

**Figure 7.** Ant1[A114P,A123D] (encoded by *Slc25a4 p.A114P,A123D*) causes a dominant mitochondrial myopathy in mice. (**A**) Respirometry of isolated skeletal muscle mitochondria with complex I stimulated by glutamate (glu) and malate (mal). State 3, maximal respiratory rate after addition of ADP; state 4, oligomycin (oligo)-inhibited respiratory rate; respiratory control ratio = state 3/state 4. N=6 mice/genotype at 9 months of age; n=4 mice per genotype at 24 months of age. Three measurements were taken per mouse. p-Values were derived from repeated measures ANOVA with measurement order

*Figure 7 continued on next page*

*Figure 7 continued*
as the within-subjects variable. Data from two age groups were analyzed independently. FCCP, trifluoromethoxy carbonylcyanide phenylhydrazone. (**B**) Respirometry of isolated skeletal muscle mitochondria with complex II stimulated by succinate and complex I inhibited by rotenone. N=2 mice/genotype at 9 months of age, 4 measurements/mouse; n=4 mice/genotype at 24 months of age, 3 measurements/mouse. Data analyzed as in (**A**). (**C**) Soleus muscles stained with hematoxylin and eosin (H&E) showing smaller myofibers in 30-month-old *Slc25a4* $^{p.A114P,A123D}$/+ mice. (**D**) Feret's diameter analysis of H&E stained soleus in (**C**) reveals atrophy in *Slc25a4* $^{p.A114P,A123D}$/+ mice. At least 340 myofibers were measured per soleus. Myofiber diameters were binned into 5 μm ranges and plotted as % of total. N=3 mice/genotype. Data analyzed by two-way ANOVA with Sidak's multiple comparisons test. (**E**) Succinate dehydrogenase (SDH) histochemical activity staining of the soleus showing abnormal fibers that stain for SDH peripherally but are pale internally (arrows). (**F**) Histochemical cytochrome *c* oxidase (COX) and SDH sequential staining of the soleus shows abnormal fibers that stain for COX peripherally, but do not stain for COX or SDH internally. (**G**) Quantitation of abnormal COX fibers shown in (**F**). p-Value was calculated from Student's *t* test. (**H**) Forelimb grip strength is reduced in 30-month-old *Slc25a4* $^{p.A114P,A123D}$/+ mice. p-Value from Student's *t* test. (**I**) Maximal forelimb grip strength is reduced in 30-month-old *Slc25a4* $^{p.A114P,A123D}$/+ mice. p-Value from Student's *t* test. Data represented as mean ± SEM.

native gel electrophoresis. The respiratory complexes and supercomplexes were minimally affected (*Figure 8—figure supplement 1C–D*), as was the TIM23 complex (*Figure 8—figure supplement 1E*). To gain a global view of mitochondrial proteostasis, we performed TMT labeling quantitative proteomics on mitochondrial fractions from skeletal muscle. We found minimal changes in the steady-state levels of mitochondrial proteins, including other Ant isoforms (*Figure 8—figure supplement 1F*, *Figure 8—figure supplement 1—source data 1*). This may reflect highly efficient degradation of Ant1$^{A114P,A123D}$, which is suggested by its steady-state level being 0.1% of wild-type (*Figure 8B–C*). It is also consistent with the subtle bioenergetic defects and the overall mild neuronal and muscular phenotypes associated with the *Slc25a4* $^{p.A114P,A123D}$/+ mice. Interestingly, a focused analysis on mitochondrial proteases and chaperones suggested a subtle but statistically significant increase in the mitochondrial proteostatic machinery (p=2.7 × 10$^{-14}$, analyzed with a two-way ANOVA; *Figure 8—figure supplement 1G*), which supports proteostatic stress inside mitochondria.

Another possible explanation for minimal changes in mitochondrial proteins in the context of protein import clogging could be increased protein import TIM22 pathway. Our mitochondrial proteomics dataset suggested global increase in TIM22 pathway components (*Figure 8—figure supplement 1H*). Immunoblot validation of Tim22 showed an ~40% increase (*Figure 8—figure supplement 1I–J*). These data may represent chronic adaptation to carrier protein import clogging, though more work is required to rigorously test this hypothesis.

## Transcriptional response induced by *Slc25a4* $^{p.A114P,A123D}$ expression in mouse muscle

The integrated stress response (ISR) is activated by a diverse range of mitochondrial insults. Surprisingly, we found no evidence of ISR activation downstream of *Slc25a4* $^{p.A114P,A123D}$ expression. First, eIF2α phosphorylation, a hallmark of the ISR, was not increased in *Slc25a4* $^{p.A114P,A123D}$/+ mouse skeletal muscle at multiple ages (*Figure 8—figure supplement 2A–B*). Second, transcriptomic analysis in *Slc25a4* $^{p.A114P,A123D}$/+ skeletal muscle failed to show induction of ISR target genes (*Figure 8—figure supplement 2C*). Instead, transcription factor enrichment analysis of significantly upregulated genes (q<0.05) revealed activation of pathways controlled by novel transcriptional factors, including FOXO1 and FOXO3 that are involved in metabolic homeostasis and autophagy (*Figure 8—figure supplement 2D–E*). It appeared that *Slc25a4* $^{p.A114P,A123D}$ induces an entirely unique transcriptional signature (*Figure 8—figure supplement 2F*; *Figure 8—figure supplement 2—source data 1*), although it remains to be determined whether severe mitochondrial protein import clogging relative to Ant1$^{A114P,A123D}$ induces similar stress responses and/or activates ISR. Among the upregulated genes, *Depp1* is known to activate autophagy (*Stepp et al., 2014*). More importantly, when directly compared to *Slc25a4* knockout mice (*Morrow et al., 2017*), there is very limited overlap in the enrichment profile among significantly altered genes (*Figure 8—figure supplement 2G*). This indicates that the primary stress in *Slc25a4* $^{p.A114P,A123D}$/+ skeletal muscle is distinct from that in *Ant1* knockout mice.

In summary, we found that Ant1$^{A114P,A123D}$ dominantly causes muscle and low-penetrant neurological disease in mice that recapitulates pathological and molecular phenotypes of dominant Ant1-induced diseases in humans. This corroborates the toxicity of Aac2$^{A128P,A137D}$ in yeast (*Figure 1*), and supports the idea that mitochondrial protein import clogging by a mutant substrate preprotein is pathogenic.

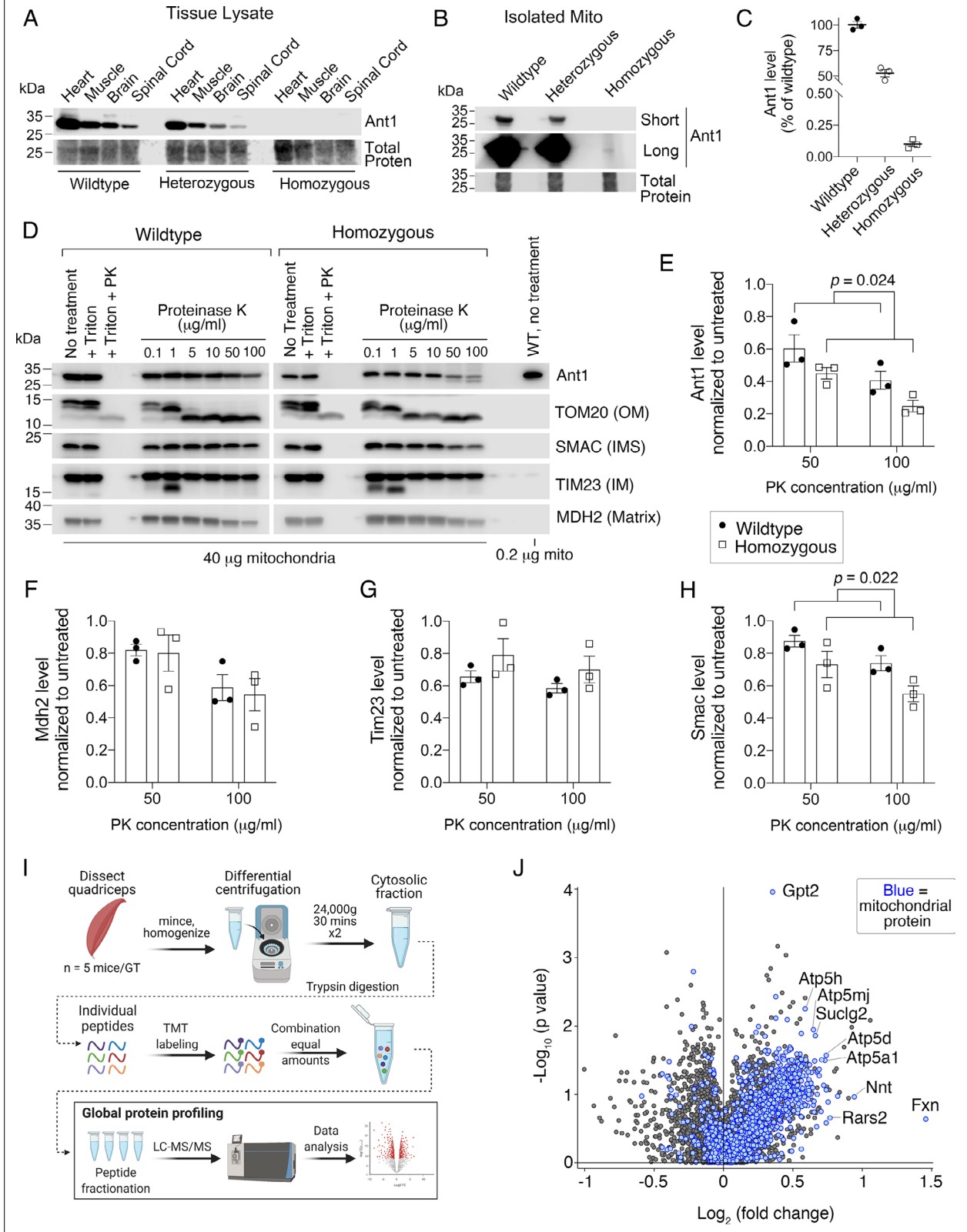

**Figure 8.** Ant1[A114P,A123D] clogs protein import in vivo. (**A**) Immunoblot analysis of tissue lysate showing low Ant1[A114P,A123D] protein levels in heterozygous and homozygous mice. (**B**) Immunoblot analysis of isolated muscle mitochondria demonstrating low Ant1[A114P,A123D] protein levels. (**C**) Quantitation of Ant1 levels in isolated muscle mitochondria from three mice per genotype, as determined by immunoblotting. Values were normalized to total protein stain and shown as relative to wild-type. (**D**) Ant1[A114P,A123D] is more sensitive to proteinase K (PK) than wild-type Ant1 in intact mitochondria. Immunoblot

*Figure 8 continued on next page*

*Figure 8 continued*

analysis after PK protection assay of isolated muscle mitochondria in isotonic buffer. Ant1[A114P,A123D] was detected using SuperSignal West Femto Maximum Sensitivity Substrate (top right panel). (**E–H**) Quantitation from protease protection assay, as shown in (**D**). n=3 mice per genotype. p-Values were calculated with a two-way ANOVA, showing significant main effect of genotype. Data represented as mean ± SEM. (**I**) Schematic of tandem mass tagged (TMT) quantitative proteomic analysis. (**J**) Volcano plot comparing the cytosolic proteome of *Slc25a4 p.A114P,A123D*/+ vs wild-type skeletal muscle, with mitochondrial proteins highlighted in blue.

The online version of this article includes the following source data and figure supplement(s) for figure 8:

**Source data 1.** Proteomic comparison of the cytosolic fraction of 30-month-old skeletal muscle from Slc25a4 *p.A114P,A123D*/+ versus wild-type mice.

**Source data 2.**

**Figure supplement 1.** Impact of Ant1[A114P,A123D] expression on mitochondrial and cytosolic proteostasis.

**Figure supplement 1—source data 1.** Proteomic comparison of the crude mitochondrial fraction of 30-month-old skeletal muscle from Slc25a4 *p.A114P,A123D*/+ versus wild-type mice.

**Figure supplement 1—source data 2.** Uncropped photos of Coommassie-stained BN-PAGE gel (*Figure 8C*) and uncropped Western blots (*Figure 8D,E,I*).

**Figure supplement 2.** A unique transcriptional response induced by Ant1[A114P,A123D] in mouse muscle.

**Figure supplement 2—source data 1.** Transcriptomic comparison of skeletal muscle from 9-month-old Slc25a4 *p.A114P,A123D*/+ versus wild-type mice.

**Figure supplement 2—source data 2.** Uncropped Western blots from *Figure 8—figure supplement 2A*.

## Discussion

Failure in mitochondrial protein import has severe physiological consequences. In addition to defective mitochondrial biogenesis and energy metabolism, it also causes toxic accumulation and aggregation of mitochondrial preproteins in the cytosol, a process termed mitochondrial Precursor Overaccumulation Stress (mPOS) (*Wang and Chen, 2015*; *Coyne and Chen, 2018*; *Song et al., 2021*). To prevent these consequences, many cellular safeguards have been identified that can maintain protein import efficiency and/or mitigate mPOS (*Backes et al., 2021*; *Weidberg and Amon, 2018*; *Boos et al., 2019*; *Izawa et al., 2012*; *Wrobel et al., 2015*; *Nargund et al., 2015*; *Izawa et al., 2017*; *Hansen et al., 2018*; *Zurita Rendón et al., 2018*; *Liu et al., 2019*; *Mårtensson et al., 2019*; *Su et al., 2019*; *Ordureau et al., 2020*; *Phu et al., 2020*; *Xiao et al., 2021*; *Shakya et al., 2021*; *Xin et al., 2022*; *Sam et al., 2021*; *Schulte et al., 2023*; *Dewar et al., 2022*; *Krämer et al., 2023*). So far, studies on mitochondrial stressors that impair protein import have been mainly focused on mutations that directly affect the core protein import machinery, synthetic clogger proteins, or on pharmacological interventions that reduce $\Delta\phi$ (*Song et al., 2021*). In this report, we have uncovered the first example of naturally occurring missense mutations in an endogenous mitochondrial protein causing toxic protein import clogging. We showed that a single protein without overexpression is sufficient to clog import, thereby inducing robust protein import stress responses, bioenergetic defects, and cytosolic proteostatic stress. The data supports a model in which import clogging kills yeast cells and contributes to muscle and neurological degeneration in mice.

### Mitochondrial protein import can be clogged by a mutant mitochondrial preprotein

The mitochondrial carrier protein family is the largest of the transporter families and has highly conserved domain and sequence features across all eukaryotes. Their import into mitochondria is unique in that they assume partially folded conformations called 'hairpin loops' that place adjacent hydrophobic α-helices antiparallel with one another (*de Marcos-Lousa et al., 2006*; *Wiedemann et al., 2001*). There are three 'hairpin loop' modules per carrier preprotein that transit through the Tom40 β-barrel in a loop-first topology, such that the C- and N-termini initially remain exposed to the cytosol. This leaves little wiggle room for the carrier preprotein considering the Tom40 β-barrel can only accommodate the width of up to two α-helical segments at a time (*Wiedemann et al., 2001*). To get through the Tom40 β-barrel pore, the hydrophobic hairpin loops are thought to be guided by an array of hydrophobic patches on the inside of the pore (*Shiota et al., 2015*; *Araiso et al., 2019*). It is in the hydrophobic α-helices of the ADP/ATP translocase ANT1 (Aac2p in yeast) that dominant pathogenic mutations are found, which introduce either a proline or an aspartic acid (see *Figure 1*). Here, we show that these mutations cause the protein to arrest at or in the TOM complex, thereby partially

obstructing general protein import. There are multiple potential mechanisms. First, as the mutations introduce a proline or aspartic acid into the hydrophobic α-helix, it may simply be loss of the necessary hydrophobic interaction with the inner surface of the Tom40 β-barrel pore. Another possibility is that the hairpin loop structure is disrupted such that it becomes too wide to fit through the narrow pore. This may be particularly relevant for the proline mutants which would introduce a kink into the α-helix. A third possibility is that mutant ANT1/Aac2p is not impaired in transit through the pore, but instead is inefficiently chaperoned by the small TIM chaperones in the IMS, leading to 'backpressure' in the pathway. Interestingly, carrier preproteins appear to form partial α-helical secondary structures while bound to the Tim9-Tim10 chaperone (*Weinhäupl et al., 2018*). Future work will be essential to rigorously decipher between these potential mechanisms of clogging.

The data strongly suggest that double mutant Aac2p preferentially clogs the TOM complex (*Figure 2*). In contrast, co-immunoprecipitation of single mutants Aac2p$^{A128P}$ and ANT1$^{A114P}$ showed increased association with both the TOM and TIM22 complexes. We interpret these data with caution, as it would be unexpected that single mutants would clog at multiple locations. However, it is possible that hairpin loops and/or hydrophobic α-helices are crucial for interaction with both complexes. As the precise mechanism of import and IMM insertion by the TIM22 complex remains unresolved, it would be premature to speculate on potential clogging mechanisms.

One of the most surprising observations of this study was the extreme toxicity of the double mutant clogger proteins that arrest at the TOM complex. The deleterious stressors downstream of clogging are likely to be multifactorial and include respiratory defects and cytosolic protestatic stress via mPOS. While it is possible that general proteostasis is perturbed by Aac2p$^{A128P,A137D}$ saturating proteolytic machinery, our data suggest this is unlikely. The IMM-associated AAA protease Yme1p (and not the proteasome) appears to be a major proteolytic pathway for clogged Aac2p$^{A128P,A137D}$. That genetic deletion of *YME1* is well tolerated in yeast cells suggests that loss of Yme1-based proteolysis does not underlie toxicity of *aac2$^{A128P,A137D}$* expression. We propose that clogging of the TOM complex is the primary mechanism of Aac2p$^{A128P,A137D}$-induced protein import defects and the ensuing cell stress. Importantly, the former is strongly supported by in vitro experiments showing that Aac2p$^{A128P,A137D}$ is defective in translocating through the TOM complex in fully energized, wild-type mitochondria.

In the context of low Aac2p$^{A128P,A137D}$ levels in cells (just 4.7% of wild-type), it is important to note that the ADP/ATP carrier is one of the most abundant proteins in mitochondria, outnumbering the Tom40 pore-forming protein by almost an order of magnitude (*Morgenstern et al., 2017*; *Pfanner et al., 2019*). Thus, if ~70% of the reduced level of Aac2p$^{A128P,A137D}$ is actively clogging TOM complexes, this would still occupy ~30% of the Tom40 channels, which may have a considerable effect on general protein import. This reflects the central importance of proper TOM complex function to mitochondrial and cell homeostasis, and also provides physiological justification for the existence of multiple pathways dedicated to unclogging the TOM complex (*Weidberg and Amon, 2018*; *Mårtensson et al., 2019*).

It will be interesting to determine whether and how Yme1 plays a general role in quality control of the TIM22 import pathway. Yme1 degradation of Aac2p$^{A128P,A137D}$ could occur at the TOM complex, in association with the small TIMs or at the TIM22 complex. A functional crosstalk between Yme1p and the TIM22 complex has been proposed (*Hwang et al., 2007*; *Kumar et al., 2023*). Detailed mechanistic studies are required.

## Mitochondrial protein import clogging as a mechanism of disease

A long-standing mystery in the mitochondrial disease field is how dominant mutations in *SLC25A4* (encoding adenine nucleotide translocase 1 [Ant1]) cause such a wide spectrum of clinical and molecular phenotypes that are not observed in patients with homozygous null *SLC25A4* (*Kaukonen et al., 2000*; *Siciliano et al., 2003*; *Simoncini et al., 2017*; *Palmieri et al., 2005*; *Napoli et al., 2001*; *Deschauer et al., 2005*; *Thompson et al., 2016*; *Echaniz-Laguna et al., 2012*; *Tosserams et al., 2018*; *Kashiki et al., 2022*). The implication is that the dominant pathogenic ANT1 proteins must have an unknown gain-of-function pathogenic mechanism. Similarly, the phenotypes observed in *Slc25a4* $^{p.A114P,A123D}$/+ mice cannot be explained by haploinsufficiency or a dominant negative effect on ADP/ATP transport for the following reasons. First, as in humans, the neurological and muscle phenotypes in *Slc25a4* $^{p.A114P,A123D}$/+ mice have never been reported in heterozygous *Slc25a4* knockout mice despite 25 years of characterization by Doug Wallace and colleagues. Second, mitochondria

from *Slc25a4* $^{p.A114P,A123D}$/+ mouse muscle are able to exceed the maximal respiration rate of wild-type mitochondria when complex II is stimulated (*Figure 6B*), indicating that one functional copy of Ant1 is sufficient to support high respiratory rates without compensatory increase in other Ant isoforms (see *Figure 8—figure supplement 1F*; *Figure 8—figure supplement 1—source data 1*; *Figure 8—figure supplement 2—source data 1*; *Figure 8—figure supplement 2F*). This is in stark contrast to *Slc25a4* knockout, which displays a >30% reduction in complex II-based respiration (*Graham et al., 1997*). Third, the transcriptional profile in *Slc25a4* $^{p.A114P,A123D}$/+ skeletal muscle is entirely distinct from that in *Slc25a4* knockout mice, suggesting different underlying mechanisms of muscle dysfunction (*Morrow et al., 2017*). Fourth, Ant1 likely functions as a monomer, making a dominant-negative mechanism of adenine nucleotide transport unlikely (*Pebay-Peyroula et al., 2003*; *Ruprecht et al., 2019*; *Kunji and Ruprecht, 2020*). Finally, the yeast equivalent of Ant1$^{A114D,A123D}$, Aac2p$^{A128P,A137D}$, is catalytically inactive yet clearly exerts toxicity even in conditions where *aac2Δ* cells are healthy (*Figure 1C*). Taken together, the evidence is clear that dominant *SLC25A4* mutations cause gain-of-function phenotypes in patients, *Slc25a4* $^{p.A114P,A123D}$/+ mice and yeast models.

In this study, we elucidated a candidate molecular mechanism of dominant Ant1-induced disease using yeast, human cells, and a novel mouse model. We found that pathogenic ANT1 mutations modeled in yeast's Aac2p cause the protein to clog global mitochondrial protein import leading to respiratory defects, cytosolic proteostatic stress (*Wang and Chen, 2015*), and loss of cell viability. Combining two mutations into a single Ant1 protein drastically enhanced clogging activity, which tightly correlated with toxicity. We extensively characterized one such 'super-clogger' variant, Aac2p$^{A-128P,A137D}$, and found that the main clogging site shifted to the TOM complex, providing an explanation for the increased toxicity. Protein import clogging was strikingly consistent in transfected human cells, as ANT1$^{A114P}$ and ANT1$^{A114P,A123D}$ (ortholog of yeast Aac2p$^{A128P,A137D}$) physically associated with the TOM and TIM22 complexes, was exposed on the outside of intact mitochondria, and caused mitochondrial protein retention in the cytosol. Even in mice, Ant1$^{A114P,A123D}$ expression correlated with retention of mitochondrial proteins in the cytosol despite minute levels of the mutant protein, presumably due to degradation. Finally, we found that *Slc25a4* $^{p.A114P,A123D}$/+ mice phenocopy ANT1-induced autosomal dominant progressive external opthalmoplegia (adPEO) patients, as *Slc25a4* $^{p.A114P,A123D}$/+ mice exhibit mild OXPHOS defects, partially COX-deficient myofibers, moderate myofiber atrophy, and muscle weakness (*Kaukonen et al., 1999*; *Komaki et al., 2002*). The low-penetrant neurodegeneration in *Slc25a4* $^{p.A114P,A123D}$/+ mice may also reflect human disease process, as a neurodegenerative phenotype has been reported in an adPEO patient carrying the *SLC25A4* $^{p.A114P}$ allele (*Simoncini et al., 2017*). Taken together, these data strongly argue that protein import clogging contributes to adPEO in humans, despite *Slc25a4* $^{p.A114P,A123D}$ not directly genocopying a particular clinical condition. We do note, however, that additional contributory mechanisms and even *SLC25A4* deficiency cannot be completely excluded without in vivo rescue experiments.

In summary, we demonstrate that pathogenic mutations in a mitochondrial carrier protein can cause clogging of the protein import pathway, which induces multifactorial cellular stress and correlates with cell toxicity and disease in mice. These findings uncovered a vulnerability of the mitochondrial protein import machinery to single amino acid substitutions in mitochondrial inner membrane preproteins. There are 53 carrier proteins in human mitochondria that have highly conserved domain organization both within and across species (*Palmieri et al., 2011*). Mutations altering conformational elements involved in protein import, such as hairpin loops, are likely strongly selected against during evolution. Those persisting in the human population would affect protein import and cause disease, as exemplified by the dominant pathogenic mutations in *SLC25A4*.

Our work provides just a first glimpse into the amino acid requirements of mitochondrial carrier proteins to maintain compatibility with the mitochondrial protein import machinery. Future work will be crucial to systematically evaluate the amino acid requirements first within ANT1/Aac2p, and then across entire carrier protein family. It will also be important to determine which specific interactions with protein import components are affected by different mutations. Overall, these efforts may enable the prediction of pathogenic mechanisms of future variants of uncertain significance as they arise with more exome sequencing of human patients.

## Materials and methods
Key resources table - see Appendix 1.

## Yeast growth conditions and genetic manipulation

Yeast cells were grown using standard media. Genotypes and sources of yeast strains are listed in Key resources table. To generate the *AAC2* expression plasmids, we amplified the gene from total genomic DNA and inserted it into pFA6a-ScAAC2-URA3-HIS3/2. The missense mutations were introduced using QuikChange Site-Directed Mutagenesis (Stratagene) and mutations were confirmed by sequencing. Mutant genes were then cloned into pRS416 for expression (*Figure 1B*) and pFA6a-KanMX6 for placing next to the *KAN* selection marker. The *aac2^A128P,A137^-KAN* cassette was then amplified for integration into the *LYS2* locus of the W303-1B strain background. All strains in *Figure 1C–E* were derived from this strain by standard genetic crosses. All combinations of *aac2* mutations were expressed by transforming an equal number of M2915-6A yeast with freshly prepared pRS416-based vectors (*URA3*) expressing wild-type or mutant *aac2*. Equal fractions of the transformant culture were plated on selective medium lacking uracil and grown at 25°C or 30°C for 3 days before being photographed (, *Figure 1B and S1A* ). *tom70Δ* and *tim18Δ* strains were generated by amplification of the knockout cassette from genomic DNA of *tom70Δ* and *tim18Δ* strains from BY4741 knockout library, followed by transformation into M2915-6A by selecting for G418^R. The disruption of the genes was confirmed by PCR using an independent primer pair surrounding the native genomic locus. *PRE9* and *UMP1* were disruption in strains of W303 background by the insertion of *kan*. These alleles were then introduced into strains expressing *aac2^A128P, A137D^* by genetic crosses. For galactose-induced expression, *AAC2* and its mutant alleles were placed under the control of the *GAL10* promoter in the integrative vector pUC-URA3-GAL. The resulting plasmids were then linearized by cutting with *Stu*I within the *URA3* gene, before being integrated into the *ura3*-1 locus by selecting for Ura^+ transformants. Correct chromosomal integration was confirmed by examining the stability of the Ura^+ phenotype and PCR amplification of the *URA3* locus. The *yme1Δ* and *pdr5Δ* alleles were introduced into these strains either by direct gene disruption or by genetic crosses.

## Cell lysis, western blotting, and signal quantitation

Unless otherwise noted, yeast cells were lysed and prepared for SDS-PAGE as previously described (*Chen, 2001*). HeLa cells and mouse tissues were lysed with RIPA buffer containing 1× HALT Protease and Phosphatase Inhibitor Cocktail (Thermo Fisher) and prepared for SDS-PAGE with Laemmli buffer. Standard procedures were used for chemiluminescent western blot detection of proteins. Membranes were imaged using a LI-COR Odyssey imager and signals quantitated in the associated ImageStudio software.

## Detergent extraction of potential insoluble protein

We took two approaches to detecting potentially aggregated Aac2p^A128P,A137D. First, spheroplasts were generated using zymolyase followed by lysis in sample buffer (containing 8 M urea+5% SDS) with or without 2% Sarkosyl. Second, the insoluble material from our typical lysis procedure (see above) was treated with different detergents. For formic acid, pellet was resuspended in 100% formic acid, incubated at 37°C for 70 min, followed by speed-vac drying of the sample and resuspension in sample buffer for SDS-PAGE. For guanidine dissociation, pellet was resuspended in 8 M guanidine HCl, incubated at 25°C for 70 min, then sodium deoxycholate was added to a final concentration of 1%. Protein was then precipitated with TCA, washed with acetone at –20°C, and resuspended in sample buffer for SDS-PAGE.

## Affinity chromatography

His-tagged Aac2p affinity purification was performed on 1 mg mitochondria per replicate. Yeast mitochondria were isolated as previously described (*Diekert et al., 2001*). Mitochondria were lysed in 'lysis buffer' (10% glycerol, 1.5% digitonin, 50 mM potassium acetate, 2 mM phenylmethylsulfonyl fluoride [PMSF], and a protease inhibitor cocktail) including 100 mM NaCl ('low salt') or 200 mM NaCl ('high salt') and incubated on ice for 30 min. Lysate was centrifuged at 21,000×*g* for 30 min. Imidazole was added to supernatant for a final concentration of 4 mM, which was applied to preequilibrated Ni-NTA agarose beads (QIAGEN) and rotated at 4°C for 2 hr. Beads were then washed three times with 'lysis buffer' (including the corresponding NaCl) but with 0.1% digitonin. Protein was eluted from the beads with 300 mM imidazole, 2% SDS, 10% glycerol in 20 mM HEPES-KOH (pH 7.4). For mass spectrometry, samples were briefly run into an SDS-PAGE gel. Whole lanes were excised, and protein

was subject to in-gel trypsin digestion. To ensure reproducibility, the eluate used for mass spectrometry was derived from two independent mitochondrial preparations per strain, and affinity chromatography was performed on 2 separate days.

HA-tagged ANT1 affinity purification was performed on whole-cell lysate. Each replicate was from an independent transfection of a 10 cm dish of HeLa cells. Briefly, 24 hr after transfection, cells were collected, washed twice in cold PBS, and then lysed in 0.5 mL lysis buffer (50 mM HEPES-KOH pH 7.4, 10% glycerol, 100 mM NaCl, 1% digitonin, 1 mM PMSF, and 1× HALT Protease and Phosphatase Inhibitor Cocktail (Thermo Fisher)) for 30 min on ice. Lysate was cleared for 30 min at 16,000×*g*, and supernatant applied to preequilibrated Pierce anti-HA agarose beads (Thermo Fisher) and incubated overnight with gentle agitation. Beads were subsequently washed five times with lysis buffer, except containing 0.1% digitonin, followed by elution with 6% SDS and 10% glycerol, in 50 mM HEPES-KOH pH 7.4.

## Sample processing for mass spectrometry

The excised lanes of yeast eluate (see above) were subjected to in-gel trypsin digestion (*Shevchenko et al., 2006*). Briefly, gel pieces were washed with 50 mM ammonium bicarbonate (Acros) in 50% acetonitrile (ACN) (Fisher), reduced with dithiothreitol (Acros) and alkylated with iodoacetamide (Sigma), washed again, and impregnated with 75 µL of 5 ng/µL trypsin (trypsin gold; Promega) solution overnight at 37°C. The resulting peptides were extracted using solutions of 50% and 80% ACN with 0.5% formic acid (Millipore), and the recovered solution dried down in a vacuum concentrator. Dried peptides were dissolved in 60 µL of 0.1% trifluoroacetic acid (TFA, Sigma), and desalted using 2-core MCX stage tips (3 M) (*Rappsilber et al., 2003*). The stage tips were activated with ACN followed by 3% ACN with 0.1% TFA. Next, samples were applied, followed by two washes with 3% ACN with 0.1% TFA, and one wash with 65% ACN with 0.1% TFA. Peptides were eluted with 75 µL of 65% ACN with 5% NH₄OH (Sigma), and dried.

Cytosolic fractions from HeLa cells were processed using the FASP method (*Wiśniewski et al., 2009*). Briefly, in-solution proteins were reduced and denatured with DTT and SDS, mixed with urea to 8 M, and concentrated on a 10 kDa MWCO membrane filter (Pall, OD010C34). Cysteine residues were alkylated using iodoacetamide (Sigma) at room temperature in a dark location for 25 min. The proteins were rinsed with urea and ammonium bicarbonate solutions and digested overnight at 37°C using trypsin gold (Promega) at a ratio of 1:100. The resulting peptides were recovered from the filtrate and a 10 µg aliquot was desalted on 2-core MCX stage tips as above.

## LC-MS methods

Samples were dissolved in 20–35 µL of water containing 2% ACN and 0.5% formic acid to 0.25 µg/µL. Two µL (0.5 µg) were injected onto a pulled tip nano-LC column with 75 µm inner diameter packed to 25 cm with 3 µm, 120 Å, C18AQ particles (Dr. Maisch). The column was maintained at 45°C with a column oven (Sonation GMBH). The peptides were separated using a 60 min gradient from 3–28% ACN over 60 min, followed by a 7 min ramp to 85% ACN. The column was connected inline with an Orbitrap Lumos via a nanoelectrospray source operating at 2.2 kV. The mass spectrometer was operated in data-dependent top speed mode with a cycle time of 2.5 s. MS[1] scans were collected at 60,000 resolution with AGC target of 6.0E5 and maximum injection time of 50 ms. HCD fragmentation was used followed by MS[2] scans in the Orbitrap at 15,000 resolution with AGC target 1.0E4 and 100 ms maximum injection time.

## Database searching and label-free quantification

The MS data was searched using SequestHT in Proteome Discoverer (version 2.4, Thermo Scientific) against the *Saccharomyces cerevisiae* proteome from Uniprot, containing 6637 sequences, concatenated with common laboratory contaminant proteins. Enzyme specificity for trypsin was set to semi-tryptic with up to two missed cleavages. Precursor and product ion mass tolerances were 10 ppm and 0.6 Da, respectively. Cysteine carbamidomethylation was set as a fixed modification. Methionine oxidation was set as a variable modification. The output was filtered using the Percolator algorithm with strict FDR set to 0.01. Label-free quantification was performed in Proteome Discoverer with normalization set to total peptide amount.

## Label-free quantitation data processing

For analysis, label-free quantitation with Proteome Discoverer software generated protein abundances for each sample. Protein abundances were then analyzed using Metaboanalyst software (*Pang et al., 2020*). First, we refined the protein list to include only proteins whose fold-change is >1.5 (p<0.1) in Aac2p-HIS6 eluate compared with the null control. Then, to eliminate any bias introduced by mutant bait protein Aac2p$^{A128P}$-HIS6 being present at ~50% the level of Aac2-HIS6 (see *Figure 4—figure supplement 1C*; *Figure 4—figure supplement 2A*), we normalized each prey protein abundance by that sample's bait protein level (i.e. Aac2p level). Finally, we performed multiple *t* testing on these values, which generated the FDR-corrected p-values and were ultimately normalized to wild-type level for presentation in *Figure 4C*, *Figure 4—figure supplement 1I*, *Figure 4—figure supplement 2B and E*. Gene Ontology (GO) and other enrichment analyses were performed using STRING version 11.0 (*Szklarczyk et al., 2019*).

For the analysis shown in *Figure 5F*, the protein list was manually curated to include only proteins that had a GO Cellular Component term for 'mitochondrion', and not that of any other organelle. Proteins with missing values were excluded. The levels of each protein were averaged within an experimental group and those average values were normalized to the average value for wild-type *SLC25A4*-transfected samples. It is these wild-type-normalized average values for each protein that were subject to Student's *t* test to probe for a significant difference in mitochondrial protein levels present in the cytosol of mutant compared with wild-type *SLC25A4*-transfected HeLa cells.

## Targeted quantitative proteomics

Unused yeast peptides from the label-free quantification experiment were used for absolute quantification using parallel reaction monitoring on the instrument described above. The instrument method consisted of one MS scan at 60,000 resolution followed by eight targeted Orbitrap MS$^2$ scans at 30,000 resolution using quadrupole isolation at 1.6 m/z and HCD at 35%.

Two heavy-labeled proteotypic peptides for Aac2p and Tim22p were purchased from New England Peptide. Their sequences were TATQEGVISFWR and SDGVAGLYR; and VYTGFGLEQISPAQK and TVQQISDLPFR, respectively, each with N-terminal $^{13}C_6$ $^{15}N_4$ arginine or $^{13}C_6$ $^{15}N_2$ lysine. An equal portion of the peptides contained in each gel band was combined with a mixture of proteotypic peptides resulting in an on-column load of 1 fmol of each Tim22p heavy peptide and 250 fmol of each Aac2p heavy peptide. Assays were developed for the doubly charged precursor and the four or five most intense singly charged product y-ions. The data was analyzed in Skyline (version 20.2) (*MacLean et al., 2010*).

## qRT-PCR

Total RNA was isolated from ~$1.5 \times 10^7$ yeast cells using Quick-RNA Fungal/Bacterial Microprep Kit (Zymo Research). Quantitative real-time PCR (qRT-PCR) was executed using 50 ng RNA in Power SYBR Green RNA-to-Ct 1-step Kit (Applied Biosystems). A CFX384 Touch Real-Time PCR Detection System (Bio-Rad) was used with the following cycling parameters: 48°C 30 min, 95°C 10 min, followed by 44 cycles of 95°C 15 s, 60°C 1 min. *TFC1* was used as reference, as its mRNA level was previously established as stable across culture conditions (*Teste et al., 2009*). The specificity of each primer pair was validated using RNA extracted from knockout strains. Ct values were determined using CFX Maestro software (Bio-Rad) and analyzed manually using the $2^{-\Delta\Delta CT}$ method.

## In vitro mitochondrial protein import assays

The *AAC2* variants were cloned into pGEM4z plasmids. The constructs were used for coupled in vitro transcription and translation, using a cell-free system based on reticulocyte lysate. The wild-type strains YPH499 and BY4741 and yeast expressing Tom40-HA were grown in YPG (1% [wt/vol] yeast extract, 2% [wt/vol] bacto-peptone, and 3% [vol/vol] glycerol) at 30°C. Mitochondria were isolated by differential centrifugation. The import of the Aac2p variants were performed as described (*Ellenrieder et al., 2019*). Isolated mitochondria were incubated for different time periods with radiolabeled Aac2p variants in the presence of 4 mM ATP and 4 mM NADH in import buffer (3% [wt/vol] bovine serum albumin; 250 mM sucrose; 80 mM KCl; 5 mM MgCl$_2$; 5 mM methionine; 2 mM KH$_2$PO$_4$; 10 mM MOPS/KOH, pH 7.2). The import reaction was stopped by addition of 8 µM antimycin A, 1 µM valinomycin, and 20 µM oligomycin (AVO; final concentrations). In control reactions, the membrane

potential was depleted by addition of the AVO mix. Subsequently, mitochondria were washed with SEM buffer (250 mM sucrose; 1 mM EDTA; 10 mM MOPS-KOH, pH 7.2), lysed with 1% (wt/vol) digitonin in lysis buffer (0.1 mM EDTA; 50 mM NaCl; 10% [vol/vol] glycerol; 20 mM Tris-HCl, pH 7.4) for 15 min on ice analysis and protein complexes were separated on blue native gels. To remove non-imported preproteins, mitochondria were treated with 50 µg/mL proteinase K for 15 min on ice. The protease was inactivated by addition of 1 mM PMSF for 10 min on ice. To study the accumulation of Aac2p variants at the TOM translocase, the Aac2p variants were imported into Tom40-HA mitochondria followed by affinity purification via anti-HA beads (Roche) (*Ellenrieder et al., 2019*). After the import reaction, mitochondria were lysed with 1% (wt/vol) digitonin in lysis buffer for 15 min on ice. After removing insoluble material, the mitochondrial lysate was incubated with anti-HA matrix for 60 min at 4°C. Subsequently, beads were washed with an excess amount of 0.1% (wt/vol) digitonin in lysis buffer and bound proteins were eluted under denaturing conditions using SDS sample buffer (2% [wt/vol] SDS; 10% [vol/vol] glycerol; 0.01% [wt/vol] bromophenol blue; 0.2% [vol/vol] β-mercaptoethanol; 60 mM Tris/HCl, pH 6.8).

## Pulse-chase analysis of Aac2p$^{A128P, A137D}$

Cells with disrupted endogenous *AAC2* but expressing *GAL10-aac2$^{A128P, A137D}$*, with or without the disruption of *YME1* and *PDR5*, were grown in YPD at 30°C overnight. Cells were then subcultured in complete galactose plus raffinose medium for 4 hr at 30°C. Cycloheximide was added to a concentration of 1 mg/mL, with or without the addition of 100 µM MG132. Cycloheximide chase was pursued at 30°C for 30, 60, and 120 min before cells were collected for western blot analysis.

## Human cell culture

HeLa cells were cultured in DMEM (Gibco) with 10% fetal bovine serum (Sigma) at 37°C in a humidified atmosphere of 5% $CO_2$. The HeLa cell line was purchased from ATCC (CCL-2). Cell line was authenticated by ATCC with short tandom repeat analysis. Cells tested free of mycoplasma by the VenorGem Mycoplasma Detection Kit (Sigma-Aldrich #MP0025).

## Expression of *SLC25A4* in HeLa cells

*SLC25A4* cDNA was cloned into pCDNA3.1 with an HA epitope added to the C-terminus, as previously described (*Liu et al., 2019*). Mutant *SLC25A4* alleles were generated by in vitro mutagenesis using QuikChange Site-Directed Mutagenesis (Stratagene) and confirmed by sequencing. Cells were transfected using Lipofectamine 3000 (Invitrogen) according to the manufacturer's protocol and harvested 24 hr after transfection for all experiments.

## HeLa cell fractionation

One 10 cm dish per sample was harvested, washed twice in cold PBS, homogenized in 1 mL isotonic buffer (250 mM sucrose, 1 mM EDTA, 10 mM Tris-HCl, pH 7.4,) including HALT Protease and Phosphatase Inhibitor Cocktail (Thermo Fisher) with 10 slow strokes in a 2 mL Dounce homogenizer (KIMBLE, pestle B, clearance = 0.0005–0.0025 inches). Homogenate was centrifuged at 600×*g* for 15 min. To pellet mitochondria, supernatant was spun at 10,000×*g* for 25 min. Supernatant was the cytosolic fraction and pellet (the mitochondrial fraction) was used subsequently for protease protection or western blotting. All steps were performed on ice.

## Protease protection assay in HeLa cells

For protease protection, isolated mitochondria from four 10 cm plates were combined to generate each replicate indicated in *Figure 5E*. All replicates were from independent transfections on different days. Twenty µg aliquots of mitochondria were pelleted and resuspended in either isotonic buffer (as above, without protease inhibitors) or hypotonic buffer (10 mM KCl, 2 mM HEPES, pH 7.2) for swelling on ice for 20 min. Where indicated, proteinase K was added at a final concentration of 7 µg/mL, incubated at room temperature for 20 min, and inhibited with 5 mM PMSF on ice for 15 min. Where indicated, 1% Triton X-100 treatment on ice for 10 min was used to lyse the mitochondrial membranes. Ultimately, half of each reaction was loaded onto two independent gels for western blotting. Immunoblotting of reference proteins in *Figure 5D* was done for each replicate. If a reference protein was not properly protected/degraded, this sample was discarded.

## Apoptosis assay and relative Δψ determination

Cells were harvested 24 hr post-transfection and processed for flow cytometry detection of apoptosis and membrane potential after staining with propidium iodide and Annexin V-FITC and JC-1, respectively (*Liu et al., 2019*).

## *Slc25a4* *p.A114P,A123D*/+ knock-in mice

All procedures were approved by the Animal Care and Use Committee (IACUC) at State University of New York Upstate Medical University and were in accordance with guidelines established by the National Institutes of Health.

The *Slc25a4* targeting vector was prepared by recombineering as previously described (*Lee et al., 2001*). Briefly, 13.8 kb of *Slc25a4* genomic sequence containing all four exons plus 4.4 kb of 5' upstream and 4.96 kb 3' downstream untranslated sequences was retrieved from the RP24-108A1 BAC clone obtained from the BACPAC Resources Center (Children's Hospital Oakland Research Institute, Oakland, CA, USA). The first loxP site together with the Frt-PGKnew-Frt cassette was inserted approximately 0.4 kb 3' of exon 4, which contains the polyA signal sequence and the 3'UTR. Two unique restriction sites, *Asc*I and *Asi*SI, were also introduced into the 3' end of the second loxP site to allow insertion of the *Slc25a4* mini-cDNA containing the A114P and A123D knock-in mutations as well as the His-tag at the carboxyl terminus.

The *Slc25a4* mini-cDNA was prepared by fusion PCR using primers with sequences overlapping different exons. The mini-cDNA contains *Asc*I and *Asi*SI unique restriction sites in the 5' and 3' end, respectively, together with 254 bp of intron 1 sequence together with the splice acceptor followed by exon 2 with the two mutations, exons 2 and 4, and 70 bp of 3' downstream sequence. The mini-cDNA was cloned into pSK+ and sequenced to confirm its identity prior to insertion into the *Asc*I and *Asi*SI sites in the targeting vector. The final targeting vector was then linearized by *Not*I digestion, purified, and resuspended in PBS at 1 mg/μL for electroporation into ES cells derived from F1 (129Sv/C57BL6j) blastocyst. Targeted ES clones were identified by long-range nested PCR using Platinum HiFi Taq (Invitrogen).

Chimeric animals were generated by aggregation of ES cells with the CD1 morula. Chimeric males were bred with ROSA26-Flpe female (Jackson Labs stock no: 009086) to remove the PGKnew cassette and generate F1 pups with *Slc25a4* floxed allele. Positive pups were identified by PCR genotyping using primer Lox gtF (5'-ATCCATCTCAAAGGCAAACG-3') and Lox gtR (5'- AAATTCCCTGCAGGCT TATG-3') to detect a fragment of 364 bp specific to the 5'-Lox site and a fragment of 270 bp specific to the wild-type allele. A heterozygous floxed male was bred with Hprt-Cre female (Jackson Labs stock no: 004032). The same primers were used for genotyping knock-in mice, with the 270 bp band present only with the knock-in allele, and no band produced from a wild-type locus with lacking loxP sites. The mixed background heterozygous knock-in (i.e. *Slc25a4* *p. A114P,A123D*/+) male mice were back-crossed with C57BL/6NTac females (Taconic Catalog no: B6-F) for seven generations before experiments were performed.

## Mouse histology

For spinal cord histology, mice were sacrificed by isoflurane overdose, followed by intracardial perfusion with PBS followed by PBS+4% paraformaldehyde (PFA). Perfused mice were soaked in PBS+4% PFA overnight at 4°C, and then spinal cord was dissected and cryopreserved with increasing concentrations of sucrose in PBS. Tissue was then embedded in OCT and snap-frozen in 2-methylbutane on liquid $N_2$. Tissue was sectioned at 8 μm with a cryostat (Leica) and sections stained with Cresyl Violet Stain Solution according to the manufacturer's protocol (Abcam) or used for indirect immunofluorescence. For the latter, tissue was blocked with 10% horse serum and then incubated with rat monoclonal anti-GFAP antibody (Invitrogen) overnight at 4°C. After rinsing the tissue was incubated with fluorescently conjugated anti-rat secondary antibody (ImmunoResearch, West Grove, PA, USA).

For muscle histology, the soleus muscles were quickly dissected and fresh-frozen in 2-methylbutane on liquid $N_2$. Tissue was cryosectioned at 10 μm and stained with hematoxylin and eosin using standard procedures. Feret's diameter of myofibers was determined using ImageJ software; all soleus myofibers from a single muscle section per mouse (n>340 fibers per mouse) were quantitated by a blinded observer. For SDH-only staining, sections were air-dried, incubated at 37°C for 45 min in SDH medium (0.1 M succinic acid, 0.1 M sodium phosphate buffer pH 7, 0.2 mM phenazine methosulfate,

+1 mg/mL NBT added fresh), drained, fixed in 10% formalin, rinsed well with water, and mounted in water soluble mounting medium. For sequential COX/SDH staining, sections were air-dried, incubated in cytochrome *c* medium (0.5 mg/mL 3'3'-diaminobenzidine tetrahydrochloride, 75 mg/mL sucrose in 50 mM sodium phosphate buffer, pH 7.4, with freshly added cytochrome *c* and catalase at 1 and 0.1 mg/mL, respectively) at 37°C for 1 hr, followed by a quick wash in water and SDH staining as described above. Abnormal COX/SDH-stained fibers were scored manually from decoded images of whole soleus sections.

## Electron microscopy

For electron microscopy, *Slc25a4* [p.A114P,A123D]/+ mice and littermate controls were processed as previously described (*Massa et al., 2004*). Briefly, mice were anesthetized with isoflurane and perfused intracardially with PBS initially, followed by fixative (1% PFA, 1% glutaraldehyde, 0.12 M sodium cacodylate buffer pH 7.1, and 1 mM $CaCl_2$). Perfused animals were refrigerated overnight, and CNS tissues were dissected the next day and processed for TEM. The samples were examined with a JOEL JEM1400 transmission electron microscope and images were acquired with a Gaten DAT-832 Orius camera.

## Bioenergetic analysis

For mitochondrial respiration experiments, the mice were sacrificed by decapitation via guillotine without the use of $CO_2$ asphyxiation or anesthetic. Skeletal muscle mitochondria were isolated and respiration measured as previously described (*Garcia-Cazarin et al., 2011*). Briefly, after mitochondria isolation by differential centrifugation, oxygen tension was measured using an Oxygraph Plus oxygen electrode (Hansatech Instruments) in 0.5 mL experimental buffer containing 150 µg mitochondria in a temperature-controlled 37°C chamber. For complex I measurements, glutamate and malate were added for a final concentration of 5 and 2.5 mM, respectively. For complex II measurements, rotenone was added before succinate, for final concentrations of 5 µM and 10 mM, respectively.

## Protease protection assay of mouse skeletal muscle mitochondria

For protease protection assay, skeletal muscle mitochondria were isolated as for bioenergetic analysis with slight modifications, including the addition of 1 mM PMSF in the homogenization buffer, and resuspending mitochondria in a modified isotonic buffer lacking BSA (75 mM sucrose, 215 mM mannitol, 1 mM EGTA in 20 mM HEPES-KOH pH 7.4). Eighty µg aliquots were treated with the indicated concentrations of proteinase K (Sigma) for 30 min at room temperature and quenched with 5 mM PMSF on ice for 10 min. Where indicated, mitochondria were lysed with 1% Triton X-100 for 30 min on ice. Laemmli buffer was ultimately added to the samples such that the final protein concentration was ~40 µg mitochondria per 15 µL, which was the amount loaded onto the SDS-PAGE gel for western blotting. Such high protein amounts were required for detection of Ant1[A114P,A123D].

## BN-PAGE

Snap-frozen mouse skeletal muscle mitochondria isolated for bioenergetic analysis were used for BN-PAGE. Protein complexes were solubilized with digitonin at a 4:1 detergent:protein ratio and loaded into native PAGE gel (Invitrogen) per the manufacturer's instructions. For western blotting, gels were washed in 2% SDS followed by standard electrophoretic transfer to a PVDF membrane. Following transfer, membrane was fixed with acetic acid and dried followed by standard western blotting techniques.

## Mouse skeletal muscle sub-cellular fractionation

Five biological replicates of wild-type and *Slc25a4* [p.A114P,A123D]/+ muscle samples were fractionated to obtain cytosolic and mitochondrial fractions using differential centrifugation, as follows. One-hundred mg Quadriceps muscle was rapidly dissected and immediately placed in ice-cold PBS. Muscle was minced, centrifuged for 1 min at 500×*g*, and resuspended in 1 mL buffer STM (250 mM sucrose, 50 mM Tris-HCl pH 7.4, 5 mM $MgCl_2$ plus HALT protease and phosphatase inhibitors added fresh). Tissue was homogenized in a 2 mL Dounce homogenizer (0.15–0.25 mm clearance) with two strokes using a bench top drill press set to 570 rotations per minute. Homogenate was centrifuged for 15 min at 800×*g* twice, and the pellets were discarded after both spins. To obtain the mitochondrial

fraction, supernatant was centrifuged for 11,000×g for 10 min. Mitochondrial fraction was washed twice in STM and frozen for further analysis. Meanwhile, the supernatant was centrifuged for 30 min at 21,000×g twice to remove any contaminating mitochondria from the cytosolic fraction, although this may not completely eliminate mitochondrial proteins trapped in small mitochondria-derived vesicles.

## Mouse skeletal muscle cytosolic and mitochondrial protein digestion, labeling, cleanup, and fractionation

Skeletal muscle cytosolic and mitochondrial fractions were then processed separately for multiplexed quantitative mass spectrometry as follows. Samples were buffer exchanged on a 3 kDa molecular weight cutoff filter (Amicon 3k Ultracel) using four additions of 50 mM triethylammonium bicarbonate, pH 8.0 (Thermo). Following a Bradford assay, 50 µg of each cytosolic fraction was taken for digestion using an EasyPep Mini MS sample prep kit (Thermo, A40006). To each buffer-exchanged sample, 70 µL of lysis buffer was added followed by 50 µL of reduction solution and 50 µL of alkylating solution. Samples were incubated at 95°C for 10 min, then cooled to room temperature. To each sample 2.5 µg of trypsin/Lys-C protease was added and the reaction was incubated at 37°C overnight. TMT reagents were reconstituted with 40 µL ACN and the contents of each label added to a digested sample. After 60 min, 50 µL of quenching solution was added, consisting of 20% formic acid and 5% ammonium hydroxide (vol/vol) in water. The labeled digests were cleaned up by a solid-phase extraction contained in the EasyPep kit, and dried by speed-vac. The 10 cytosolic fractions were dissolved in 50 µL of 30% ACN and 0.1% formic acid (v/v) in water, combined, and dried again.

Following an LC-MS experiment to check digestion and labeling quality of the pooled samples, these were fractionated using a Pierce High pH Reversed-Phase Peptide Fractionation Kit (part # 84868), per the manufacturer's instructions for TMT-labeled peptides. In brief, samples were dissolved in 300 µL of 0.1% TFA in water and applied to the conditioned resin. Samples were washed first with water and then with 300 µL of 5% ACN, 0.1% triethylamine (TEA) in water. The second wash was collected for analysis. Peptides were step eluted from the resin using 300 µL of solvent consisting of 5–50% ACN with 0.1% TEA in eight steps. All collected fractions were dried in a speed-vac.

## LC-MS/MS for TMT-labeled samples

Dried fractions were reconstituted in 50 µL of load solvent consisting of 3% ACN and 0.5% formic acid in water. Of these, 2 µL were injected onto a pulled tip nano-LC column (New Objective, FS360-75-10-N) with 75 µm inner diameter packed to 25 cm with 2.2 µm, 120 Å, C18AQ particles (Dr. Maisch GmbH). The column was maintained at 50°C with a column oven (Sonation GmbH, PRSO-V2). The peptides were separated using a 135 min gradient consisting of 3–12.5% ACN over 60 min, 12.5–28% over 60 min, 28–85% ACN over 7 min, a 3 min hold, and 5 min re-equilibration at 3% ACN. The column was connected inline with an Orbitrap Lumos (Thermo) via a nanoelectrospray source operating at 2.3 kV. The mass spectrometer was operated in data-dependent top speed mode with a cycle time of 3 s. MS[1] scans were collected from 375 to 1500 m/z at 120,000 resolution and a maximum injection time of 50 ms. HCD fragmentation at 40% collision energy was used followed by MS[2] scans in the Orbitrap at 50,000 resolution with a 105 ms maximum injection time.

## Database searching and reporter ion quantification

The MS data was searched using SequestHT in Proteome Discoverer (version 2.4, Thermo Scientific) against the *Mus musculus* proteome from Uniprot, containing 50,887 sequences, concatenated with common laboratory contaminant proteins. Enzyme specificity for trypsin was set to semi-tryptic with up to two missed cleavages. Precursor and product ion mass tolerances were 10 ppm and 0.6 Da, respectively. Cysteine carbamidomethylation, TMT 10-plex at any N-terminus, and TMT 10-plex at lysine were set as a fixed modifications. Methionine oxidation was set as a variable modification. The output was filtered using the Percolator algorithm with strict FDR set to 0.01. Quantification parameters included the allowance of unique and razor peptides, reporter abundance based on intensity, lot-specific isotopic purity correction factors, normalization based on total peptide amount, protein ratio based on protein abundance, and hypothesis testing (ANOVA) for individual proteins.

## Grip strength measurements

Grip strength measurements were performed using Bio-CIS software connected with the Grip Strength Test Model GT3 according to the manufacturer's protocol for forelimb-only measurement using the grid (BIOSEB). Briefly, mice were held by the tail above the grid that's connected to a force meter, slowly lowered to allow the forelimbs to grip the grid, and then slowly and smoothly pulled horizontally along the axis of the sensor until the grasp was released. The maximum force generated is recorded by the software and reported as the average of five consecutive trials or the maximum force generated over five trials.

## RNA extraction, sequencing, and analysis

Skeletal muscle was disputed in QIAzol lysis reagent, using the QIAGEN TissueRuptor. RNA was then extracted using the Qiagen miRNeasy Mini Kit. RNA quality and quantity were assessed with the RNA 6000 Nano kit on the Agilent 2100 Bioanalyzer. Sequencing libraries were prepared using the Illumina TruSeq Stranded mRNA Library Prep kit, using 1 µg total RNA as input. Library size was assessed with the DNA 1000 Kit on the Agilent 2100 Bioanalyzer. Libraries were quantified using the Quant-IT High Sensitivity dsDNA Assay (Invitrogen) on a Qubit 3.0 Fluorometer. Libraries were sequenced on the NextSeq 500 instrument, with paired end 2×75 bp reads.

For proper comparison with Ant1 knockout mice, thresholds for significance and fold-change were replicated exactly as previously published (*Morrow et al., 2017*): a significance threshold of $p<0.01$, >1.5-fold increase for upregulated genes, >0.3-fold decrease for downregulated genes, and enrichment analysis was done using the Database for Annotation, Visualization and Integrity Discovery (DAVID) software (*Dennis et al., 2003*). Databases analyzed were also kept consistent with (*Dennis et al., 2003*), and included GO, Sequence Features (Seq), InterPro, Kyoto Encyclopedia of Genes and Genomes (KEGG), and Protein Information Resource (PIR). We were unable to perform a direct comparison of significantly changed genes in *Slc25a4* $^{p.A114P,A123D}$/+ vs Ant1 knockout mice because the data are not publicly available.

## Statistical analysis

Statistical analyses were performed using GraphPad Prism. For details on statistical testing of specific data, please see figure legends.

# Acknowledgements

We thank Nikolaus Pfanner for support and anti-sera, Joyce Qi for help with electron microscopy, and Siu-Pok Yee (University of Connecticut) for *Slc25a4* $^{p.A114P,A123D}$ knock-in mouse generation. We're also grateful to Yumiko Umino and Eduardo Solessio for confirming visual acuity and contrast sensitivity in *Slc25a4* $^{p.A114P,A123D}$/+ mice.

# Additional information

### Funding

| Funder | Grant reference number | Author |
|---|---|---|
| National Institute on Aging | R01AG063499 | Xin Jie Chen |
| National Institute on Aging | R01AG061204 | Xin Jie Chen |
| National Institute on Aging | F30AG060702 | Liam P Coyne |
| Deutsche Forschungsgemeinschaft | project ID 269925409 | Thomas Becker |
| Deutsche Forschungsgemeinschaft | BE 4679 2/2 | Thomas Becker |

The funders had no role in study design, data collection and interpretation, or the decision to submit the work for publication.

## Author contributions

Liam P Coyne, Conceptualization, Data curation, Formal analysis, Funding acquisition, Validation, Investigation, Visualization, Methodology, Writing – original draft, Project administration, Writing – review and editing; Xiaowen Wang, Data curation, Formal analysis, Validation, Investigation, Visualization, Methodology; Jiyao Song, Formal analysis, Validation, Investigation, Visualization, Methodology; Ebbing de Jong, Data curation, Validation, Methodology; Karin Schneider, Data curation, Visualization; Paul T Massa, Formal analysis, Supervision, Validation, Investigation, Writing – review and editing; Frank A Middleton, Formal analysis, Supervision, Writing – review and editing; Thomas Becker, Data curation, Formal analysis, Supervision, Funding acquisition, Validation, Visualization, Methodology, Writing – review and editing; Xin Jie Chen, Conceptualization, Resources, Data curation, Formal analysis, Supervision, Funding acquisition, Validation, Investigation, Visualization, Methodology, Writing – original draft, Project administration, Writing – review and editing

## Author ORCIDs

Liam P Coyne ⓘ http://orcid.org/0000-0002-4480-126X
Xin Jie Chen ⓘ http://orcid.org/0000-0002-8488-6587

## Ethics

This study was performed in strict accordance with the recommendations in the Guide for the Care and Use of Laboratory Animals of the National Institutes of Health. All of the animals were handled according to approved institutional animal care and use committee (IACUC) protocols (D1600318) of the State University of New York Upstate Medical University. The protocol was approved by the Committee on the Ethics of Animal Experiments of SUNY Upstate Medical University (Permit Number: #268).

## Decision letter and Author response

Decision letter https://doi.org/10.7554/eLife.84330.sa1
Author response https://doi.org/10.7554/eLife.84330.sa2

# Additional files

## Supplementary files

• MDAR checklist

## Data availability

The small datasets generated in the study are included as source data files. The RNA-seq data has been deposited to NCBI Gene Expression Omnibus/Sequence Read Archive with the accession number GSE227295.

The following dataset was generated:

| Author(s) | Year | Dataset title | Dataset URL | Database and Identifier |
|---|---|---|---|---|
| Coyne LP, Chen XJ | 2023 | Mitochondrial protein import clogging as a mechanism of disease | http://www.ncbi.nlm.nih.gov/geo/query/acc.cgi?acc=GSE227295 | NCBI Gene Expression Omnibus, GSE227295 |

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

# Appendix 1

## Appendix 1—key resources table

| Reagent type (species) or resource | Designation | Source or reference | Identifiers | Additional information |
|---|---|---|---|---|
| Cell line (*Homo sapiens*) | HeLa | ATCC | CCL-2 | |
| Antibody | Polyclonal rabbit ant-Aac2 | Chen Lab | | 1:3000 |
| Antibody | Polyclonal rabbit anti-Ilv5 | Chen Lab | | 1:5000 |
| Antibody | Polyclonal rabbit anti-Hsp60 | Chen Lab | | 1:10,000 |
| Antibody | Polyclonal rabbit anti-TFAM | Sigma | Cat. #SAB1401383-100UG | 1:1000 |
| Antibody | Monoclonal mouse anti-hemagglutinin (HA) | Covance | Cat. #MMS-101R | 1:2000 |
| Antibody | Polyclonal rabbit anti-Tim22 (yeast) | Nikoalus Pfanner Lab | 5113 | 1:10,000 |
| Antibody | Polyclonal rabbit anti-Tim23 (yeast) | Ron Butow Lab | | 1:5000 |
| Antibody | Monoclonal rabbit anti-Tom20 (human) | Cell Signaling | Cat. #42406 | 1:2000 |
| Antibody | Monoclonal rabbit anti-Tom40 (human) | Abcam | Cat. #ab185543 | 1:1000 |
| Antibody | Polyclonal rabbit anti-Tim22 (human) | Protein Tech | Cat. #14927–1-AP | 1:2000 |
| Antibody | Monoclonal mouse anti-Tim23 (human) | BD Biosciences | Cat. #611222 | 1:5000 |
| Antibody | Polyclonal rabbit anti-Smac (human) | Abcam | Cat. #ab8114 | 1:2000 |
| Antibody | Polyclonal rabbit anti-Ant1 | Sigma | Cat. #SAB2108761-100UL | 1:2000 |
| Antibody | Monoclonal rabbit anti-Mdh2 (D8Q5S) | Cell Signaling | Cat. #11908 | 1:2000 |
| Antibody | Monoclonal rat anti-GFAP (immunostaining) | Invitrogen | Cat. #13-0300 | 10 µg/mL |
| Antibody | Monoclonal mouse anti-GFAP (western blot) | Chemicon International | Cat. #MAB360 | 1:1000 |
| Antibody | Monoclonal mouse anti-GAPDH | Abcam | Cat. #ab9482 | 1:2000 |
| Antibody | Monoclonal mouse anti-Pgk1 | Invitrogen | Cat. #459250 | 1:4000 |
| Antibody | Polyclonal rabbit anti-Sml1 | Rothstein lab | | 1:5000 |
| Antibody | Polyclonal rabbit ant-eIF2α | Cell Signaling | Cat. #9722 | 1:2000 |
| Antibody | Monoclonal rabbit ant-Phospho-eIF2α | Cell Signaling | Cat. #3597 | 1:1000 |
| Antibody | Monoclonal mouse anti-NDUFA9 | Abcam | Cat. #ab14713 | 1:1000 |
| Antibody | Monoclonal mouse MitoProfile Total OXPHOS Human WB Antibody Cocktail | Abcam | Cat. #ab110411 | 1:500 |
| Chemical compound, drug | L-[$^{35}$S]-methionine | Perkin Elmer | Cat. #NEG009005MC | |
| Commercial assay or kit | Anti-HA affinity matrix (used for yeast) | Roche | Cat. #11815016001 | |
| Commercial assay or kit | Anti-HA beads (used for HeLa cells) | Thermo Scientific | Cat. #26181 | |
| Commercial assay or kit | Ni-NTA agarose beads | QIAGEN | Cat. #1018244 | |
| Commercial assay or kit | HALT Protease and Phosphatase Inhibitor Cocktail | Thermo Scientific | Cat. #1861284 | |
| Commercial assay or kit | JC-1 | Life Technologies | Cat. #T3168 | |

*Appendix 1 Continued on next page*

*Appendix 1 Continued*

| Reagent type (species) or resource | Designation | Source or reference | Identifiers | Additional information |
|---|---|---|---|---|
| Peptide, recombinant protein | Aac2 Peptide 1 | New England Peptide | | TATQEGVISFWR |
| Peptide, recombinant protein | Aac2 Peptide 2 | New England Peptide | | SDGVAGLYR |
| Peptide, recombinant protein | Tim22 Peptide 1 | New England Peptide | | VYTGFGLEQISPAQK |
| Peptide, recombinant protein | Tim22 Peptide 2 | New England Peptide | | TVQQISDLPFR |
| Commercial assay or kit | TNT Quick Coupled Reaction Mix | Promega | Cat. #L2080 | |
| Commercial assay or kit | FITC Annexin V Apoptosis Detection Kit with PI | BioLegend | Cat. #640914 | |
| Commercial assay or kit | Quick-RNA Fungal/Bacterial Microprep Kit | Zymo | Cat. #R2010 | |
| Commercial assay or kit | Power SYBR Green RNA-to-Ct 1-step Kit | Thermo Fisher Scientific | Cat. #4389986 | |
| Commercial assay or kit | Lipofectamine 3000 | Invitrogen | Cat. #L3000-015 | |
| Commercial assay or kit | QuikChange Site-Directed Mutagenesis | Stratagene | Cat. #200518 | |
| Commercial assay or kit | Revert 700 Total Protein Stain | LI-COR | Cat. #926–11021 | |
| Biological sample (*Saccharomyces cerevisiae*) | W303-1B | R. Rothstein | | *MAT**a**, ade2, trp1, his3, leu2, ura3* |
| Biological sample (*Saccharomyces cerevisiae*) | CS1382-4A | This study | | as W303-1B, but *trp1Δ::aac2$^{A128P}$-URA3* |
| Biological sample (*Saccharomyces cerevisiae*) | CS1458/1 | This study | | as W303-1B, but *trp1Δ::aac2$^{A137D}$-URA3* |
| Biological sample (*Saccharomyces cerevisiae*) | CS1763-5A | This study | | as W303-1B, but *lys2Δ::aac2$^{A128P, A137D}$-kan* |
| Biological sample (*Saccharomyces cerevisiae*) | CS341/1 | Chen lab | | as W303-1B, but *aac2Δ::kan* |
| Biological sample (*Saccharomyces cerevisiae*) | CY4193 | This study | | asW303-1B, but *aac2Δ::LEU2, trp1Δ::aac2$^{A128P}$-URA3* |
| Biological sample (*Saccharomyces cerevisiae*) | CS1762/2-8A | This study | | as W303-1B, but *aac2Δ:kan, lys2Δ::aac2$^{A137D}$-kan* |
| Biological sample (*Saccharomyces cerevisiae*) | CS1763-7D | This study | | as W303-1B, but *aac2Δ:kan, lys2Δ::aac2$^{A128P,A137D}$-kan* |
| Biological sample (*Saccharomyces cerevisiae*) | CY6518 | This study | | as W303-1B, but *ura3::pUC-URA-GAL10-AAC2* |
| Biological sample (*Saccharomyces cerevisiae*) | CY6519 | This study | | as W303-1B, but *ura3::pUC-URA-GAL10-aac2$^{A128P}$* |

*Appendix 1 Continued on next page*

*Appendix 1 Continued*

| Reagent type (species) or resource | Designation | Source or reference | Identifiers | Additional information |
|---|---|---|---|---|
| Biological sample (*Saccharomyces cerevisiae*) | CY6520 | This study | | as W303-1B, but *ura3::pUC-URA-GAL10- aac2*$^{A137D}$ |
| Biological sample (*Saccharomyces cerevisiae*) | CY6521 | This study | | as W303-1B, but *ura3::pUC-URA-GAL10- aac2*$^{A128P, A137D}$ |
| Biological sample (*Saccharomyces cerevisiae*) | CY6513 | This study | | as W303-1B, but *aac2Δ::Kan, lysΔ::aac2*$^{A128P, A137D}$*-kan, pdr5Δ::Kan* |
| Biological sample (*Saccharomyces cerevisiae*) | CY6503 | This study | | as W303-1B, but *aac2Δ::kan, lys2Δ::aac2*$^{A128P, A137D}$*-kan.* |
| Biological sample (*Saccharomyces cerevisiae*) | CY6510 | This study | | as W303-1B, but *lys2Δ::aac2*$^{A128P, A137D}$*-kan, ump1Δ::kan* |
| Biological sample (*Saccharomyces cerevisiae*) | CY6511 | This study | | as W303-1B, but *aac2Δ::kan, lys2Δ::aac2*$^{A128P, A137D}$*-kan, pre9Δ::kan* |
| Biological sample (*Saccharomyces cerevisiae*) | CY6540 | This study | | as W303-1B, but *aac2Δ::kan, ura3::pUC-URA-GAL10-AAC2* |
| Biological sample (*Saccharomyces cerevisiae*) | CY6542 | This study | | as W303-1B, but *aac2Δ::kan, ura3::pUC-URA-GAL10-aac2*$^{A128P}$ |
| Biological sample (*Saccharomyces cerevisiae*) | CY6544 | This study | | as W303-1B, but *aac2Δ::kan, ura3::pUC-URA-GAL10-aac2*$^{A137D}$ |
| Biological sample (*Saccharomyces cerevisiae*) | CY6546 | This study | | as W303-1B, but *aac2Δ::kan, ura3::pUC-URA-GAL10-aac2*$^{A128P, A137D}$ |
| Biological sample (*Saccharomyces cerevisiae*) | CY6558 | This study | | as W303-1B, but *aac2Δ::kan, yme1Δ::Kan, ura3::pUC-URA-GAL10-aac2*$^{A128P, A137D}$ |
| Biological sample (*Saccharomyces cerevisiae*) | CY6562 | This study | | as W303-1B, but *aac2Δ::kan, pdr5Δ::Kan, ura3::pUC-URA-GAL10-aac2*$^{A128P, A137D}$ |
| Biological sample (*Saccharomyces cerevisiae*) | CY6569 | This study | | as W303-1B, but *aac2Δ::kan, pdr5Δ::Kan, yme1Δ::LEU2, ura3::pUC-URA-GAL10-aac2*$^{A128P, A137D}$ |
| Biological sample (*Saccharomyces cerevisiae*) | CY6440 | This study | | as W303-1B, but *pre9Δ::kan* |
| Biological sample (*Saccharomyces cerevisiae*) | CY6504 | This study | | as W303-1B, but *ump1Δ::kan* |
| Biological sample (*Saccharomyces cerevisiae*) | CY6581 | This study | | as W303-1B, but *ura3::pUC-URA-GAL10-aac2*$^{R96H}$ |
| Biological sample (*Saccharomyces cerevisiae*) | CY6583 | This study | | as W303-1B, but *ura3::pUC-URA-GAL10-aac2*$^{R252G}$ |
| Biological sample (*Saccharomyces cerevisiae*) | CY6775 | This study | | *MAT**a**/a, ade2/ade2, trp1/trp1, his3/his3, leu2/leu2, ura3/ura3, psd1Δ::kan/+, lys2Δ::aac2*$^{A128P, A137D}$*-kan/+* |
| Biological sample (*Saccharomyces cerevisiae*) | CY6777 | This study | | *MAT**a**/a, ade2/ade2, trp1/trp1, his3/his3, leu2/leu2, ura3/ura3, pel1Δ::LEU2/+, lys2Δ::aac2*$^{A128P, A137D}$*-kan/+* |

*Appendix 1 Continued on next page*

*Appendix 1 Continued*

| Reagent type (species) or resource | Designation | Source or reference | Identifiers | Additional information |
|---|---|---|---|---|
| Biological sample (*Saccharomyces cerevisiae*) | CY3326 | Chen lab | | *MAT**a**, his3Δ1, leu2Δ0, lys2Δ0, ura3Δ0, ura3Δ::aac2$^{A128P}$-URA3* |
| Biological sample (*Saccharomyces cerevisiae*) | BY4742/AG3 | Chen lab | | *MAT**a**, his3Δ1, leu2Δ0, lys2Δ0, ura3Δ0, trp1 Δ::GAL10-AAC2-HIS3* |
| Biological sample (*Saccharomyces cerevisiae*) | CY3322 | Chen lab | | *MAT**a**, his3Δ1, leu2Δ0, lys2Δ0, ura3Δ0, trp1 Δ::GAL10-AAC2$^{A128P}$-HIS3* |
| Biological sample (*Saccharomyces cerevisiae*) | BY4741 | EUROSCARF | | *MAT**a**, his3Δ1, leu2Δ0, met15Δ0, ura3Δ0* |
| Biological sample (*Saccharomyces cerevisiae*) | BY4741/ssa4D | Open Biosystems | | as BY4741, but *ssa4Δ::kan* |
| Biological sample (*Saccharomyces cerevisiae*) | BY4741/rpn4D | Open Biosystems | | as BY4741, but *rpn4Δ::kan* |
| Biological sample (*Saccharomyces cerevisiae*) | BY4741/hsp82D | Open Biosystems | | as BY4741, but *hsp82Δ::kan* |
| Biological sample (*Saccharomyces cerevisiae*) | BY4741/ssa3D | Open Biosystems | | as BY4741, but *ssa3Δ::kan* |
| Biological sample (*Saccharomyces cerevisiae*) | BY4741/cis1D | Open Biosystems | | as BY4741, but *cis1Δ::kan* |
| Biological sample (*Saccharomyces cerevisiae*) | BY4741/TOM40-HA | *Ellenrieder et al., 2019* | | as BY4741, but *tom40::TOM40HA-HIS3MX6* |
| Biological sample (*Saccharomyces cerevisiae*) | M2915-6A | Chen lab | | *MAT**a**, ade2, leu2, ura3* |
| Biological sample (*Saccharomyces cerevisiae*) | CY3323 | Chen lab | | *MAT**a**, ade2, ura3, leu2, ura3Δ::aac2$^{A128P}$-URA3* |
| biological sample (*Saccharomyces cerevisiae*) | YKSL210 | This study | | as M2916-6A, but *aac2Δ::LEU2, lys2Δ::AAC2-HIS$_6$-kan* |
| Biological sample (*Saccharomyces cerevisiae*) | YKSL211 | This study | | as M2916-6A, but *aac2Δ::LEU2, lys2Δ::aac2$^{A128P}$-HIS$_6$-kan* |
| Biological sample (*Saccharomyces cerevisiae*) | CY3904 | This study | | as M2915-6A, but *tom70Δ::kan* |
| biological sample (*Saccharomyces cerevisiae*) | CY6316 | This study | | as M2915-6A, but *tim18Δ::Kan* |
| Biological sample (*Saccharomyces cerevisiae*) | YPH499 | *Sikorski and Hieter, 1989* | | *MAT**a** ura3-52, lys2-801_amber, ade2-101_ochre, trp1-Δ63, his3-Δ200, leu2-Δ1* |
| Biological sample (*Mus musculus*) | C57BL6/NTac | Taconic | Cat. #: B6-F | |
| Biological sample (*Mus musculus*) | Hprt-Cre female | Jackson Labs | Stock no: 004032 | |

*Appendix 1 Continued on next page*

*Appendix 1 Continued*

| Reagent type (species) or resource | Designation | Source or reference | Identifiers | Additional information |
|---|---|---|---|---|
| Biological sample (*Mus musculus*) | Ant1^A114P,A123D knock-in mice | This study | | See Materials and methods |
| Sequence-based reagent | TFC1 fwd (B) | *Teste et al., 2009* | PCR primers | GCTGGCACTCATATCTTATCGTTTCACAATGG |
| Sequence-based reagent | TFC1 rev (B) | *Teste et al., 2009* | PCR primers | GAACCTGCTGTCAATACCGCCTGGAG |
| Sequence-based reagent | HSP82 fwd | *Boos et al., 2019* | PCR primers | GCTGCTTTGGCTAAGTTGTTACGTTAC |
| Sequence-based reagent | HSP82 rev | *Boos et al., 2019* | PCR primers | GAGATTCACCAGTGATGTAGTAGATGTTC |
| Sequence-based reagent | RPN4 fwd | *Boos et al., 2019* | PCR primers | GCAACAAGAGCAACACCAAGAGGAG |
| Sequence-based reagent | RPN4 rev | *Boos et al., 2019* | PCR primers | CTGTCCATGTTAGAGTCAACGTAACTG |
| Sequence-based reagent | CIS1 fwd | *Boos et al., 2019* | PCR primers | ATCAGTAATTGTCCCATCGGGTTAGTTTC |
| Sequence-based reagent | CIS1 rev | *Boos et al., 2019* | PCR primers | CCTGGGCAGCCTTGAGTAAATCATATC |
| Sequence-based reagent | SSA3 fwd (A) | This study | PCR primers | GGATAAGAAAGGCAGGGCTGA |
| Sequence-based reagent | SSA3 rev (A) | This study | PCR primers | CTGCGGTAGCCTTAACCTCAA |
| Sequence-based reagent | SSA4 fwd (B) | This study | PCR primers | AGGCAAGCAACAAAAGATGCC |
| Sequence-based reagent | SSA4 rev (B) | This study | PCR primers | TTGTCCAGCCCATACGCAATA |
| Sequence-based reagent | TOM70P1 | This study | PCR primers | GAAAGAGTTTCATTGCCATTAG |
| Sequence-based reagent | TOM70P2 | This study | PCR primers | TTGTGGTTTATACGCACTGC |
| Sequence-based reagent | TOM70P3 | This study | PCR primers | AACACTGTGCAGGCAACTTC |
| Sequence-based reagent | TOM70P4 | This study | PCR primers | CTCCGCAAATTGGCGAGG |
| Sequence-based reagent | TIM18KOFP | This study | PCR primers | CCATTCTCGCAAAAGATCGG |
| Sequence-based reagent | TIM18KORP | This study | PCR primers | TCTGGATTTCGAGAAGAAGG |
| Sequence-based reagent | TIM18GTFP | This study | PCR primers | GTCAGTGCCCTCGAGAGC |
| Sequence-based reagent | TIM18GTRP | This study | PCR primers | cccaagcttCGCAGATAGTGCGATAGTTG |
| Sequence-based reagent | Lox gtF | This study | PCR primers | ATCCATCTCAAAGGCAAACG |
| Sequence-based reagent | Lox gtR | This study | PCR primers | AAATTCCCTGCAGGCTTATG |
| Recombinant DNA reagent | pRS416 | Chen Lab | | |
| Recombinant DNA reagent | pRS416-AAC2 | This study | | See Materials and methods |

*Appendix 1 Continued on next page*

*Appendix 1 Continued*

| Reagent type (species) or resource | Designation | Source or reference | Identifiers | Additional information |
|---|---|---|---|---|
| Recombinant DNA reagent | pRS416-aac2(A106P) | This study | | See Materials and methods |
| Recombinant DNA reagent | pRS416-aac2(M114P) | This study | | See Materials and methods |
| Recombinant DNA reagent | pRS416-aac2(A128P) | This study | | See Materials and methods |
| Recombinant DNA reagent | pRS416-aac2(A137D) | This study | | See Materials and methods |
| Recombinant DNA reagent | pRS416-aac2(A106D,M114P) | This study | | See Materials and methods |
| Recombinant DNA reagent | pRS416-aac2(A106D,A128P) | This study | | See Materials and methods |
| Recombinant DNA reagent | pRS416-aac2(A106D,A137D) | This study | | See Materials and methods |
| Recombinant DNA reagent | pRS416-aac2(M114P,A128P) | This study | | See Materials and methods |
| Recombinant DNA reagent | pRS416-aac2(M114P,A137D) | This study | | See Materials and methods |
| Recombinant DNA reagent | pRS416-aac2(A128P,A137D) | This study | | See Materials and methods |
| Recombinant DNA reagent | pCDNA3.1 | This study | | See Materials and methods |
| Recombinant DNA reagent | pCDNA3.1-Ant1 | This study | | See Materials and methods |
| Recombinant DNA reagent | pCDNA3.1-Ant1(A90D) | This study | | See Materials and methods |
| Recombinant DNA reagent | pCDNA3.1-Ant1(L98P) | This study | | See Materials and methods |
| Recombinant DNA reagent | pCDNA3.1-Ant1(A114P) | This study | | See Materials and methods |
| Recombinant DNA reagent | pCDNA3.1-Ant1(A123D) | This study | | See Materials and methods |
| Recombinant DNA reagent | pCDNA3.1-Ant1(A90D,L98P) | This study | | See Materials and methods |
| Recombinant DNA reagent | pCDNA3.1-Ant1(A90D,A114P) | This study | | See Materials and methods |
| Recombinant DNA reagent | pCDNA3.1-Ant1(A90D,A123D) | This study | | See Materials and methods |
| Recombinant DNA reagent | pCDNA3.1-Ant1(L98P,A114P) | This study | | See Materials and methods |
| Recombinant DNA reagent | pCDNA3.1-Ant1(L98P,A123D) | This study | | See Materials and methods |
| Recombinant DNA reagent | pCDNA3.1-Ant1(A114P,A123D) | This study | | See Materials and methods |
| Recombinant DNA reagent | pCDNA3.1-Ant1(A90D,A114P,A123D) | This study | | See Materials and methods |
| Recombinant DNA reagent | pCDNA3.1-Ant1(A90D,L98P,A114P,A123D) | This study | | See Materials and methods |
| Recombinant DNA reagent | pGEM-4Z-AAC2 (*Saccharomyces cerevisiae*) | This study | | See Materials and methods |

*Appendix 1 Continued*

| Reagent type (species) or resource | Designation | Source or reference | Identifiers | Additional information |
|---|---|---|---|---|
| Recombinant DNA reagent | pGEM-4Z-aac2(A128P) | This study | | See Materials and methods |
| Recombinant DNA reagent | pGEM-4Z-aac2(A137D) | This study | | See Materials and methods |
| Recombinant DNA reagent | pGEM-4Z-aac2(A128P,A137D) | This study | | See Materials and methods |
| Software, algorithm | ImageJ | NIH | | |
| Software, algorithm | Multi Gauge v.3.2 | FujiFilm | | |
| Software, algorithm | Image Studio | LI-COR | | |
| Software, algorithm | Prism version 9 | GraphPad, LLC | | |
| Software, algorithm | Proteome Discoverer version 2.4 | Fisher | | |
| Software, algorithm | Metaboanalyst | *Pang et al., 2020* | | |
| Software, algorithm | STRING version 11.0 | *Szklarczyk et al., 2019* | | |
| Software, algorithm | Skyline version 20.2 | MacCoss Lab Software | | |
| Software, algorithm | CFX Maestro software | Bio-Rad | | |
| Software, algorithm | BioCIS | BIOSEB | | |
| Other | Grip Strength Test Model GT3 | BIOSEB | | Force meter that measures mouse grip strength. |
| Other | Oxygraph Plus System Version 2.1 | Hansatech Instruments | | Apparatus with Clark-type electrode to measure oxygen tension for oxygen consumption measurements from isolated mitochondria. |
| Commercial assay or kit | Clontech Labs 3P TaKaRa LA Taq DNA Polymerase | Fisher Scientific | Cat. #50-443-973 | |

