## [Editor Report]

This manuscript describes important insight into the molecular mechanism by which destabilized mitochondrial proteins 'clog' import channels and contribute to the pathologic mitochondrial and cellular dysfunction implicated in human disease. The evidence supporting this conclusion is convincing, utilizing yeast, mammalian cell culture, and mouse models. This work, which defines an interesting mechanism of disease pathogenesis, will be of broad interest to researchers in the fields of mitochondrial biology, protein quality control and proteostasis.

---

## [Decision Letter]

**Decision letter after peer review:**

Thank you for submitting your article "Mitochondrial protein import clogging as a mechanism of disease" for consideration by *eLife*. Your article has been reviewed by 3 peer reviewers, one of whom is a member of our Board of Reviewing Editors, and the evaluation has been overseen by Benoît Kornmann as the Senior Editor. The following individual involved in review of your submission has agreed to reveal their identity: Hilla Weidberg (Reviewer #3).

Essential revisions:

1) The differentiation of the loss-of-function and gain-of-toxicity phenotypes associated with mutant AAC2 expression are well separated in the yeast models. However, the evidence supporting the specific contributions of gain-of-toxicity phenotypes in mammalian cell culture and mice expressing mutant ANT1 is less convincing. Please provide additional evidence to support the importance of gain-of-toxicity, as opposed to loss-of-function phenotypes induced by mutant ANT1 overexpression in these models.

2) The impact of ANT1/AAC2 mutant expression on biological processes including cytosolic/mitochondrial proteostasis, integrated stress response signaling, and phospholipid regulation should be expanded in a revised manuscript.

3) Please expand on the discussion regarding the mechanistic basis for clogging of mitochondrial protein import channels, as outlined in the reviewer comments.

*Reviewer #1 (Recommendations for the authors):*

In this manuscript, the authors evaluate the potential for destabilizing, missense mutations in ANT1 to 'clog' the mitochondrial protein import pathway and promote pathologic mitochondrial dysfunction in the context of disease. Initially using a yeast system, they show that two independent mutations in the yeast ANT1, AAC2, obstructs protein import into the mitochondria monitoring import both in vitro and in yeast. They go on to show that mutant AAC2 is degraded by the IMM AAA+ protease YME1 in yeast, consistent with this protein being clogged in the import pathway and being partially accessible within the IMS. They then go on to demonstrate that expression of mutant ANT in HeLa cells similarly appears to similarly impair import. Finally, they generate a mouse model expressing a 'super-clogger' ANT containing two mutations linked to impaired import in yeast/cells. Using these mice, they show that expression of this superclogger impairs mitochondrial protein import, disrupts mitochondrial function, and promotes age-dependent mitochondrial myopathy phenotypes.

This is an interesting manuscript that highlights the potential for 'clogging' of import channels by mutant proteins to promote mitochondrial dysfunction in disease. One of the challenges with this study is deconvoluting potential loss-of-function phenotypes associated with reductions in ANT1/AAC2 from gain-of-toxicity phenotypes linked to import clogging. This was addressed primarily in yeast, showing that phenotypes associated with overexpression of mutants (e.g., reduced growth on glucose media). The experiment showing that yeast AAC2 clogs import was also convincing including both in vitro and in vivo characterization, although it isn't clear why the proteomic experiments were performed with acute expression of A128P instead of the 'superclogger' double mutant. The extension of this work to mammalian cells and then mice is also admirable. However, the quality of characterization does begin to decline when moving into mammalian models. For example, there is no clear evidence that observed phenotypes can be attributed to gain of toxicity instead of loss of function in mammalian cells and mice. There are similarities to yeast, but this needs to be better defined. Lastly, I have questions related to the mouse model, such as how do these phenotypes compare to KO animals and why were homozygous mice used in some situations and heterozygous mice used in others.

Ultimately, the strengths of this manuscript lie in the yeast work. While the expansion into cells and mice is admirable, those characterizations are a bit weaker. I do think that it would be useful to allow authors an opportunity to address these concerns with additional experimentation and revision of the text, and I do think with additional characterization this manuscript would be of interest to the readership of *eLife*.

1. Does yme1 deletion increase interactions between mutant AAC2 and TOM subunits? One would expect that stabilizing the mutant protein should show even larger increases in interactions with the import complex.

2. It isn't clear to me why the MS experiments described in Figure 4 weren't performed with the double mutant 'superclogger', especially as this one showed the most interaction with TOM40 in Figure 2. I assume this is because the double mutant has lower levels, but regardless, one would expect that the double mutant would show larger recovery of cytosolic chaperones, TOM subunits, and TIM23 subunits. Could the authors explain their choice here? They claim that the A128P mutant blocks import of HSP60 in Figure S2B, which is fair, but it doesn't appear to do that in Figure 2H. Further, A128P doesn't seem to induce the same phenotypes as the double mutant 'superclogger' (e.g., Figure 1C). If the phenotypes are observed with A128P, what is the relevance of these increased interactions with import pathway? How can this be linked to import clogging? I guess the argument is linked to the acute expression in this experiment, but it isn't clear to me.

3. The transition into human cells is intriguing, but lacks some of the same necessary characterization performed in yeast. It would be useful to confirm that ANT1 mutants are causing gain of toxicity and not loss-of function by performing similar experiments in HeLa cells deficient in ANT1 to that performed in yeast growing cells in normal media or respiratory media, etc.

4. Along the same lines, as #2 above, it isn't clear to me why the authors probed interactions between A114P and TOM/TIM22 instead of the double mutant. If issues are coming with clogging, then one could expect that recovery of TOM/TIM22 subunits would be increased in double mutant purifications independent of expression levels of this mutant.

5. I don't understand how clogging of the TOM complex could lead to substrate specificity in mammalian cell models. Could the authors expand on that? I don't think it needs to be solved in this paper, but it is important to provide some context around this point.

6. ANT1-deficient mice have been generated and induce mitochondrial defects in muscle. How do the phenotypes observed in heterozygous mice expressing the 'superclogger' compare to the knockout mice? The authors indicate that the increase in maximal respiration observed at 9 months indicates that the observed phenotypes are independent of ATP/ADP transport, but this needs to be better characterized. Chronic changes could account for this.

7. It isn't clear to me why the authors use the homozygous mice to evaluate clogging in vivo, while the hets were used for all of the phenotypes. If the goal is to demonstrate phenotypes are linked to clogging, then why wouldn't the authors evaluate clogging in the context of the same mice used for phenotyping? What happens to the homozygous mice from a phenotypic perspective? One would expect it is worse considering the larger impact on 'clogging'.

*Reviewer #2 (Recommendations for the authors):*

There are some caveats to this work that I summarize below. After these points are largely addressed the study is well suitable for publication in *eLife*.

Specific points

1. The abstract states "We propose that secondary structures of mitochondrial preproteins play an essential role in preventing clogging and disease". While it is tempting to generalize the authors' findings, this statement is not supported by presented data or discussion.

2. When describing Figure 1C, the authors state that cells co-expressing aac2 A128P, A137D and AAC2 "form petites at much higher frequency…". They should provide actual numbers and include quantification to support this statement.

3. Figure 4A. Why is the Aac2p A128P mutant toxic at 25C but not at 30C? This observation is basically left unexplained.

4. Does mutant Aac2p/ANT1 expression affect mitochondrial phospholipids? One additional scenario to consider would be an impaired membrane homeostasis.

5. The graph shown in Figure 5F is confusing and not well explained. What exactly do those multiple lines depict?

6. Do mutant Aac2/ANT1 variants affect stability of the TOM40 or TOM22 complexes?

7. Does expression of mutant ANT1 induce integrated stress response (ISR) and whether some of the cytotoxic effects observed could be mitigated by ISR inhibition?

8. Would treating mice with pharmacological inhibitors of protein import (e.g. MitoBloCK-1) phenocopy the effects of mutant ANT1 expression? This would strengthen the present study and ascertain generality of the proposed model.

My suggestions are generally summarized in the Specific points section. I feel that addressing the potential involvement of phospholipids, probing mutant ANT1 contribution to integrated stress response, and testing whether pharmacological inhibition of import in vivo could mimic the effects in question would significantly strengthen the present study and its conclusions.

*Reviewer #3 (Recommendations for the authors):*

1. The experiments convincingly show that Aac2/Ant1 mutants clog TOM. However, the possibility that the mutants cause other type of stress, such as aggregation in the mitochondria or toxic accumulation in the cytosol, is not excluded. Therefore, the connection between clogging and the observed phenotypes is correlative and not necessarily causative. In fact, expression of Aac2 mutant in yeast deleted of Tom70 was synthetic lethal. As Tom70 is required for the entry of Aac2 to mitochondria and the TOM channel, this result might suggest that the Aac2 toxicity is caused by a cytosolic and not mitochondrial population of the mutant. The authors conclusion that this genetic interaction supports the clogging activity is thus unclear.

It will be helpful to clarify: 1. Were all mice phenotypes excluded in Ant1+/- mice? 2. Is it possible that some of the phenotypes, such as neurodegeneration, are due to a cytosolic burden caused by the Ant1 mutant, or misfolding of this mutant in the mitochondria, rather than TOM clogging?

As the evidence are still correlative the authors should tone down their conclusion and emphasize that the other possibilities were not excluded.

2. Do Aac2/Ant1 variants clog TOM, TIM22, or both? Including hypotheses in the discussion session could be helpful. In particular in regards to stabilization of mutant aac2 in yme1 deletion and not in proteasome impaired conditions, which demonstrated that Aac2 is degraded in the IMS and raises the possibility that the degradation is coupled to association with TIM22.

3. The involvement of Yme1 in Aac2's degradation is novel and exciting but not very well developed. Does mutant Aac2 interact with Yme1? Does Yme1-medaited degradation require interaction of mutant Aac2 with small TIMs? Does it require interaction with TIM22? Does Yme1 has a role in degrading other type of mitochondrial cloggers, such as Cyb2DHFR?

The above might be beyond the scope of the manuscript but could be at least raised in the discussion. In addition, Yme1-mediated degradation was shown using overexpression of Aac2A128P,A137D mutant. It would be beneficial to show stabilization of overexpressed Aac2A128P and Aac2A137D single mutants as well.

4. Figure S5- An empty vector control is missing to confirm that the overexpression of wild type Aac2 has no effect. The time frame of induction between the RPN4, HSP82, SSA3, SSA4 and CIS1 is very different. Addition of time points between 4h and 24h could clarify this issue.

5. Figure 7I- lines 317-8- "Moreover, Smac also appears to be more sensitive to proteinase K in Ant1A114P,A123D/Ant1A114P,A123D mitochondria, while Mdh2 and Tim23 does not (Figure S7I-K)." Smac is not the only protein that seems to be sensitive to proteinase K: a. Mdh2 levels are decreased as well as its size (appears as a smaller band), b. Tim23 levels seems lower as well, although its signal is oversaturated. A quantification of all proteins +/- protease could test whether they are less sensitive to the protease than Ant mutants. Same applies to Figure 5D.

The authors added the following helpful comment: "While this could indicate instability or partial rupture of the OMM in mutant mitochondria, it could also be a result of clogging and imply that clogging does not affect all mitochondrial preproteins equally. Consistent with this concept, our experiments in yeast showed minimal effects of clogging on Ilv5, a mitochondrial matrix protein, in contrast to Hsp60 (Figure 1E; 2H-I)." I agree with this conclusion and add that Tim23 is a TIM22 substrate and might be more sensitive to clogging of TIM22.

6. Figure 7G- The cytosolic fraction was collected after a relatively low speed centrifugation and might contain more than soluble proteins. For example, if the Ant1 mutants induce mitophagy or formation of MDVs, such structures might contain small mitochondrial fractions that could be detected as "cytosol".

---

## [Author Response]

Essential revisions:1) The differentiation of the loss-of-function and gain-of-toxicity phenotypes associated with mutant AAC2 expression are well separated in the yeast models. However, the evidence supporting the specific contributions of gain-of-toxicity phenotypes in mammalian cell culture and mice expressing mutant ANT1 is less convincing. Please provide additional evidence to support the importance of gain-of-toxicity, as opposed to loss-of-function phenotypes induced by mutant ANT1 overexpression in these models.

We appreciate this important point regarding loss- vs gain-of-function mechanisms. To provide additional evidence to support gain-of-toxicity of Ant1^A114P,A123D^ (encoded by *Slc25a4 ^p.A114P,A123D^*), we profiled the transcriptome of *Slc25a4 ^p.A114P,A123D^*/+ mouse skeletal muscle, reasoning that if loss-of-function played a major role in Ant1^A114P,A123D^-induced pathology in mice, then the transcriptional signature should resemble that of *Slc25a4* knock-out mice (Morrow et al., 2017). *Slc25a4* knockout mouse muscle show a robust increase in the myokines *Fgf21* and *Gdf15* by up to 50-fold and 12-fold, respectively (Morrow et al., 2017). *Fgf21* was not even detected in two of four *Slc25a4 ^p.A114P,A123D^*/+ samples despite a sequencing depth of ~30 million reads/sample. *Gdf15* was reduced by 25% (*p* = 0.77). Pathway analyses of significantly upregulated genes in *Slc25a4* knockout muscle showed robust enrichment for genes involved in oxidative phosphorylation (OXPHOS) (Morrow et al., 2017). Using the same statistical tests and cutoffs, we didn’t find a single OXPHOS gene significantly upregulated in *Slc25a4 ^p.A114P,A123D^*/+ muscle (see Data S6). *Pgc-1*a, the transcriptional regulator of mitochondrial biogenesis, was upregulated by 2.46-fold in *Slc25a4* knockout muscle but unchanged in *Slc25a4 ^p.A114P,A123D^*/+. Again, using the same statistical cutoffs and enrichment analysis software, we compared the top 25 most enriched pathways in significantly up- and down-regulated genes, and found only a single pathway of the 50 per genotype overlapped (Figure 8—figure supplement 2G). Clearly, the transcriptional signature of *Slc25a4 ^p.A114P,A123D^*/+ skeletal muscle is entirely distinct from *Slc25a4* knockout, excluding the possibility that loss of Ant1 function is the major pathomechanism in *Slc25a4 ^p.A114P,A123D^*/+ mice.

Other comparisons with *Slc25a4* knockout mice are illuminating as well. Respirometry of mitochondria isolated from *Slc25a4 ^p.A114P,A123D^*/+ skeletal muscle showed a significant increase in maximal respiration rate when complex II is stimulated (Figure 6). As we noted in our original manuscript, this indicates that one functional copy of Ant1 is sufficient to support high respiratory rates, arguing against ADP/ATP transport as a limiting factor. What we failed to note in our original manuscript is that this is not observed in *Slc25a4* knockout mice; in fact, complex II-based respiration is reduced by >30% in *Slc25a4* knockout mice (Graham et al., Nature Genetics, 1997). We have now added this to the manuscript along with additional observations that strongly argue against loss of Ant1 function as a primary driver of *Slc25a4 ^p.A114P,A123D^*/+ muscle phenotypes.

Additional clinical context may be informative as well, and we have clarified this point in the revised manuscript. Patients heterozygous for dominant *SLC25A4* mutations, such as p.A114P, exhibit clinical features that are not present in patients with homozygous loss of ANT1 function, such as neurodegenerative and neuropsychiatric phenotypes (Deschauer et al., 2005; Echaniz-Laguna et al., 2012; Kashiki et al., 2022; Kaukonen et al., 2000; Napoli et al., 2001; Palmieri et al., 2005; Siciliano et al., 2003; Simoncini et al., 2017; Thompson et al., 2016; Tosserams et al., 2018). This strongly implies gain-of-function toxicity in patients. Moreover, some *Slc25a4 ^p.A114P,A123D^*/+ mice undergo frank neurodegeneration ending in paralysis and death within 2-3 weeks of symptom onset, while there are no neurological features of *Slc25a4* knockout mice. This is, in and of itself, a gain-of-function phenotype.

The transition to mammalian cell culture was intended to biochemically test if mutant ANT1 clogs protein import like yeast Aac2p. From this perspective, the original manuscript clearly demonstrated biochemical gain-of-function phenotypes, i.e. mutant ANT1 accumulation at the import machinery, mutant ANT1 being exposed on the outer membrane, and mitochondrial proteins accumulating in the cytosol. These biochemical phenotypes cannot reasonably be expected to result from loss of ANT1 function. We also note that total ANT levels (including ANT1, ANT2 and ANT3) can be drastically depleted in human cell culture without severely affecting OXPHOS or cell viability (Lu et al., 2017).

However, we absolutely agree with the broader point that at this stage we cannot completely exclude some contribution of haploinsufficiency in the context of clogging. We have adjusted our language in the manuscript accordingly.

2) The impact of ANT1/AAC2 mutant expression on biological processes including cytosolic/mitochondrial proteostasis, integrated stress response signaling, and phospholipid regulation should be expanded in a revised manuscript.

– To assess the impact on cytosolic proteostasis, we performed detailed analyses on the cytosolic proteome from both mammalian cell culture and mouse skeletal muscle, which showed modest increase in proteostatic stress signaling and chaperones, respectively. These data suggest possible cytosolic proteostatic stress. The data can be found in Figure 5H, Figure 5—figure supplement 1A-B, and Figure 8—figure supplement 1A-B.

– To assess the impact on mitochondrial proteostasis, we performed BN-PAGE analysis and found that, in mice, the respiratory complexes, respiratory supercomplexes, and Tim23 complex are minimally affected in *Slc25a4 ^p.A114P,A123D^*/+ mouse muscle mitochondria (Figure 8—figure supplement 1C-E). This suggests no severe effect on proteostasis on the inner mitochondrial membrane. We also performed quantitative proteomics on mitochondria isolated from mouse muscle (Figure 8—figure supplement 1F). While the mitochondrial proteome was overall minimally affected, we did observe a general increase in the levels of mitochondrial chaperones and proteases (Figure 8—figure supplement 1), suggesting there is at least a low level of mitochondrial proteostatic stress.

– To assess the impact on the integrated stress response (ISR), we performed western blot analysis of eIF2a and also performed RNA sequencing of *Slc25a4 ^p.A114P,A123D^*/+ mouse skeletal muscle, as described above. We found no evidence of ISR activation (Figure 8—figure supplement 2A-C).

– To explore the impact on phospholipid homeostasis, we used yeast genetics to test whether cells defective in cardioipin (CL) (*pel1D*) and/or phosphatidylethanolamine (PE) synthesis (*psd1D*) are hypersensitive to expression a super-clogger *aac2* variant. Indeed, we found that expression of *aac2^A128P,A137D^* strongly inhibited the growth of cells in both genetic backgrounds (Figure 4—figure supplement 3G-H). The data are consistent with previous observations that CL and PE facilitate mitochondrial protein import (Becker et al., 2013; Gebert et al., 2009). Alternatively, mitochondrial protein import clogging may affect phospholipid homoeostasis, which synergizes with *pel1D* and *psd1D* to cause cell lethality.

3) Please expand on the discussion regarding the mechanistic basis for clogging of mitochondrial protein import channels, as outlined in the reviewer comments.

We expanded discussion on the mechanistic basis for clogging in the revised manuscript.

Reviewer #1 (Recommendations for the authors):In this manuscript, the authors evaluate the potential for destabilizing, missense mutations in ANT1 to 'clog' the mitochondrial protein import pathway and promote pathologic mitochondrial dysfunction in the context of disease. Initially using a yeast system, they show that two independent mutations in the yeast ANT1, AAC2, obstructs protein import into the mitochondria monitoring import both in vitro and in yeast. They go on to show that mutant AAC2 is degraded by the IMM AAA+ protease YME1 in yeast, consistent with this protein being clogged in the import pathway and being partially accessible within the IMS. They then go on to demonstrate that expression of mutant ANT in HeLa cells similarly appears to similarly impair import. Finally, they generate a mouse model expressing a 'super-clogger' ANT containing two mutations linked to impaired import in yeast/cells. Using these mice, they show that expression of this superclogger impairs mitochondrial protein import, disrupts mitochondrial function, and promotes age-dependent mitochondrial myopathy phenotypes.This is an interesting manuscript that highlights the potential for 'clogging' of import channels by mutant proteins to promote mitochondrial dysfunction in disease. One of the challenges with this study is deconvoluting potential loss-of-function phenotypes associated with reductions in ANT1/AAC2 from gain-of-toxicity phenotypes linked to import clogging. This was addressed primarily in yeast, showing that phenotypes associated with overexpression of mutants (e.g., reduced growth on glucose media). The experiment showing that yeast AAC2 clogs import was also convincing including both in vitro and in vivo characterization, although it isn't clear why the proteomic experiments were performed with acute expression of A128P instead of the 'superclogger' double mutant. The extension of this work to mammalian cells and then mice is also admirable. However, the quality of characterization does begin to decline when moving into mammalian models. For example, there is no clear evidence that observed phenotypes can be attributed to gain of toxicity instead of loss of function in mammalian cells and mice. There are similarities to yeast, but this needs to be better defined. Lastly, I have questions related to the mouse model, such as how do these phenotypes compare to KO animals and why were homozygous mice used in some situations and heterozygous mice used in others.Ultimately, the strengths of this manuscript lie in the yeast work. While the expansion into cells and mice is admirable, those characterizations are a bit weaker. I do think that it would be useful to allow authors an opportunity to address these concerns with additional experimentation and revision of the text, and I do think with additional characterization this manuscript would be of interest to the readership of eLife.

We thank the reviewer for the thoughtful comments. Regarding the loss vs gain-of-function phenotypes, please see the discussion above. We agree with the reviewer that the extent of characterization in the mammalian systems is less than that in yeast, but note that the original manuscript did contain substantial data from mammalian cells (11 panels of data and 1 data file) and our novel mouse model (31 panels of data, 1 data file, and 2 movies). Nevertheless, we put forth a good-faith effort to better characterize the mammalian systems, as follows:

More extensive analysis of the cytosolic proteome in human cells (Figure 5H) and mouse muscle (Figure 8—figure supplement 1A-B).Additional isolation of mouse muscle mitochondria for native gel electrophoresis followed by Coomassie staining, which can visualize the mitochondrial respiratory complexes and supercomplexes (Figure 8—figure supplement 1C). This is coupled with native gel electrophoresis with western blot to further assess the mitochondrial complexes/supercomplexes and the TIM23 complex (Figure 8—figure supplement 1D-E)To further implement reviewer’s comments, we analyzed mitochondrial proteomics in *Slc25a4 ^p.A114P,A123D^*/+ mouse skeletal muscle (Figure 8—figure supplement 1F-H).Immunoblot validation of Tim22 increase suggested by mitochondrial proteomics data (Figure 8—figure supplement 1I-J).Immunoblot analysis of eIF2a in *Slc25a4 ^p.A114P,A123D^*/+ mouse skeletal muscle to assess integrated stress response signaling (Figure 8—figure supplement 2A-B)Finally, we used RNA-sequencing and determine transcriptional changes in *Slc25a4 ^p.A114P,A123D^*/+ mouse skeletal muscle (Figure 8—figure supplement 2C-G)

1. Does yme1 deletion increase interactions between mutant AAC2 and TOM subunits? One would expect that stabilizing the mutant protein should show even larger increases in interactions with the import complex.

We thank the reviewer for this excellent question. The interaction between mutant Aac2p and Yme1p is complex. With acute, inducible expression of mutant Aac2p (as in Figure 3), we would expect that yme1 deletion would increase Aac2p’s association with the TOM complex as suggested by the reviewer. However, deletion of Yme1p does not increase levels of chronically expressed Aac2p^A128P^ (Liu et al., 2015), and actually leads to the general reduction in the biogenesis of wild-type carrier proteins (Kumar et al., 2023). The mechanistic details of Yme1p-based proteolysis of mutant Aac2p and potential anti-clogging activity will be important areas for future research but may be beyond the scope of the current study.

2. It isn't clear to me why the MS experiments described in Figure 4 weren't performed with the double mutant 'superclogger', especially as this one showed the most interaction with TOM40 in Figure 2. I assume this is because the double mutant has lower levels, but regardless, one would expect that the double mutant would show larger recovery of cytosolic chaperones, TOM subunits, and TIM23 subunits. Could the authors explain their choice here?

The reviewer is correct, we elected not to pull down double mutant Aac2p for mass spectrometry of co-eluted proteins because of the extremely low protein accumulation. From a technical standpoint, this experiment is unlikely to yield any interpretable information. First, co-IP experiments with wild-type Aac2p/ANT1 is notorious for yielding a large number of proteins, many of which are likely associating nonspecifically due to the protein-dense, two-dimensional environment of the inner mitochondrial membrane (Claypool et al., 2008; Lu et al., 2017). Given this high background in wild-type Aac2p/ANT1, reducing the level of the protein >20-fold will even further reduce the signal-to-noise ratio. Second, the sensitivity of mass spectrometry compounds this problem. In our experiment, the control samples lacking a HIS tag still reliably eluted >1,500 proteins, hence our workflow to eliminate this increased noise, as shown in Figure 4—figure supplement 1B. In our view, repeating this experiment with the double mutant is unlikely to be helpful.

They claim that the A128P mutant blocks import of HSP60 in Figure S2B, which is fair, but it doesn't appear to do that in Figure 2H.

In Figure 2H, expression of the mutant is from a centromeric vector and cells were grown in minimal selective medium against plasmid loss. Growth in minimal medium is expected to globally reduce protein synthesis and alleviate cytosolic proteostatic stress (mPOS). In addition, in Figure 2—figure supplement 1B, the mutant *aac2* allele is expressed from a galactose-inducible promoter and the cells were grown in complete medium. We expect that the latter has higher expression of the mutant protein, which could also contribute to increased effect on protein import. However, we note that pre-Hsp60p does accumulate at steady state when mutant *aac2* is integrated into the genome and cells were grown in compete medium (Figure 2G), which may be most physiologically relevant.

Further, A128P doesn't seem to induce the same phenotypes as the double mutant 'superclogger' (e.g., Figure 1C). If the phenotypes are observed with A128P, what is the relevance of these increased interactions with import pathway? How can this be linked to import clogging? I guess the argument is linked to the acute expression in this experiment, but it isn't clear to me.

Regarding lack of severe phenotype of single mutants in Figure 1C: First, W303 is well known to be particularly tolerant to many mitochondrial insults, including single mutant *aac2* mutants (Wang et al., 2008a). Second, and perhaps relatedly, *aac2* mutants are rho-zero lethal in non-W303 strains, meaning these mutants cannot survive the loss of mtDNA. Thus, to (1) increase the probably that the strain is viable, and (2) allow us to observe the effect on mtDNA stability, we chose to integrate the double mutant into the W303 background.

The single mutants (including A128P) are highly toxic in M2915-6A background, which is the background in which we observed Aac2p^A128P^ accumulation at the translocase complexes (Figure 4). A more in-depth explanation of the strain-dependent effects of mitochondrial dysfunction on cell viability can be found in our response to Reviewer 3.

3. The transition into human cells is intriguing, but lacks some of the same necessary characterization performed in yeast. It would be useful to confirm that ANT1 mutants are causing gain of toxicity and not loss-of function by performing similar experiments in HeLa cells deficient in ANT1 to that performed in yeast growing cells in normal media or respiratory media, etc.

As noted above, severe depletion of all ADP/ATP transporter isoforms is surprisingly very well-tolerated by in mammalian cells in culture, even in regards to bioenergetics (Lu et al., 2017). The implication of this finding is that a very low level of ADP/ATP carriers is sufficient to support OXPHOS, which is surprising but consistent with our bioenergetic data from *Slc25a4 ^p.A114P,A123D^*/+ mice. In our view, performing CRISPR knockout of *SLC25A4* in HeLa cells followed by respiratory growth assay would have a limited contribution to the overall message of the manuscript, especially considering the considerable evidence of gain-of-toxicity and substantial bioenergetic characterization already present in mice.

4. Along the same lines, as #2 above, it isn't clear to me why the authors probed interactions between A114P and TOM/TIM22 instead of the double mutant. If issues are coming with clogging, then one could expect that recovery of TOM/TIM22 subunits would be increased in double mutant purifications independent of expression levels of this mutant.

Please see the above explanation (response to #2) explaining our decision not to immunoprecipitate Ant1^A114P,A123D^, which accumulates to just 2.2% of the wild-type level.

5. I don't understand how clogging of the TOM complex could lead to substrate specificity in mammalian cell models. Could the authors expand on that? I don't think it needs to be solved in this paper, but it is important to provide some context around this point.

This is a very interesting question. The data in this study suggest that there is substrate specific TOM complex clogging in both yeast and mammalian models. One likely contributor is the observation that the Tom40 b-barrel pore contains several distinct protein paths (Araiso et al., 2019; Shiota et al., 2015). Thus, although the pore is narrow, it is certainly conceivable that some protein paths are more affected than others. This explanation has been added to the manuscript (Lines 469-471).

6. ANT1-deficient mice have been generated and induce mitochondrial defects in muscle. How do the phenotypes observed in heterozygous mice expressing the 'superclogger' compare to the knockout mice? The authors indicate that the increase in maximal respiration observed at 9 months indicates that the observed phenotypes are independent of ATP/ADP transport, but this needs to be better characterized. Chronic changes could account for this.

We apologize for not making the phenotype comparison clearer in the original manuscript. A more rigorous comparison can be found above (in response to Essential Revisions) and in the revised manuscript (Figure 8—figure supplement 2G as well as an expanded Discussion section).

7. It isn't clear to me why the authors use the homozygous mice to evaluate clogging in vivo, while the hets were used for all of the phenotypes. If the goal is to demonstrate phenotypes are linked to clogging, then why wouldn't the authors evaluate clogging in the context of the same mice used for phenotyping? What happens to the homozygous mice from a phenotypic perspective? One would expect it is worse considering the larger impact on 'clogging'.

We bred a small number of homozygous mice to detect and characterize the low-abundance mutant protein, which is otherwise overwhelmed by the wildtype in heterozygous mice (Figure 8A-E). We deliberately refrained from phenotypic characterization of homozygous mice because the A123D mutation in Ant1^A114P,A123D^ renders the protein unable to transport ADP/ATP (see (Palmieri et al., 2005), which was corroborated in Figure 1C). Thus, disentangling gain-of-function vs loss of function in homozygous mice would be impossible.

Reviewer #2 (Recommendations for the authors):There are some caveats to this work that I summarize below. After these points are largely addressed the study is well suitable for publication in eLife.Specific points1. The abstract states "We propose that secondary structures of mitochondrial preproteins play an essential role in preventing clogging and disease". While it is tempting to generalize the authors' findings, this statement is not supported by presented data or discussion.

We thank the reviewer for pointing this out. We can understand how this statement may have been more misleading than we intended, as there are no structural data in the manuscript. We modified this statement to be more precise about what we do show, i.e. that amino acid substitutions in a carrier protein unexpectedly cause clogging.

2. When describing Figure 1C, the authors state that cells co-expressing aac2 A128P, A137D and AAC2 "form petites at much higher frequency…". They should provide actual numbers and include quantification to support this statement.

This experiment has been repeated, quantified, and depicted graphically in the revised manuscript (Figure 1E).

3. Figure 4A. Why is the Aac2p A128P mutant toxic at 25C but not at 30C? This observation is basically left unexplained.

Cold sensitivity of mutant *aac2* strains has been observed for many years (Wang et al., 2008a; Wang et al., 2008b). We previously found that, specifically at 25C, mutant *aac2* strains lose membrane potential which is associated with accumulation of the precursor form of Hsp60p (Wang et al., 2008b). With subsequent studies (Wang and Chen, 2015) and now the discovery of import clogging, it is likely the cold-induced reduction in membrane potential synergizes with clogging to further reduce protein import and challenge cytosolic proteostasis with the accumulation of mitochondrial precursors.

4. Does mutant Aac2p/ANT1 expression affect mitochondrial phospholipids? One additional scenario to consider would be an impaired membrane homeostasis.

This is an interesting idea that we previously investigated in single mutant *aac2,* hypothesizing that Aac2p misfolding causes membrane stress (Liu et al., 2015). To assess membrane protein homeostasis in double mutant Ant1 the mammalian systems, we performed a series of BN-PAGE experiments using human and mouse mitochondria. If membrane homeostasis is severely affected, we would expect a reduction in the assembly state of membrane protein complexes. However, we did not see a drastic effect in transfected HeLa cells (see Response Figure 1 below) or *Slc25a4 ^p.A114P,A123D^*/+ mice (Figure 8-fuigure supplement 1C-E). Thus, proteostatic stress on the IMM may contribute to cell stress independently of clogging or simply only occur in the yeast mutants.

Prompted by the comment on mitochondrial phospholipids, we also investigated the effect of mutant *aac2* expression in the context of defective mitochondrial phospholipid production. This was described above and the data can be found in the revised manuscript (Figure 4—figure supplement 3G-H).

5. The graph shown in Figure 5F is confusing and not well explained. What exactly do those multiple lines depict?

Each line was meant to connect the same protein found in the three samples (WT, ANT1^A114P^, and ANT1^A114P,A123D^-transfected cytosolic fractions). The idea was to show that there were a subset of proteins that were increased in ANT1^A114P^ cells compared to WT, and then further increased in ANT1^A114P,A123D^ cells. This is not an essential point so we elected to remove the lines to avoid further confusion.

6. Do mutant Aac2/ANT1 variants affect stability of the TOM40 or TOM22 complexes?

This is an interesting question, especially in light of our previous work that showed single mutant Aac2p proteins causing TIM22 destabilization (Liu et al., 2015). However, we did not observe this in transfected human cells, where mutant ANT1-transfected cells are essentially indistinguishable from wild-type (Response Figure 1). This suggests that IMM complexes in immortalized human cells may be less susceptible to mutant ANT1 relative to yeast.

**Author response image 1. sa2fig1:** Mutant ANT1 does not drastically affect the assembly state of TIM22 complex, the TOM complex, or respiratory Complex I in HeLa cells. (A-C) BN-PAGE followed by immunoblot analysis of isolated mitochondria from transfected HeLa cells. Protein ladders deduced from Coommassie-stained gel shown on the left. B and C are depicting the same membrane with sequential blotting. Regarding methods, mitochondrial isolation, BN-PAGE, and subsequent western blotting were performed 24 hours after transfection as previously described (Timon-Gomez et al., 2020) solubilizing the complexes with a 1:2 protein:digitonin ratio.

7. Does expression of mutant ANT1 induce integrated stress response (ISR) and whether some of the cytotoxic effects observed could be mitigated by ISR inhibition?

This is another very interesting question. As shown in the revised manuscript, we rigorously evaluated whether the ISR is activated in *Slc25a4 ^p.A114P,A123D^*/+ mice and found no evidence to support this (Figure 8—figure supplement 2A-F). The data suggest that clogging with Ant1^A114P,A123D^ is a unique and/or potentially mild stressor in vivo given the highly efficient degradation of the protein, as suggested by extremely low mutant protein accumulation (Figure 8A-C).

8. Would treating mice with pharmacological inhibitors of protein import (e.g. MitoBloCK-1) phenocopy the effects of mutant ANT1 expression? This would strengthen the present study and ascertain generality of the proposed model.

Treating mice with MitoBloCK-1 would be an extremely exciting experiment. We predict that its effect in vivo would largely depend on its bioavailability and tissue distribution, and may be unlikely to replicate the effects of mutant Ant1 because Ant1 expression is largely restricted to the muscle, heart and central nervous system. Given this caveat, the likely result of a distinct phenotype in MitoBloCK-1-treated mice would not provide much valuable information on the clogging model. Nevertheless, it could provide additional hypothesis-generating observations (which tissues are most susceptible to protein import defects and why?) so we thank the reviewer for this suggestion.

Reviewer #3 (Recommendations for the authors):1. The experiments convincingly show that Aac2/Ant1 mutants clog TOM. However, the possibility that the mutants cause other type of stress, such as aggregation in the mitochondria or toxic accumulation in the cytosol, is not excluded. Therefore, the connection between clogging and the observed phenotypes is correlative and not necessarily causative. In fact, expression of Aac2 mutant in yeast deleted of Tom70 was synthetic lethal. As Tom70 is required for the entry of Aac2 to mitochondria and the TOM channel, this result might suggest that the Aac2 toxicity is caused by a cytosolic and not mitochondrial population of the mutant. The authors conclusion that this genetic interaction supports the clogging activity is thus unclear.It will be helpful to clarify: 1. Were all mice phenotypes excluded in Ant1+/- mice? 2. Is it possible that some of the phenotypes, such as neurodegeneration, are due to a cytosolic burden caused by the Ant1 mutant, or misfolding of this mutant in the mitochondria, rather than TOM clogging?As the evidence are still correlative the authors should tone down their conclusion and emphasize that the other possibilities were not excluded.

We thank the reviewer for this important comment. We completely agree that we cannot exclude all possible mechanisms by which mutant Aac2p/ANT1 might be toxic, and have toned down our conclusion and emphasized our openness to other possibilities in the revised manuscript.

We also apologize for not making our working model clear in the manuscript. We expect that import clogging has far-reaching consequences, both intra- and extra-mitochondrial. Particularly in the cytosol, we absolutely expect that one of the effects of protein import clogging would be toxic mitochondrial protein accumulation and aggregation, which could include mutant Aac2p/ANT1 “waiting in the traffic jam”. This would be coherent with a model of clogging-induced toxicity.

With that said, there are some key experiments that argue against predominant cytosolic aggregation of mutant Aac2p/ANT1 independent of clogging. First, in our in vitro import experiments, the mutant proteins have increased association with the TOM complex, suggesting they are properly delivered to the mitochondria surface (Figure 2E-F). Second, we were unable to detect an increase in mutant Aac2p levels when using harsh solubilization conditions on whole cells (Figure 1—figure supplement 1B-C), suggesting the protein is not aggregated in the cytosol or mitochondria. We were similarly unable to extract any potentially aggregated quadruple mutant ANT1^A90D, L98P,A114P,A123D^ from transfected human cells (Liu et al., 2019). Thus, we think it’s unlikely that cytosolic aggregation of mutant Aac2p/ANT1 in the absence of clogging is a major mechanism of toxicity.

We thank the reviewer for her careful thinking regarding the genetic interaction between mutant *aac2* and *tom70D*. We would like to clarify that Tom70p is not required for the entry of Aac2 into mitochondria and, surprisingly, deletion of *TOM70* and its paralog *TOM71* only reduce Aac2p biogenesis by ~50% (Backes et al., 2021). Tom20p and Tom22p serve as receptors for carrier proteins in the absence of Tom70p (Steger et al., 1990; Yamano et al., 2008). In this case, the mutant Aac2 would still be able to engage in clogging. Interestingly, it was recently proposed that Tom70p’s primary function is to recruit chaperones to the mitochondrial surface to prevent proteotoxicity in this protein-dense region (Backes et al., 2021). Thus, deletion of *TOM70* may affect *aac2^A128P^* cells by two mechanisms: first, by further reducing carrier protein import and second by impairing cytosolic proteostasis in the context of pre-existing mitochondrial protein overaccumulation in the cytosol (Wang and Chen, 2015).

Regarding intramitochondrial protein aggregation, this was actually our original hypothesis for the mechanism of toxicity of double mutant Aac2p based on the aggregating properties of the mutant proteins in isolated mitochondria (Liu et al., 2015). However, we found no evidence of severe proteostatic stress on the inner mitochondrial membrane in the mammalian cells or mice, as discussed above.

Unfortunately, we are unable to exclude all phenotypes in *Slc25a4^+/-^* mice as we do not possess this mouse model. However, we do note that a neurodegenerative phenotype has never been reported in *Slc25a4^+/-^* or *Slc25a4^-/-^* mice despite >25 years of characterization by Doug Wallace and colleagues. Likewise, a myopathy has never been reported in *Slc25a4^+/-^* mice, consistent with normal clinical, biochemical and pathological features in heterozygous loss of function humans (Echaniz-Laguna et al., 2012).

2. Do Aac2/Ant1 variants clog TOM, TIM22, or both? Including hypotheses in the discussion session could be helpful. In particular in regards to stabilization of mutant aac2 in yme1 deletion and not in proteasome impaired conditions, which demonstrated that Aac2 is degraded in the IMS and raises the possibility that the degradation is coupled to association with TIM22.

We thank the reviewer for this helpful suggestion and have clarified our thoughts in the discussion in the revised manuscript.

3. The involvement of Yme1 in Aac2's degradation is novel and exciting but not very well developed. Does mutant Aac2 interact with Yme1? Does Yme1-medaited degradation require interaction of mutant Aac2 with small TIMs? Does it require interaction with TIM22? Does Yme1 has a role in degrading other type of mitochondrial cloggers, such as Cyb2DHFR?The above might be beyond the scope of the manuscript but could be at least raised in the discussion. In addition, Yme1-mediated degradation was shown using overexpression of Aac2A128P,A137D mutant. It would be beneficial to show stabilization of overexpressed Aac2A128P and Aac2A137D single mutants as well.

These are excellent questions for future investigation and we have added some of these ideas to the discussion in the revised manuscript.

4. Figure S5- An empty vector control is missing to confirm that the overexpression of wild type Aac2 has no effect. The time frame of induction between the RPN4, HSP82, SSA3, SSA4 and CIS1 is very different. Addition of time points between 4h and 24h could clarify this issue.

We apologize for not being clearer in the figure legend; this has been corrected. The strains used in Figure 4—figure supplement 3B-F (formerly Figure S5B-F) have a chromosomally integrated copy of wild-type and mutant *aac2* driven by the GAL10 promoter from the *trp1* locus.

5. Figure 7I- lines 317-8- "Moreover, Smac also appears to be more sensitive to proteinase K in Ant1A114P,A123D/Ant1A114P,A123D mitochondria, while Mdh2 and Tim23 does not (Figure S7I-K)." Smac is not the only protein that seems to be sensitive to proteinase K: a. Mdh2 levels are decreased as well as its size (appears as a smaller band), b. Tim23 levels seems lower as well, although its signal is oversaturated. A quantification of all proteins +/- protease could test whether they are less sensitive to the protease than Ant mutants. Same applies to Figure 5D.

We thank the reviewer for pointing this out. We agree, there does seem to be some sensitivity of TIM23 and MDH2 to proteinase K, but this is the case in wild-type mitochondria. and there is no difference between wild-type and mutant mitochondria. We have quantified all proteins +/- protease, relative to untreated, and there is no difference between wild-type and mutant samples. We moved these data from supplemental to a main figure to consolidate the data (Figure 8E-H).

Similarly, in Figure 5D, there is some level of sensitivity to proteinase K even in wild-type ANT1-transfected cells. For example, the mitochondrial matrix protein TFAM is reduced in each proteinase K-treated sample with or without swelling. This likely represents physical disruption of a fraction of mitochondria during isolation. We controlled for this by normalizing the ANT1-HA level to the level of TFAM. We appreciate the reviewer’s keen eye in this regard.

The authors added the following helpful comment: "While this could indicate instability or partial rupture of the OMM in mutant mitochondria, it could also be a result of clogging and imply that clogging does not affect all mitochondrial preproteins equally. Consistent with this concept, our experiments in yeast showed minimal effects of clogging on Ilv5, a mitochondrial matrix protein, in contrast to Hsp60 (Figure 1E; 2H-I)." I agree with this conclusion and add that Tim23 is a TIM22 substrate and might be more sensitive to clogging of TIM22.6. Figure 7G- The cytosolic fraction was collected after a relatively low speed centrifugation and might contain more than soluble proteins. For example, if the Ant1 mutants induce mitophagy or formation of MDVs, such structures might contain small mitochondrial fractions that could be detected as "cytosol".

This caveat has been added to the manuscript.

References:

Araiso, Y., Tsutsumi, A., Qiu, J., Imai, K., Shiota, T., Song, J., Lindau, C., Wenz, L.S., Sakaue, H., Yunoki, K.*, et al.* (2019). Structure of the mitochondrial import gate reveals distinct preprotein paths. Nature *575*, 395-401.

Backes, S., Bykov, Y.S., Flohr, T., Raschle, M., Zhou, J., Lenhard, S., Kramer, L., Muhlhaus, T., Bibi, C., Jann, C.*, et al.* (2021). The chaperone-binding activity of the mitochondrial surface receptor Tom70 protects the cytosol against mitoprotein-induced stress. Cell Rep *35*, 108936.

Becker, T., Horvath, S.E., Bottinger, L., Gebert, N., Daum, G., and Pfanner, N. (2013). Role of phosphatidylethanolamine in the biogenesis of mitochondrial outer membrane proteins. J Biol Chem *288*, 16451-16459.

Claypool, S.M., Oktay, Y., Boontheung, P., Loo, J.A., and Koehler, C.M. (2008). Cardiolipin defines the interactome of the major ADP/ATP carrier protein of the mitochondrial inner membrane. J Cell Biol *182*, 937-950.

Deschauer, M., Hudson, G., Muller, T., Taylor, R.W., Chinnery, P.F., and Zierz, S. (2005). A novel ANT1 gene mutation with probable germline mosaicism in autosomal dominant progressive external ophthalmoplegia. Neuromuscul Disord *15*, 311-315.

Dimitrov, L.N., Brem, R.B., Kruglyak, L., and Gottschling, D.E. (2009). Polymorphisms in multiple genes contribute to the spontaneous mitochondrial genome instability of *Saccharomyces cerevisiae* S288C strains. Genetics *183*, 365-383.

Echaniz-Laguna, A., Chassagne, M., Ceresuela, J., Rouvet, I., Padet, S., Acquaviva, C., Nataf, S., Vinzio, S., Bozon, D., and Mousson de Camaret, B. (2012). Complete loss of expression of the ANT1 gene causing cardiomyopathy and myopathy. J Med Genet *49*, 146-150.

Gebert, N., Joshi, A.S., Kutik, S., Becker, T., McKenzie, M., Guan, X.L., Mooga, V.P., Stroud, D.A., Kulkarni, G., Wenk, M.R.*, et al.* (2009). Mitochondrial cardiolipin involved in outer-membrane protein biogenesis: implications for Barth syndrome. Curr Biol *19*, 2133-2139.

Kashiki, T., Kido, J., Momosaki, K., Kusunoki, S., Ozasa, S., Nomura, K., Imai-Okazaki, A., Tsuruoka, T., Murayama, K., Koga, Y.*, et al.* (2022). Mitochondrial DNA depletion syndrome with a mutation in SLC25A4 developing epileptic encephalopathy: A case report. Brain Dev *44*, 56-62.

Kaukonen, J., Juselius, J.K., Tiranti, V., Kyttala, A., Zeviani, M., Comi, G.P., Keranen, S., Peltonen, L., and Suomalainen, A. (2000). Role of adenine nucleotide translocator 1 in mtDNA maintenance. Science *289*, 782-785.

Kumar, A., Waingankar, T.P., and D'Silva, P. (2023). Functional crosstalk between the TIM22 complex and YME1 machinery maintains mitochondrial proteostasis and integrity. J Cell Sci *136*.

Liu, Y., Wang, X., and Chen, X.J. (2015). Misfolding of mutant adenine nucleotide translocase in yeast supports a novel mechanism of Ant1-induced muscle diseases. Mol Biol Cell *26*, 1985-1994.

Liu, Y., Wang, X., Coyne, L.P., Yang, Y., Qi, Y., Middleton, F.A., and Chen, X.J. (2019). Mitochondrial carrier protein overloading and misfolding induce aggresomes and proteostatic adaptations in the cytosol. Mol Biol Cell *30*, 1272-1284.

Lu, Y.W., Acoba, M.G., Selvaraju, K., Huang, T.C., Nirujogi, R.S., Sathe, G., Pandey, A., and Claypool, S.M. (2017). Human adenine nucleotide translocases physically and functionally interact with respirasomes. Mol Biol Cell *28*, 1489-1506.

Morrow, R.M., Picard, M., Derbeneva, O., Leipzig, J., McManus, M.J., Gouspillou, G., Barbat-Artigas, S., Dos Santos, C., Hepple, R.T., Murdock, D.G.*, et al.* (2017). Mitochondrial energy deficiency leads to hyperproliferation of skeletal muscle mitochondria and enhanced insulin sensitivity. Proc Natl Acad Sci U S A *114*, 2705-2710.

Napoli, L., Bordoni, A., Zeviani, M., Hadjigeorgiou, G.M., Sciacco, M., Tiranti, V., Terentiou, A., Moggio, M., Papadimitriou, A., Scarlato, G.*, et al.* (2001). A novel missense adenine nucleotide translocator-1 gene mutation in a Greek adPEO family. Neurology *57*, 2295-2298.

Palmieri, L., Alberio, S., Pisano, I., Lodi, T., Meznaric-Petrusa, M., Zidar, J., Santoro, A., Scarcia, P., Fontanesi, F., Lamantea, E.*, et al.* (2005). Complete loss-of-function of the heart/muscle-specific adenine nucleotide translocator is associated with mitochondrial myopathy and cardiomyopathy. Hum Mol Genet *14*, 3079-3088.

Shiota, T., Imai, K., Qiu, J., Hewitt, V.L., Tan, K., Shen, H.H., Sakiyama, N., Fukasawa, Y., Hayat, S., Kamiya, M.*, et al.* (2015). Molecular architecture of the active mitochondrial protein gate. Science *349*, 1544-1548.

Siciliano, G., Tessa, A., Petrini, S., Mancuso, M., Bruno, C., Grieco, G.S., Malandrini, A., DeFlorio, L., Martini, B., Federico, A.*, et al.* (2003). Autosomal dominant external ophthalmoplegia and bipolar affective disorder associated with a mutation in the ANT1 gene. Neuromuscul Disord *13*, 162-165.

Simoncini, C., Siciliano, G., Tognoni, G., and Mancuso, M. (2017). Mitochondrial ANT-1 related adPEO leading to cognitive impairment: is there a link? Acta Myol *36*, 25-27.

Steger, H.F., Sollner, T., Kiebler, M., Dietmeier, K.A., Pfaller, R., Trulzsch, K.S., Tropschug, M., Neupert, W., and Pfanner, N. (1990). Import of ADP/ATP carrier into mitochondria: two receptors act in parallel. J Cell Biol *111*, 2353-2363.

Thompson, K., Majd, H., Dallabona, C., Reinson, K., King, M.S., Alston, C.L., He, L., Lodi, T., Jones, S.A., Fattal-Valevski, A.*, et al.* (2016). Recurrent de novo Dominant Mutations in SLC25A4 Cause Severe Early-Onset Mitochondrial Disease and Loss of Mitochondrial DNA Copy Number. Am J Hum Genet *99*, 860-876.

Timon-Gomez, A., Perez-Perez, R., Nyvltova, E., Ugalde, C., Fontanesi, F., and Barrientos, A. (2020). Protocol for the Analysis of Yeast and Human Mitochondrial Respiratory Chain Complexes and Supercomplexes by Blue Native Electrophoresis. STAR Protoc *1*.

Tosserams, A., Papadopoulos, C., Jardel, C., Lemiere, I., Romero, N.B., De Lonlay, P., Wahbi, K., Voermans, N., Hogrel, J.Y., and Laforet, P. (2018). Two new cases of mitochondrial myopathy with exercise intolerance, hyperlactatemia and cardiomyopathy, caused by recessive SLC25A4 mutations. Mitochondrion *39*, 26-29.

Wang, X., and Chen, X.J. (2015). A cytosolic network suppressing mitochondria-mediated proteostatic stress and cell death. Nature *524*, 481-484.

Wang, X., Salinas, K., Zuo, X., Kucejova, B., and Chen, X.J. (2008a). Dominant membrane uncoupling by mutant adenine nucleotide translocase in mitochondrial diseases. Hum Mol Genet *17*, 4036-4044.

Wang, X., Zuo, X., Kucejova, B., and Chen, X.J. (2008b). Reduced cytosolic protein synthesis suppresses mitochondrial degeneration. Nat Cell Biol *10*, 1090-1097.

Yamano, K., Yatsukawa, Y., Esaki, M., Hobbs, A.E., Jensen, R.E., and Endo, T. (2008). Tom20 and Tom22 share the common signal recognition pathway in mitochondrial protein import. J Biol Chem *283*, 3799-3807.